# Grokking Finite-Dimensional Algebra

**Pascal Jr Tikeng Notsawo** [1 2]   **Guillaume Dumas** [1 2 3]   **Guillaume Rabusseau** [1 2 4]

## Abstract

This paper investigates the grokking phenomenon, which refers to the sudden transition from a long memorization to generalization observed during neural networks training, in the context of learning multiplication in finite-dimensional algebras (FDA). While prior work on grokking has focused mainly on group operations, we extend the analysis to more general algebraic structures, including non-associative, non-commutative, and non-unital algebras. We show that learning group operations is a special case of learning FDA, and that learning multiplication in FDA amounts to learning a bilinear product specified by the algebra's structure tensor. For algebras over the reals, we connect the learning problem to matrix factorization with an implicit low-rank bias, and for algebras over finite fields, we show that grokking emerges naturally as models must learn discrete representations of algebraic elements. This leads us to experimentally investigate the following core questions: (i) how do algebraic properties such as commutativity, associativity, and unitality influence both the emergence and timing of grokking, (ii) how structural properties of the structure tensor of the FDA, such as sparsity and rank, influence generalization, and (iii) to what extent generalization correlates with the model learning latent embeddings aligned with the algebra's representation. Our work provides a unified framework for grokking across algebraic structures and new insights into how mathematical structure governs neural network generalization dynamics.

## 1. Introduction

One of the primary objectives of machine learning is generalization, which refers to the ability of models to per-

[1] Université de Montréal [2]Mila, Montréal [3]CHU Sainte-Justine Research Center, Montréal [4]CIFAR AI Chair. Correspondence to: Pascal Jr Tikeng Notsawo <pascalnotsawo@gmail.com>.

*Proceedings of the $43^{rd}$ International Conference on Machine Learning*, Seoul, South Korea. PMLR 306, 2026. Copyright 2026 by the author(s).

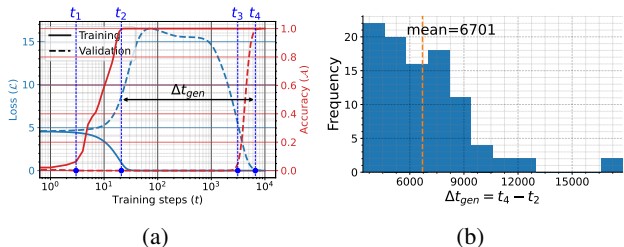

*Figure 1.* **(a)** Grokking (non-trivial $\Delta t_{\text{gen}} = t_4 - t_2$) on a 2-layer MLP with ReLU activation trained to learn multiplication in the algebra of complex numbers over the field $\mathbb{F}_7 = \mathbb{Z}/7\mathbb{Z}$, where $(a+b\mathbf{i})(c+d\mathbf{i}) = ac+bd\mathbf{i}^2+(ad+bc)\mathbf{i}$ and $\mathbf{i}^2 = -1 \equiv 6 \bmod 7$. **(b)** Histogram of grokking delay $\Delta t_{\text{gen}}$ under single modifications of the multiplication rules of complex numbers in $\mathbb{F}_7$, where each perturbation consists in changing the outcome of one multiplication relation (e.g., replacing $\mathbf{i}^2 = -1$ with $\mathbf{i}^2 = 1$).

form well on data they have not encountered during training. Even if these models perform exceptionally well in different domains today, the underlying reason for their good generalization ability remains an open question. By introducing the grokking phenomenon, Power et al. (2022) recently provides a new lens to view the generalization capabilities of overparametrized neural networks, challenging previous notions that such networks merely memorize training data.

The grokking phenomenon has been studied mainly on algorithmic tasks, which consist of predicting $z = x \circ y$ given $(x, y) \in \mathcal{S}^2$, with $\circ$ a binary mathematical operator and $\mathcal{S}$ a finite set of discrete symbols with no internal structure; for instance, finite cyclic groups of order 1 with its binary operation $\circ$ (such as addition in $\mathcal{S} = \mathbb{Z}/p\mathbb{Z}$ for some prime integer $p$). In such settings, it is now established that models like Transformers generalize by recovering the representation of the group itself (Gromov, 2023; Nanda et al., 2023; Chughtai et al., 2023; Stander et al., 2024). In this work, we investigate how grokking extends beyond these well-understood settings in terms of algebraic complexity, i.e, moving from groups to more general algebras. Our goal is to understand how specific algebraic or structural properties of the task influence the emergence and timing of the grokking phenomenon.

**Motivations**   While group operations provide a basic structure, algebras encompass a richer and more complex set of operations, including non-unital, non-commutative, and

non-associative behaviors. For example, many systems (in physics, robotics, control, etc) evolve according to infinitesimal transformations whose interaction is governed by a Lie bracket $[\cdot, \cdot]$, a bilinear, non-commutative, non-associative, and non-unital operation. Predicting such infinitesimal updates corresponds to learning a FDA.

In drug discovery, predicting how molecules interact with biological targets is crucial for identifying potential drug candidates. This involves understanding the complex relationships and transformations between different molecular structures. An example of a task is chemical reaction prediction (Fooshee et al., 2018), where given two molecules $x$ and $y$, the task is to predict the product $xy$ when these molecules undergo a chemical reaction (and the reaction pathway). If a third molecule $z$ is involved, the reaction sequence $(xy)z$ might yield a different product than $x(yz)$. Also, chemical reactions often involve complex mechanisms that cannot be simply reversed or do not have a natural identity element (a molecule that would leave another unchanged in a reaction).

In natural language processing, understanding how the meanings of smaller units (words or phrases) combine to form the meanings of larger units (sentences or documents) is a fundamental challenge. This process is inherently compositional, similar to operations in an algebraic structure where the operation combines elements to produce another element within the same set, and for which the order in which the elements are taken is very important. Consider the semantic composition and decomposition tasks (Turney, 2014) as a concrete example. The first task involves understanding the meaning of a text by composing the meanings of the individual words in this text, and the second involves understanding the meaning of an individual word by decomposing it into various aspects that are latent in the word's meaning. For example, given a noun-modifier bigram 'x y', with 'x' the head noun and 'y' the noun or adjective, the semantic composition task can be to find a noun unigram 'z' in the vocabulary which is a synonym of 'x y'. Conversely, the semantic decomposition can be to find a bigram 'x y' synonym to a given 'z' from the vocabulary. Examples of (x y, z) are (presidential term, presidency), (electrical power, wattage), (milk sugar, lactose), (bass fiddle, contrabass), etc. This kind of problem is often solved by optimizing $\arg\max_{\mathbf{z}} g(f(\mathbf{x}, \mathbf{y}), \mathbf{z})$, where $\mathbf{x}$, $\mathbf{y}$ and $\mathbf{z}$ are vectors representing the words 'x', 'y' and 'z'; $f$ a function representing 'x y' from $\mathbf{x}$ and $\mathbf{y}$; and $g$ a measure of similarity. Using a symmetric $f$ is common, such as $f(\mathbf{x}, \mathbf{y}) = \mathbf{x} + \mathbf{y}$, but they are order independent and gives the same representation to 'x y' and 'y x', so that, for example, "good morning" = "morning good" under $f$. Landauer (2002) estimates that $80\%$ of the meaning of English text comes from word choice, and the remaining $20\%$ comes from word order. Therefore, methods like vector addition, which do not account for word order, potentially overlook at least $20\%$

of the meaning in a bigram (Turney, 2014). So, the order of words affects the meaning; "good morning" conveys a different sentiment and usage than "morning good". For associativity, when combining more than two words, the order in which they are combined can affect the resulting meaning. For example, combining words in the phrase "old country house" can yield different interpretations based on which words are combined first.

**Contributions**   Our contributions are fourfold:

- Leveraging the fact that any tensor $\mathcal{C} \in \mathbb{F}^{n \times n \times n}$ defines an $n$-dimensional algebra over a field $\mathbb{F}$ ($n\mathbb{F}$-FDA), with associativity, commutativity, and unitality encoded as tensor identities, and matrix representations characterized by explicit tensor constraints; we establishe the equivalence between $\mathbb{F}^{n \times n \times n}$ and all $n\mathbb{F}$-FDA, allowing us to focus on structure tensors rather than abstract algebras.

- We show that prior work on grokking in groups (e.g., (Nanda et al., 2023), (Chughtai et al., 2023)) is a special case of our tensorial framework, which naturally extends to non-associative, non-commutative, and non-unital algebras.

- We connect FDA learning over $\mathbb{R}$ to matrix factorization, while over finite fields grokking emerges when models learn explicit embeddings of FDA elements.

- We provide the first systematic study of grokking beyond groups, showing that algebraic constraints (associativity, commutativity, unitality) and structural features of the structure tensor (sparsity, rank) significantly affect grokking delay and sample complexity, and that generalization coincides with the emergence of algebra-consistent representations in model layers[1].

**Related Work**   On finite groups like $\mathbb{Z}/p\mathbb{Z}$ and $S_n$, it has been shown that models like Transformers grok by recovering the representation of the group itself (Gromov, 2023; Nanda et al., 2023; Chughtai et al., 2023; Stander et al., 2024). On the algorithmic task (Power et al., 2022), Liu et al. (2022) shows representation learning to be the main underlying factor of the existence of generalizing solutions for modular arithmetic, supporting Power et al. (2022)'s preliminary observations. Gromov (2023) write out the exact-form expression of the weights of a 2-layer MLP (with activation function $x \mapsto x^2$) that yield $100\%$ accuracy on modular addition. Nanda et al. (2023) write out the closed-form solution of final trained weights in the case of a one-layer transformer trained on the same task, which is

---

[1]The code to reproduce all figures is available at https://github.com/Tikquuss/grokking_fda.

also made of sinusoids as in (Gromov, 2023). Chughtai et al. (2023) reverse engineer (small) neural networks learning finite group composition via mathematical representation theory. Stander et al. (2024) do the same, with the focus on composition in the symmetric group $S_5$.

To the best of our knowledge, we are the first to study grokking in the general setting of FDAs. Prior work on grokking studies algorithmic tasks almost exclusively through group operations. These structures enforce a rigid bundle of properties (associativity, unitality, etc), making it impossible to isolate which algebraic traits influence grokking and whether these properties generalize beyond groups. Our FDA framework provides the first setting in which all previously studied group tasks embed naturally, and non-unital and non-associative operations become accessible. Our experiments show, for instance, that associativity, commutativity, and unitality each induce distinct grokking delays and generalization patterns, and that these patterns persist across all the algebras sampled from each regime.

**Structure of the document** Section 2 introduces the structure-tensor view of FDA, characterizes algebraic properties as tensor identities, and provides a criterion for matrix representations. Section 3 shows how the framework subsumes groups, and Section 4 presents the learning formulation together with its theoretical motivation. Experimental results and discussion are given in Section 5 and Section 6, respectively, before concluding in Section 7.

**Notations** For $n \in \mathbb{N}^*$, $[n] := \{1, \ldots, n\}$. For $\mathbf{A} \in \mathbb{R}^{m \times n}$, $\text{vec}(\mathbf{A}) \in \mathbb{R}^{mn}$ denotes the vector obtained by stacking the columns of $\mathbf{A}$. The $i$th row and column of $\mathbf{A}$ are denoted by $\mathbf{A}_{i,:}$ (or $\mathbf{A}_i$) and $\mathbf{A}_{:,i}$, respectively. This notation extends naturally to tensor slices. We denote by $\odot$ the Hadamard product, $\otimes$ the Kronecker product, $\star$ the Khatri–Rao product, $\bullet$ the face-splitting product, and $\times_n$ the mode-$n$ tensor–vector product (Kolda & Bader, 2009).

## 2. Finite Dimensional Algebra

In this section, we work with standard notions from FDA and adopt a concrete tensorial viewpoint. Fixing a basis, any $n$-dimensional algebra is identified with a structure tensor $\mathcal{C} \in \mathbb{F}^{n \times n \times n}$. This formulation provides a unified and operational description of algebraic structure that is directly compatible with our setting. In particular, it allows us to treat $\mathcal{C}$ as a concrete object that can be compared with learned representations, forming the basis for the alignment perspective developed later in the paper.

### 2.1. Definitions

An **algebra** $\mathfrak{A}$ over a field $(\mathbb{F}, +, *)$ is a vector space equipped with a $\mathbb{F}$-bilinear product $\cdot : \mathfrak{A} \times \mathfrak{A} \to \mathfrak{A}$ such

that $(\mathbf{u} + \mathbf{v}) \cdot \mathbf{w} = \mathbf{u} \cdot \mathbf{w} + \mathbf{v} \cdot \mathbf{w}$ (right distributivity), $\mathbf{w} \cdot (\mathbf{u} + \mathbf{v}) = \mathbf{w} \cdot \mathbf{u} + \mathbf{w} \cdot \mathbf{v}$ (left distributivity) and $(\alpha \mathbf{u}) \cdot (\beta \mathbf{v}) = (\alpha * \beta)(\mathbf{u} \cdot \mathbf{v})$ (compatibility with scalars) for all $\mathbf{u}, \mathbf{v}, \mathbf{w} \in \mathfrak{A}$ and $\alpha, \beta \in \mathbb{F}$. When $\cdot$ is associative (resp. commutative), $\mathfrak{A}$ is said to be associative (resp. commutative). When there exists an element $1_{\mathfrak{A}} \in \mathfrak{A}$ such that $1_{\mathfrak{A}} \cdot \mathbf{u} = \mathbf{u} \cdot 1_{\mathfrak{A}} = \mathbf{u} \; \forall \mathbf{u} \in \mathfrak{A}$, $\mathfrak{A}$ is said to be unital. The dimension of $\mathfrak{A}$ is its dimension as a $\mathbb{F}$-vector space, denoted by $\dim_{\mathbb{F}} \mathfrak{A}$. We say that $\mathfrak{A}$ is finite-dimensional if $n = \dim_{\mathbb{F}} \mathfrak{A}$ is finite. This means that there exists a finite basis $\{\mathbf{a}^{(i)}\}_{i \in [n]}$ of $\mathfrak{A}$ (as a vector space over $\mathbb{F}$) such that for every $\mathbf{u} \in \mathfrak{A}$, $\mathbf{u} = \sum_{i=1}^n \alpha_i \mathbf{a}^{(i)}$ with $\alpha_i \in \mathbb{F}$.

Let $(\mathfrak{A}, \cdot)$ and $(\mathfrak{B}, \times)$ be two $\mathbb{F}$-algebras. A homomorphism of $\mathbb{F}$-algebras $\phi : \mathfrak{A} \to \mathfrak{B}$ is a $\mathbb{F}$-linear map satisfying $\phi(\mathbf{u} \cdot \mathbf{v}) = \phi(\mathbf{u}) \times \phi(\mathbf{v})$ for all $\mathbf{u}, \mathbf{v} \in \mathfrak{A}$; and $\phi(1_{\mathfrak{A}}) = 1_{\mathfrak{B}}$ when $\mathfrak{A}$ and $\mathfrak{B}$ are unitals. This homomorphism $\phi$ is called an isomorphism if it is bijective. Two $\mathbb{F}$-algebras $\mathfrak{A}$ and $\mathfrak{B}$ are isomorphic if there exists an algebra isomorphism $\phi : \mathfrak{A} \to \mathfrak{B}$ between them. In other words, $\mathfrak{A}$ and $\mathfrak{B}$ may look different at the level of their elements or chosen bases, but they share exactly the same algebraic structure.

In the following, $\mathbb{F}$-FDA means FDA over the field $\mathbb{F}$, and $n\mathbb{F}$-FDA means $\mathbb{F}$-FDA of dimension $n$.

### 2.2. Structure Constants of FDA

Let $(\mathfrak{A}, \cdot)$ be an $n\mathbb{F}$-FDA and $B = \{\mathbf{a}^{(i)}\}_{i \in [n]}$ a basis of $\mathfrak{A}$. The structure of $\mathfrak{A}$ in $B$ is entirely captured by the tensor $\mathcal{C}^{(B)} \in \mathbb{F}^{n \times n \times n}$ defined by $\mathbf{a}^{(i)} \cdot \mathbf{a}^{(j)} = \sum_{k=1}^n \mathcal{C}^{(B)}_{ijk} \mathbf{a}^{(k)}$. The entries of $\mathcal{C}^{(B)}$ are called structure coefficients/constants of $\mathfrak{A}$ in $B$. We will call the tensor $\mathcal{C}^{(B)}$ the **structure tensor** of $\mathfrak{A}$ in the basis $B$, and when the context is clear, we will omit $B$ from the notation. That said, we have for all $\mathbf{u} = \sum_{i=1}^n \mathbf{u}_i \mathbf{a}^{(i)}$ and $\mathbf{v} = \sum_{i=1}^n \mathbf{v}_i \mathbf{a}^{(i)}$ in $\mathfrak{A}$, $\mathbf{u} \cdot \mathbf{v} = \sum_{k=1}^n (\mathcal{C} \times_1 \mathbf{u} \times_2 \mathbf{v})_k \mathbf{a}^{(k)}$ with

$$\mathcal{C} \times_1 \mathbf{u} \times_2 \mathbf{v} := \left[ \mathbf{u}^\top \mathcal{C}_{:,:,k} \mathbf{v} \right]_{k \in [n]}^\top = \mathcal{C}_{(3)} (\mathbf{v} \otimes \mathbf{u}) \quad (1)$$

In this equation, $\mathcal{C}_{(3)} \in \mathbb{F}^{n \times n^2}$ is the mode-3 unfolding of $\mathcal{C}$: $\left( \mathcal{C}_{(3)} \right)_{k,:} = \text{vec}(\mathcal{C}_{:,:,k}) \in \mathbb{F}^{n^2} \; \forall k \in [n]$.

*Example* 2.1. The complex numbers $\mathbb{C} = (\mathbb{R}^2, \cdot)$ can be seen as a $2\mathbb{R}$-FDA, with the product $\cdot$ defined as $(a + b\mathbf{i}) \cdot (c + d\mathbf{i}) = (ac - bd) + (ad + bc)\mathbf{i}$, where $\mathbf{i}^2 = -1$. The structure tensor of $\mathbb{C}$ in $(\mathbf{a}^{(1)}, \mathbf{a}^{(2)}) \equiv (1, \mathbf{i})$ is

$$\begin{bmatrix} \mathcal{C}_{111} & \mathcal{C}_{112} \\ \mathcal{C}_{121} & \mathcal{C}_{122} \end{bmatrix}, \begin{bmatrix} \mathcal{C}_{211} & \mathcal{C}_{212} \\ \mathcal{C}_{221} & \mathcal{C}_{222} \end{bmatrix} = \begin{bmatrix} 1 & 0 \\ 0 & 1 \end{bmatrix}, \begin{bmatrix} 0 & 1 \\ -1 & 0 \end{bmatrix} \quad (2)$$

In the field $\mathbb{F} = \mathbb{F}_p := \mathbb{Z}/p\mathbb{Z}$ for $p$ prime (or power of prime), we have $\mathbf{i}^2 = (-1) \mod p = p - 1$ instead, so $\mathcal{C}_{221} = p - 1$ (the other coefficients remain unchanged).

One may ask whether every tensor $\mathcal{C} \in \mathbb{F}^{n \times n \times n}$ defines an $n\mathbb{F}$-FDA. If this holds, then one could in principle sam-

ple the entries of $\mathcal{C}$ over $\mathbb{F}$ and study the resulting algebra through the properties of its structure tensor. The following proposition shows that analyzing the space $\mathbb{F}^{n \times n \times n}$ is equivalent to studying all $n\mathbb{F}$-FDA. What remains open, however, is how to sample $\mathcal{C}$ in a principled way so as to obtain meaningful and general insights about the phenomena of interest. We return to this issue in Section 4.

**Proposition 2.1.** *For all $\mathcal{C} \in \mathbb{F}^{n \times n \times n}$, there exists an $n\mathbb{F}$-FDA whose structure tensor is $\mathcal{C}$ in some basis. Moreover, all algebras that have $\mathcal{C}$ as a structure tensor in one of their bases are isomorphic to each other.*

The proof of this proposition is constructive. We define on the vector space $\mathbb{F}^n$ the bilinear map $\mathbf{u} \times \mathbf{v} = \mathcal{C} \times_1 \mathbf{u} \times_2 \mathbf{v}$, and show that $(\mathbb{F}^n, \times)$ is an $n\mathbb{F}$-FDA with structure tensor $\mathcal{C}$ in the canonical basis $\{\mathbf{e}^{(i)}\}_{i \in [n]}$, $\mathbf{e}_j^{(i)} = \delta_{ij}$ (Lemma B.1). We call this algebra the algebra generated by $\mathcal{C}$, and denote it $\mathbb{F}[\mathcal{C}]$. We then show that $\mathbb{F}[\mathcal{C}]$ is isomorphic to any $n\mathbb{F}$-FDA whose structure tensor is $\mathcal{C}$ in some basis (Lemma B.2). Although any $\mathcal{C} \in \mathbb{F}^{n \times n \times n}$ defines an $n\mathbb{F}$-FDA, a key question is how the algebraic properties of $(\mathfrak{A}, \cdot)$ are reflected in $\mathcal{C}$. The following proposition makes this connection explicit by characterizing associativity, commutativity, and unitality in terms of tensor equations.

**Proposition 2.2.** *An $n\mathbb{F}$-FDA $(\mathfrak{A}, \cdot)$ with structure tensor $\mathcal{C} \in \mathbb{F}^{n \times n \times n}$ in the basis $B = \{\mathbf{a}^{(i)}\}_{i \in [n]}$ is*

*(i)* **associative** *if and only if $\sum_{k=1}^n \mathcal{C}_{ijk}\mathcal{C}_{klm} = \sum_{k=1}^n \mathcal{C}_{ikm}\mathcal{C}_{jlk} \ \forall i, j, l, m \in [n]$;*

*(ii)* **commutative** *if and only if $\mathcal{C}$ is symmetric in its first two modes, i.e. $\mathcal{C}_{ijk} = \mathcal{C}_{jik} \ \forall i, j, k \in [n]$;*

*(iii)* **unital** *if and only if there exists $\lambda \in \mathbb{F}^n$ such that $\sum_{i=1}^n \lambda_i \mathcal{C}_{i,:,:} = \sum_{i=1}^n \lambda_i \mathcal{C}_{:,i,:} = \mathbb{I}_n$. In that case, $\lambda$ is unique and $1_{\mathfrak{A}} = \sum_{i=1}^n \lambda_i \mathbf{a}^{(i)}$.*

*Proof.* See Proposition B.5. $\square$

### 2.3. Representations of FDA

For $m \in \mathbb{N}^*$, let $\mathcal{M}_m(\mathbb{F})$ denote the set of all $m \times m$ matrices over $\mathbb{F}$, which is an $m^2\mathbb{F}$-FDA under standard matrix multiplication. A representation of dimension $m$ of a $\mathbb{F}$-algebra $(\mathfrak{A}, \cdot)$ involves a homomorphism $\rho : \mathfrak{A} \to \mathcal{M}_m(\mathbb{F})$ such that $\rho(\alpha\mathbf{u} + \beta\mathbf{v}) = \alpha\rho(\mathbf{u}) + \beta\rho(\mathbf{v})$ and $\rho(\mathbf{u} \cdot \mathbf{v}) = \rho(\mathbf{u})\rho(\mathbf{v})$ for all $\mathbf{u}, \mathbf{v} \in \mathfrak{A}$ and $\alpha, \beta \in \mathbb{F}$. If $\mathfrak{A}$ is unital, it is further required that $\rho(1_{\mathfrak{A}}) = \mathbb{I}_m$. The following proposition characterizes when such a representation exists in terms of the structure tensor.

**Proposition 2.3.** *Let $(\mathfrak{A}, \cdot)$ be an $n\mathbb{F}$-FDA with structure tensor $\mathcal{C} \in \mathbb{F}^{n \times n \times n}$ in $B = \{\mathbf{a}^{(i)}\}_{i \in [n]}$. Fix $m \in \mathbb{N}^*$ and $\mathcal{R} \in \mathbb{F}^{n \times m \times m}$. Set $\mathcal{R}_k = \mathcal{R}_{k,:,:} \in \mathbb{F}^{m \times m}$ for all $k \in [n]$. The linear map $\rho : \mathfrak{A} \to \mathcal{M}_m(\mathbb{F})$ defined by*

$\rho\left(\sum_{k=1}^n \alpha_k \mathbf{a}^{(k)}\right) := \sum_{k=1}^n \alpha_k \mathcal{R}_k \ \forall \alpha \in \mathbb{F}^n$ *is a representation of $\mathfrak{A}$ if and only if*

$$\mathcal{R}_i \mathcal{R}_j = \sum_k \mathcal{C}_{ijk} \mathcal{R}_k \quad \forall i, j \in [n] \tag{3}$$

*and, if $\mathfrak{A}$ is unital with $1_{\mathfrak{A}} = \sum_i \lambda_i \mathbf{a}^{(i)}$, $\sum_i \lambda_i \mathcal{R}_i = \mathbb{I}_m$.*

*Proof.* See Proposition B.16. $\square$

*Example* 2.2. With the algebra of complex numbers in $\mathbb{F} = \mathbb{R}$, the defining relations for $(\mathbf{a}^{(1)}, \mathbf{a}^{(2)}) = (1, \mathbf{i})$ are $\mathcal{R}_1^2 = \mathcal{R}_1$, $\mathcal{R}_1\mathcal{R}_2 = \mathcal{R}_2\mathcal{R}_1 = \mathcal{R}_2$, $\mathcal{R}_2^2 = -\mathcal{R}_1$; together with $\mathcal{R}_1 = \mathbb{I}_m$ since $1_{\mathfrak{A}} = \mathbf{a}^{(1)}$. Hence $\mathcal{R}_2^2 = -\mathbb{I}_m$, which forces $m$ to be even. Writing $m = 2k$, the general solution is $\mathcal{R}_2 = \mathbf{S}(\mathbb{I}_k \otimes \mathcal{C}_2^\top)\mathbf{S}^{-1} \ \forall \mathbf{S} \in \mathrm{GL}_m(\mathbb{R})$, with $\mathrm{GL}_m(\mathbb{R})$ the general linear group, and $\mathcal{C}_2^\top = \left[\begin{smallmatrix} 0 & -1 \end{smallmatrix}\right], \left[\begin{smallmatrix} 1 & 0 \end{smallmatrix}\right]$ (the standard real representation of multiplication by $\mathbf{i}$). In $\mathbb{F}_p$, $\mathcal{R}_2^2 = -\mathbb{I}_m$ becomes $\mathcal{R}_2^2 = (p-1)\mathbb{I}_m$. There is no parity restriction on $m$, and a convenient normal form is $\mathcal{R}_2 = (p-1)^{1/2}\mathbf{S}\,\mathrm{diag}\,(\epsilon)\mathbf{S}^{-1} \ \forall \mathbf{S} \in \mathrm{GL}_m(\mathbb{R}), \epsilon \in \{\pm 1\}^m$.

It is worth asking, for a given $\mathcal{C} \in \mathbb{F}^{n \times n \times n}$, what is the smallest $m$ (or the values of $m$) for which the system of equations (3) admits a non-trivial solution. This question amounts to determining the minimal dimension of a faithful representation of $\mathfrak{A}$, which is a classical but non-trivial problem in representation theory. For associatives $n\mathbb{F}$-FDA, this smallest $m$ is less than or equal to $n$ (Proposition B.18). Since our goal in Section 5 is to investigate whether neural networks trained on data from $\mathfrak{A} = \mathbb{F}[\mathcal{C}]$ implicitly learn such representations, we do not attempt to solve the minimality problem here. Instead, we will assume a sufficiently large $m$ so that the system can be solvable and concentrate on analyzing how the learned embedding dimension relates to the properties of $\mathfrak{A}$ and its structure tensor.

## 3. From Groups to FDA

We show in this section that group learning is a special case of FDA learning. In fact, the study of grokking in groups naturally extends to algebras through their structure tensors. To see this, consider a finite group $(G, \circ)$ with $n$ elements $\{g_i\}_{i \in [n]}$. One can construct the Cayley table of the group as a matrix $\mathbf{A} \in G^{n \times n}$, where each entry is given by $\mathbf{A}_{ij} = g_i \circ g_j, \ \forall i, j \in [n]$. This table contains $n^2$ operations, which are then randomly partitioned into two disjoint non-empty subsets, a training set and a test set (Power et al., 2022). More precisely, the model is trained to predict $\Psi(g_i \circ g_j)$ given $\Psi(g_i)$ and $\Psi(g_j)$, with $\Psi : G \to \{0, 1\}^n$ the one-hot encoding defined on $G$, $\Psi(g_i) := [\mathbb{1}(g_k = g_i)]_{k \in [n]} \in \{0, 1\}^n \ \forall i \in [n]$, $\mathbb{1}$ the indicator function.

Now, for a field $\mathbb{F}$, define the set $\mathbb{F}[G] := \left\{\sum_{i=1}^n \alpha_i \mathbf{a}^{(i)} \mid \alpha \in \mathbb{F}^n\right\}$, where $\mathbf{a}^{(i)} = \Psi(g_i) \in \{0_{\mathbb{F}}, 1_{\mathbb{F}}\}^n \ \forall i \in [n]$. For all $\mathbf{u} = \sum_i \mathbf{u}_i \mathbf{a}^{(i)}$ and $\mathbf{v} = \sum_j \mathbf{v}_j \mathbf{a}^{(j)}$ in $\mathbb{F}[G]$, let $\mathbf{u} \cdot \mathbf{v} := \sum_{i,j} \mathbf{u}_i \mathbf{v}_j \Psi(g_i \circ g_j) =$

$\sum_k \left( \sum_{i,j} \mathbf{u}_i \mathbf{v}_j \mathbb{1}(g_i \circ g_j = g_k) \right) \mathbf{a}^{(k)}$. The following proposition shows that $(\mathbb{F}[G], \cdot)$ is an $n\mathbb{F}$-FDA, and that training a model on the group $(G, \circ)$ is equivalent to training it on the multiplication in $(\mathbb{F}[G], \cdot)$. In fact, applying the encoding $\Psi$ to each entry of the Cayley table $\mathbf{A}$ of $G$ directly yields the structure tensor $\mathcal{C}$ of $(\mathbb{F}[G], \cdot)$ in $B = \{\mathbf{a}^{(i)}\}_{i \in [n]}$. Therefore, any framework developed for algebras already subsumes the case of groups. In particular, by investigating grokking in algebras, we not only extend the scope of previous studies but also recover them as a special case.

**Proposition 3.1.** *For a finite group $(G, \circ)$ with $n$ elements $\{g_i\}_{i \in [n]}$ and identity $e$, $(\mathbb{F}[G], \cdot)$ is an $n$-dimensional associative and unital $\mathbb{F}$-FDA with $1_{\mathbb{F}[G]} = \Psi(e)$. Also, $(\mathbb{F}[G], \cdot)$ is commutative if and only if $G$ is commutative. Moreover, the structure tensor of $\mathbb{F}[G]$ in $B = \{\Psi(g_i)\}_{i \in [n]}$ is given by $\mathcal{C}_{ijk} = \mathbb{1}_{\mathbb{F}}(g_i \circ g_j = g_k) \forall i, j, k \in [n]$; and we have $\Psi(g_i \circ g_j) = \mathcal{C} \times_1 \Psi(g_i) \times_2 \Psi(g_j) \forall i, j \in [n]$.*

*Proof.* See Proposition B.10. □

# 4. Learning Finite Dimensional Algebra

In this section, we study supervised learning of multiplication in a finite-dimensional algebra (FDA). We show in the previous section that any $n$-dimensional FDA $\mathfrak{A}$ over a field $\mathbb{F}$ with structure tensor $\mathcal{C}^* \in \mathbb{F}^{n \times n \times n}$ is isomorphic to $\mathbb{F}^n$ endowed with the bilinear product $(\mathbf{u}, \mathbf{v}) \mapsto \mathcal{C}^* \times_1 \mathbf{u} \times_2 \mathbf{v} \in \mathbb{F}^n$ (Proposition 2.1). Given inputs $\mathbf{x} = (\mathbf{u}, \mathbf{v}) \in \mathbb{F}^n \times \mathbb{F}^n$, the supervised target is $\mathbf{y}^*(\mathbf{x}) = \mathcal{C}^* \times_1 \mathbf{u} \times_2 \mathbf{v}$. The structure of $\mathcal{C}^*$ (e.g., rank, sparsity pattern) is inherited from $\mathfrak{A}$ and can shape both sample complexity and the dynamics of generalization. **We propose to investigate how these properties impact generalization and the underlying mechanism by which they do so**.

## 4.1. Grokking Regimes in FDAs

Liu et al. (2023) argue that grokking arises when task performance hinges on learning a useful representation. For algorithmic data (Power et al., 2022), representation quality typically determines all-or-nothing accuracy (random guess vs. 100%), while on natural data (e.g., MNIST) representation quality may separate 95% from 100% accuracy. In our setting, this viewpoint yields a concrete mechanism: for a finite-field FDA ($\mathbb{F} = \mathbb{F}_p = \mathbb{Z}/p\mathbb{Z}$, $p$ prime), treat $\mathfrak{A}$ as a vocabulary, index each element $\mathbf{u} \in \mathfrak{A}$ by $\langle \mathbf{u} \rangle \in [q]$ with $q = |\mathfrak{A}|$, and learn an embedding matrix $\mathbf{E} \in \mathbb{R}^{q \times d}$ so that $\mathbf{E}_{\langle \mathbf{u} \rangle}$ is the trainable vector attached to $\mathbf{u}$. Grokking corresponds to the point at which $\mathbf{E}$ and the downstream layers collectively represent the algebra, so a simple linear readout recovers the correct output token.

By contrast, for $\mathbb{F} = \mathbb{R}$, a model that operates directly on $(\mathbf{u}, \mathbf{v}) \in \mathbb{F}^n \times \mathbb{F}^n$ and parameterizes a (bi)linear map can often fit without a prolonged representation-formation phase; grokking (in the strict "delayed generalization after memorization" sense) typically requires being induced via, e.g., large-scale initialization with small weight decay (Liu et al., 2023; Lyu et al., 2024) or by strongly misaligning the Neural Tangent Kernel with the target (Kumar et al., 2024). Moreover, Levi et al. (2024) and Notsawo et al. (2025) document cases of *"grokking without understanding"*: a sharp decrease in the test error during training, driven by changes in the $\ell_2$-norm of the model parameters, but that does not result in convergence to an optimal solution.

From this analysis, we distinguish $\mathbb{R}$ from $\mathbb{F}_p$. The present paper focuses on the finite-field setting, where representation learning is intrinsic and grokking phenomena are most salient (Figure 1). We include a concise treatment of the real case below, where learning reduces to a linear inverse problem where explicit or implicit low-rank bias governs recovery, and defer rank/coherence details to the supplement.

## 4.2. A Linear Inverse View for $\mathbb{F} = \mathbb{R}$

Let $\mathbf{U}, \mathbf{V}, \mathbf{Y} \in \mathbb{F}^{N \times n}$ be such that for each $s \in [N]$, $\mathbf{Y}_s = \mathcal{C}^* \times_1 \mathbf{U}_s \times_2 \mathbf{V}_s = \mathcal{C}^*_{(3)}(\mathbf{V}_s \otimes \mathbf{U}_s)$. Stacking gives $\mathbf{Y} = \mathbf{X}\mathcal{C}^{*\top}_{(3)}$ with design $\mathbf{X} := \mathbf{V} \bullet \mathbf{U} \in \mathbb{R}^{N \times n^2}$, where $\bullet$ is the face-splitting product ($\mathbf{X}_s = \mathbf{V}_s \otimes \mathbf{U}_s$). We recall that $\mathbf{Y}_s$ is the $s$th row of $\mathbf{Y}$ treated as a column vector, same thing for $\mathbf{U}_s$ and $\mathbf{V}_s$. Finding another structure tensor $\mathcal{C} \in \mathbb{F}^{n \times n \times n}$ such that $\mathbf{Y}_s = \mathcal{C} \times_1 \mathbf{U}_s \times_2 \mathbf{V}_s \forall s \in [N]$ is equivalent to solving the following linear equation in $\mathcal{C}$;

$$\mathbf{X}\mathcal{C}^{\top}_{(3)} = \mathbf{Y} = \mathbf{X}\mathcal{C}^{*\top}_{(3)} \iff \mathbf{M}\mathbf{b} = \text{vec}(\mathbf{Y}) = \mathbf{M}\mathbf{b}^* \quad (4)$$

with $\mathbf{M} := \mathbb{I}_n \otimes \mathbf{X} \in \mathbb{F}^{Nn \times n^3}$ and $\mathbf{b} := \text{vec}(\mathcal{C}^{\top}_{(3)}) \in \mathbb{F}^{n^3}$, similarly for $\mathbf{b}^*$. This problem can be viewed as a matrix factorization problem with matrix $\mathcal{C}^{*\top}_{(3)} \in \mathbb{F}^{n^2 \times n}$ and measurement $(\mathbf{U}, \mathbf{V})$, or a compressed sensing problem with signal $\mathbf{b}^*$ and measurement $\mathbf{M}$. Thus, learning $\mathcal{C}^*$ from $(\mathbf{U}, \mathbf{V}, \mathbf{Y})$ is a linear inverse problem whose identifiability and sample complexity are governed by the conditioning of $\mathbf{X} = \mathbf{V} \bullet \mathbf{U}$ vis-à-vis the signal $\mathcal{C}^*_{(3)}$, and the effective low-rank structure of $\mathcal{C}^*_{(3)}$. A convenient parameterization is a linear network $\mathcal{C}^{\top}_{(3)} = \mathbf{W}^{(L)} \cdots \mathbf{W}^{(1)}$ with $\mathbf{W}^{(L)} \in \mathbb{R}^{n^2 \times d}$, $\mathbf{W}^{(i)} \in \mathbb{R}^{d \times d}$ $(1 < i < L)$, and $\mathbf{W}^{(1)} \in \mathbb{R}^{d \times n}$. For $L = 1$, explicit low-rank regularization (e.g., nuclear norm) or appropriate sparsity penalties may be necessary when $N$ is sufficiently large (Candès & Tao, 2010; Notsawo et al., 2025). For $L \geq 2$, gradient descent exhibits an implicit bias toward low-rank solutions, enabling accurate recovery without explicit regularization in many regimes (Gunasekar et al., 2017; Arora et al., 2018; 2019; Gidel et al., 2019; Gissin et al., 2019; Razin & Cohen, 2020; Li et al., 2020). Beyond rank, recovery depends on *coherence* of the left/right singular subspaces of $\mathcal{C}^{*\top}_{(3)}$ with the measurement matrix $\mathbf{X}$ (Candès & Tao, 2010; Candes

& Recht, 2012; Chen et al., 2014). We summarize these implications for Equation (4) in Section C.1.

While Equation (4) makes it straightforward to study recovery as a function of properties of $\mathcal{C}^*$, the same is not true for the induced algebra $\mathbb{R}[\mathcal{C}^*] = (\mathbb{R}^n, \times)$. Associativity, commutativity, and unitality correspond to polynomial constraints of measure zero in $\mathbb{R}^{n^3}$ (Proposition 2.2). Thus, a randomly sampled $\mathcal{C}^*$ almost surely defines a degenerate non-associative, non-commutative, non-unital product. This makes principled empirical study over $\mathbb{R}$ difficult[2] and motivates our focus on finite fields, where structured algebras occur with non-negligible probability.

### 4.3. Finite Fields $\mathbb{F} = \mathbb{Z}/p\mathbb{Z}$

Over $\mathbb{F} = \mathbb{F}_p$, algebraic properties have positive density. For moderate $(n, p)$, the space of possibilities ($p^{n^3}$ tensors) is finite and can be systematically explored and stratified by properties such as associativity, commutativity, and unitality. Our experimental program is thus: (i) vary algebraic properties of $\mathbb{F}_p[\mathcal{C}^*]$ and measure their effect on grokking delay and generalization; (ii) study how structural features of $\mathcal{C}^*$ shape learning dynamics; and (iii) probe whether models learn algebraic representations during training.

## 5. Experiments and Results

### 5.1. Experiment Setup

From now on, we work over the finite field $\mathbb{F} = \mathbb{F}_p$ (prime $p$). Identify $\mathfrak{A} \equiv \mathbb{F}^n$ and set $q := |\mathfrak{A}| = p^n$. Each element $\mathbf{u} \in \mathfrak{A}$ is treated as a vocabulary symbol with index $\langle \mathbf{u} \rangle \in [q]$. The models we use will associate to each of these symbols a trainable vector $\mathbf{E}_{\langle \mathbf{u} \rangle} \in \mathbb{R}^d$, with $\mathbf{E} \in \mathbb{R}^{V \times d}$, $V = q + 2$ the vocabulary size, 2 for the special tokens $\mathcal{S} = \{\times, =\}$.

We train the model using a classification approach. The logits for $\mathbf{x} = (\mathbf{u}, \times, \mathbf{v}, =)$ with $(\mathbf{u}, \mathbf{v}) \in \mathfrak{A}^2$ are given by $y_\theta(\mathbf{x}) = \varphi\left(\phi\left(\mathbf{E}_{\langle \mathbf{u} \rangle} \oplus \mathbf{E}_{\langle \times \rangle} \oplus \mathbf{E}_{\langle \mathbf{v} \rangle} \oplus \mathbf{E}_{\langle = \rangle}\right)\right) \in \mathbb{R}^q$, where $\oplus$ is the vector concatenation, $\phi$ the encoder and $\varphi$ the classifier[3]. The dataset $\mathcal{D} = \{((\mathbf{u}, \times, \mathbf{v}, =), \mathbf{y}^*(\mathbf{u}, \mathbf{v}) \mid (\mathbf{u}, \mathbf{v}) \in \mathfrak{A}^2\}$ is randomly partitioned into two disjoint and non-empty sets $\mathcal{D}_{\text{train}}$ and $\mathcal{D}_{\text{test}}$, the training and the validation dataset respectively, following a ratio $r := |\mathcal{D}_{\text{train}}|/|\mathcal{D}| \in (0, 1]$, which allows us to interpolate between different data-availability regimes. The models

---

[2]This limitation is a property of the algebraic landscape over $\mathbb{R}$, not our tensorial framework, which simply makes it explicit. Structured (e.g., unital) $n\mathbb{R}$-FDA form a measure-zero subset among the set of all possible $n\mathbb{R}$-FDA. Thus, random sampling yields almost always degenerate algebras.

[3]The tokens $\times$ and $=$ are not strictly necessary. Training the classifier on the simpler input pair $(\mathbf{u}, \mathbf{v})$ yields qualitatively identical conclusions. We included for consistency with the formulation in (Power et al., 2022), but the results do not rely on them.

are trained to minimize the average cross-entropy loss $\mathcal{L}_{\text{train}}(\theta) = \sum_{(\mathbf{x}, \mathbf{y}^*) \in \mathcal{D}_{\text{train}}} \ell(y_\theta(\mathbf{x}), \langle \mathbf{y}^* \rangle)$, with $\ell(\mathbf{y}, i) = -\mathbf{y}_i + \log\left(\sum_j \exp(\mathbf{y}_j)\right) \forall \mathbf{y} \in \mathbb{R}^q, i \in [q]$. We denoted by $\mathcal{A}_{\text{train}}$ the corresponding accuracy ($\mathcal{L}_{\text{test}}$ and $\mathcal{A}_{\text{test}}$ on $\mathcal{D}_{\text{test}}$).

We use a linear classifier $\varphi(\mathbf{z}) = \mathbf{b} + \mathbf{Wz} \in \mathbb{R}^q$. The learnable parameters $\theta$ consist of $\mathbf{E}, \mathbf{W}, \mathbf{b}$ together with those of the encoder $\phi$. We consider three architectures for $\phi$: an MLP, an LSTM, and a Transformer (full details are provided in Appendix D). For clarity, we present mainly the MLP results and some Transformer results in the following section. Additional results, along with experimental details, are deferred to Appendix E.

### 5.2. Representation Learning

Over a finite alphabet, the model must construct a representation in which all $q^2$ algebra products are linearly separable by the classifier head, $q := |\mathfrak{A}|$. Early training can memorize subsets of products via brittle lookup-like features; only after the embedding geometry aligns with the algebraic structure does linear decoding succeed on the full combinatorial support, producing the characteristic delayed generalization of grokking (Liu et al., 2023). To see this, assume that $(\mathfrak{A}, \times)$ has a group structure and let $\rho : \mathfrak{A} \to \mathbb{R}^{m \times m}$ be a faithful representation of $\mathfrak{A}$. If the model $\mathbf{x} \to \mathbf{W}\phi(\mathbf{x})$ encodes a pair $\mathbf{x} = (\mathbf{u}, \mathbf{v})$ as $\phi(\mathbf{x}) = \rho(\mathbf{u} \times \mathbf{v})$ and the linear classifier head stores the inverses $\rho(\mathbf{w})^{-1} \forall \mathbf{w} \in \mathfrak{A}$ in its weights, then the decoding rule recovers the correct product $\mathbf{u} \times \mathbf{v}$. In other words, under these assumptions, the model is able to generalize to all inputs.

**Proposition 5.1.** *Assume that $(\mathfrak{A}, \times)$ has a group structure, and let $\rho : \mathfrak{A} \to \mathbb{R}^{m \times m}$ be a faithful matrix representation of $\mathfrak{A}$. Suppose a model $y_\theta(\mathbf{x}) = \mathbf{W}\phi(\mathbf{x}) \in \mathbb{R}^q$ encodes pairs $\mathbf{x} = (\mathbf{u}, \mathbf{v}) \in \mathfrak{A}^2$ as $\phi(\mathbf{u}, \mathbf{v}) = \rho(\mathbf{u} \times \mathbf{v})$ and decodes with a linear classifier parameterized by weights $\mathbf{W} \in \mathbb{R}^{q \times m^2}$ containing $\rho(\mathbf{w})^{-1}$ for each $\mathbf{w} \in \mathfrak{A}$. Then the predicted label satisfies $\arg\max_{i \in [q]} y_\theta(\mathbf{x})[i] = \mathbf{u} \times \mathbf{v}$. As a consequence, the classifier linearly separates all $q$ outputs.*

*Proof.* See Proposition D.1. □

Thus, the classifier achieves exact decoding of the group product for all pairs $(\mathbf{u}, \mathbf{v}) \in \mathfrak{A}^2$ once the embedding geometry aligns with the group structure. Unfortunately, algebras do not have a group structure. However, in Proposition 2.3 we established a necessary and sufficient condition for verifying that a set of matrices represents an algebra, namely $\mathcal{R}_i \mathcal{R}_j = \sum_k \mathcal{C}_{ijk}^* \mathcal{R}_k \forall i, j \in [n]$. By setting $\mathbf{r}^{(i)} = \text{vec}(\mathcal{R}_i) \in \mathbb{F}^{m^2} \forall i \in [n]$ and defining $\mathbf{r}^{(i)} \cdot \mathbf{r}^{(j)} := \text{vec}(\mathcal{R}_i \mathcal{R}_j) = \sum_k \mathcal{C}_{ijk}^* \mathbf{r}^{(k)} \forall i, j \in [n]$ in $\mathfrak{B} := \left\{\sum_{i=1}^n \alpha_i \mathbf{r}^{(i)} \mid \alpha \in \mathbb{F}^n\right\}$, we obtain $\mathbf{u} \cdot \mathbf{v} =_{\mathfrak{B}} \mathcal{C}^* \times_1 \mathbf{u} \times_2 \mathbf{v} \forall \mathbf{u}, \mathbf{v} \in \mathfrak{B}$, where $=_{\mathfrak{B}}$ is the equality when

expressed in basis. We performed linear probing to check whether such a structure emerges in the features learned by the models.

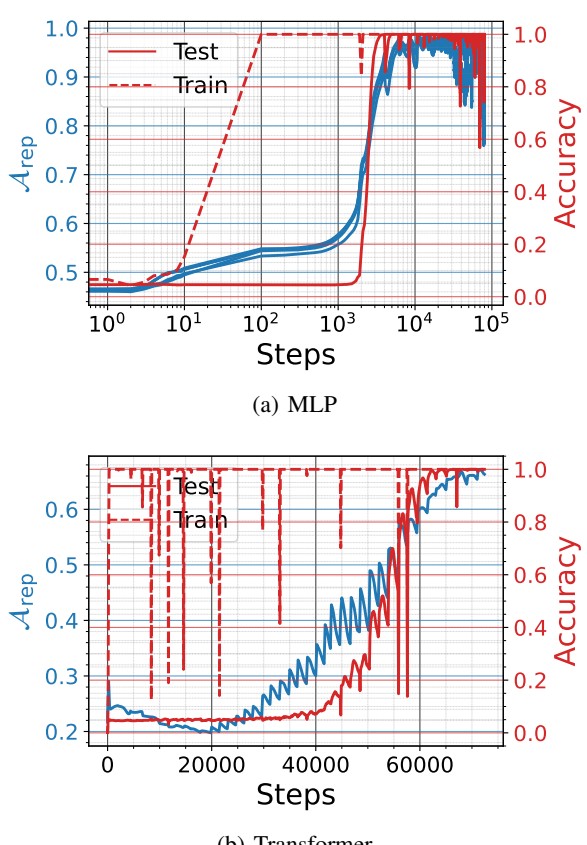

(a) MLP

(b) Transformer

*Figure 2.* Representation quality and generalization performances as a function of training steps. Before grokking $\mathcal{A}_{\text{rep}}$ remains relatively low and then sharply increases as the model groks.

More precisely, let $\mathbf{f}^{(\ell)}(\mathbf{u}) \in \mathbb{R}^{m^2}$ and $\mathbf{f}^{(r)}(\mathbf{u}) \in \mathbb{R}^{m^2}$ denote the feature vectors produced by the model for $\mathbf{u} \in \mathfrak{A}$ when $\mathbf{u}$ appears on the left and on the right of an equation (i.e. when appearing as the left/right operand), respectively; and let $\mathbf{f}(\mathbf{u}) \in \mathbb{R}^{m^2}$ denote the feature vector of $\mathbf{u}$ produced by the unembedding layer (the classifier). We want to determine whether there exists $\mathcal{W} \in \mathbb{R}^{m^2 \times m^2 \times m^2}$ such that $\mathbf{f}(\mathbf{u} \times \mathbf{v}) = \hat{\mathbf{f}}_{\mathcal{W}}(\mathbf{u}, \mathbf{v})$ for all $\mathbf{u}, \mathbf{v} \in \mathfrak{A}$, where $\hat{\mathbf{f}}_{\mathcal{W}}(\mathbf{u}, \mathbf{v}) := \mathcal{W} \times_1 \mathbf{f}^{(\ell)}(\mathbf{u}) \times_2 \mathbf{f}^{(r)}(\mathbf{v})$ is a bilinear model parameterized by $\mathcal{W}$. We use the cosine similarity $\mathcal{A}_{\text{rep}} = \frac{1}{|\mathfrak{A}^2|} \sum_{(\mathbf{u},\mathbf{v})} \langle \mathbf{f}(\mathbf{u} \times \mathbf{v}), \hat{\mathbf{f}}_{\mathcal{W}}(\mathbf{u}, \mathbf{v}) \rangle / \|\mathbf{f}(\mathbf{u} \times \mathbf{v})\| \, \|\hat{\mathbf{f}}_{\mathcal{W}}(\mathbf{u}, \mathbf{v})\|$ associated with solving this problem (i.e., finding $\mathcal{W}$) as a measure of how algebra-like structure emerges in the model's feature space. The tensor $\mathcal{W}$ is obtained by minimizing a squared reconstruction error $\sum_{(\mathbf{u},\mathbf{v})} \left\| \mathbf{f}(\mathbf{u} \times \mathbf{v}) - \hat{\mathbf{f}}_{\mathcal{W}}(\mathbf{u}, \mathbf{v}) \right\|_2^2$. This is solved using ordinary least squares with $\ell_2$ regularization (more details in Sections D.2.2 and E of the Appendix).

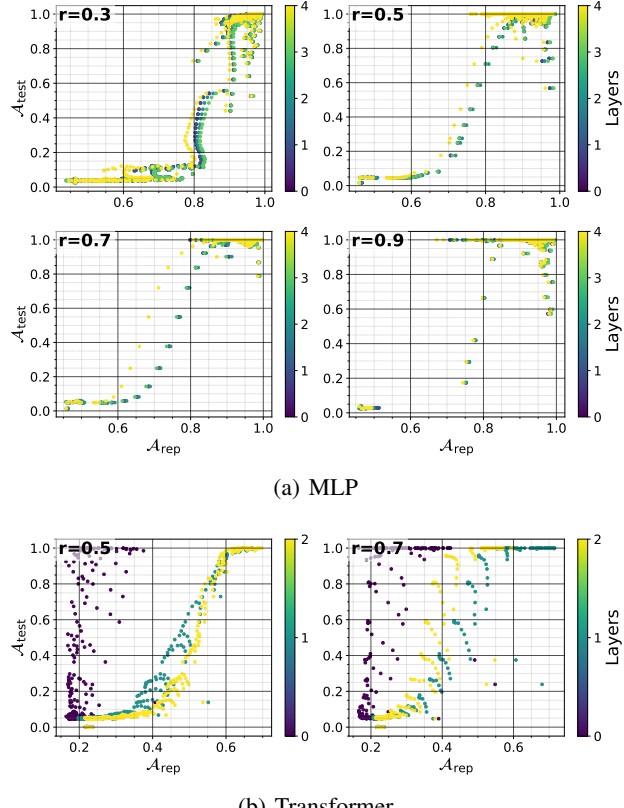

(a) MLP

(b) Transformer

*Figure 3.* Generalization accuracy as a function of representation quality for different training data size ($r$) and model layers (0 for first layer, 1 for the second, etc.) for a MLP and a Transformer Encoder: $\mathcal{A}_{\text{test}}$ increases with $\mathcal{A}_{\text{rep}}$ in a low-data regime.

As shown in Figure 2, before grokking $\mathcal{A}_{\text{rep}}$ remains relatively low and then sharply increases as the model groks. Figure 3 plots $\mathcal{A}_{\text{test}}$ as a function of $\mathcal{A}_{\text{rep}}$ across different model layers, training steps, and training dataset sizes. Each point is $(\mathcal{A}_{\text{rep}}^{(\ell)}, \mathcal{A}_{\text{test}})$ for a certain layer $\ell$. We observe that $\mathcal{A}_{\text{test}}$ increases with $\mathcal{A}_{\text{rep}}^{(\ell)}$ in a low-data regime, showing that the emergence of algebra-consistent features directly facilitates generalization when data are scarce. For Figures 2 and 3, the algebra being learned is the algebra of complex numbers over $\mathbb{F}_7$.

## 5.3. The Properties of the FDA on Generalization

We demonstrate in this section how algebraic structure (e.g., associativity) influences sample complexity, memorization time, and learning dynamics. For now on, we denote by $t_2$ (resp. $t_4$) the step at which the training (resp. test) accuracy first reaches $\approx 99\%$.

With the structure tensor $\mathcal{C}^*$ of complex numbers in $\mathbb{F} = \mathbb{Z}/7\mathbb{Z}$ (Example 2.1), we observed that the model exhibits grokking (Figure 1), but in a manner highly sensitive to the specific entries of $\mathcal{C}^*$. A single random perturbation of one entry in $\mathcal{C}^*$ can dramatically reduce/increase the

grokking delay (Figure 1). These observations suggest that the internal algebraic structure encoded by $\mathcal{C}^*$ exerts a direct influence on generalization.

Using $(n, p) = (2, 7)$, we generate $\mathcal{C}^*$ in $\mathbb{F}_p^{n \times n \times n}$. We have $p^{n^3} = 7^8 = 5764801$ possibilities for such $\mathcal{C}^*$. Let $a=associative$, $c=commutative$, $u=unital$. We have the following $2^3$ possible categories : $acu$, $ac\bar{u}$, $a\bar{c}u$, $\bar{a}cu$, $a\bar{c}\bar{u}$, $\bar{a}c\bar{u}$, $\bar{a}\bar{c}u$ and $\bar{a}\bar{c}\bar{u}$. The bar on top of the $symbols$ represents negation, and the concatenation is the Boolean conjunction. For example, $ac\bar{u}$ represents the tensor algebras that are associative and commutative but non-unital. We obtain 996 (0.017%) $acu$, 1741(0.03%) $ac\bar{u}$, 0 (0.0%) $a\bar{c}u$, 0 (0.0%) $\bar{a}cu$, 96 (0.002%) $a\bar{c}\bar{u}$, 114912 (1.99%) $\bar{a}c\bar{u}$, 0 (0.0%) $\bar{a}\bar{c}u$ and 5647056 (97.958%) $\bar{a}\bar{c}\bar{u}$. We sampled 10 tensors in each case (category). We varied $r = |\mathcal{D}_{\text{train}}|/|\mathcal{D}|$ in $\{0.2, 0.3, \ldots, 0.9\}$. For each triplet (category, $\mathcal{C}^*$, $r$), we repeated the experiment twice.

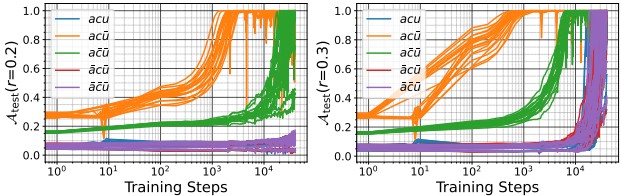

*Figure 4.* Evolution of the test accuracy $\mathcal{A}_{\text{test}}$ during training for different training data fraction $r \in \{0.2, 0.3\}$.

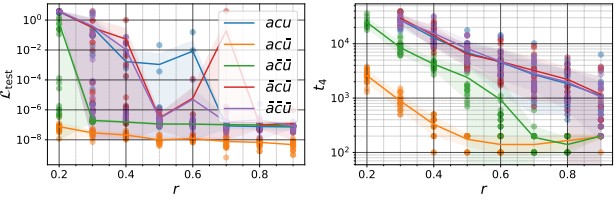

*Figure 5.* Test loss $\mathcal{L}_{\text{test}}$ and grokking step $t_4$ as a function of training data fraction $r$.

Our results are summarized in Figures 4 (for the other values of $r$, see Figure 14) and 5. We varied $r$ to show that algebra-dependent effects persist across regimes. Small $r$ leads to memorization; intermediate $r$ yields delayed generalization; and this delay decreases as $r \to 1$. This matches known behavior on algorithmic tasks (Power et al., 2022; Notsawo Jr et al., 2023).

We can observe that $ac\bar{u}$ algebras are consistently the easiest to learn (in terms of shorter grokking time and stronger generalization after grokking, measured in terms of test loss), followed by $a\bar{c}\bar{u}$, with the remaining categories showing almost indistinguishable behavior. This shows that the combination of non-unitality and associativity most strongly facilitates generalization. This finding is consistent with the previous discussion on representation learning. When the algebra is associative but non-unital ($a\bar{c}\bar{u}$ and $ac\bar{u}$), it

admits two trivial representations[4], the zero representation and left regular representation given by the tensor $\mathcal{C}^*$ itself, $\mathcal{R}_i = \mathcal{C}_i^{* \top} \forall i \in [n]$ (Proposition B.18). Moreover, the representations constraints admit many solutions: if $(\mathcal{R}_i)_{i \in [n]}$ is a representation of dimension $m$, then for any $p \geq m$ and $\mathbf{A} \in \mathbb{F}^{p \times m}$, $\mathbf{B} \in \mathbb{F}^{m \times p}$ with $\mathbf{BA} = \mathbb{I}_m$, $(\mathbf{A}\mathcal{R}_i\mathbf{B})_{i \in [n]}$ is also a representation (Proposition B.19). These provide simpler and more accessible solutions for the learning process, effectively lowering the complexity of the optimization landscape. In other words, removing the unit constraint allows the model to exploit "shortcuts" in the representation space, which explains why non-unital associative algebras generalize more quickly and grok earlier.

When the algebra is unital, the additional constraint $\sum_i \lambda_i \mathcal{R}_i = \mathbb{I}_m$ must be satisfied by its representations (Proposition 2.3). This requirement adds equations without adding degrees of freedom, drastically shrinking the feasible set and eliminating many representations, and this typically delays convergence. In fact, it significantly restricts the solution space, since it forces the representation to encode an identity element across all transformations, thereby reducing the flexibility of optimization and leading to slower convergence. In representation theory, such unit constraints are well known to eliminate many otherwise valid low-dimensional representations, leaving only more rigid (and more complex to approximate) ones.

Among the two non-unital regimes ($ac\bar{u}$ and $a\bar{c}\bar{u}$), commutativity makes $ac\bar{u}$ even easier from representation learning perspective because it reduces the number of equations in (3) from $n^2$ to $n(n + 1)/2$, because $\mathcal{R}_i\mathcal{R}_j = \sum_k \mathcal{C}_{ijk}\mathcal{R}_k$ and $\mathcal{R}_j\mathcal{R}_i = \sum_k \mathcal{C}_{ijk}\mathcal{R}_k$ represent the same equation under commutativity, for all $i, j$. Another way to see this is to observe that the models have the form $y(\mathbf{u}, \mathbf{v}) = \varphi(\phi(\mathbf{u}, \mathbf{v}))$, so that when $\phi$ learns a representation $\rho$ of the FDA, i.e. $\phi(\mathbf{u}, \mathbf{v}) = \rho(\mathbf{u} \times \mathbf{v}) \forall (\mathbf{u}, \mathbf{v})$, then whenever $\mathbf{u} \times \mathbf{v} = \mathbf{u}' \times \mathbf{v}'$ we have identical predictions $y(\mathbf{u}, \mathbf{v}) = y(\mathbf{u}', \mathbf{v}')$. In a commutative FDA, $\mathbf{u} \times \mathbf{v} = \mathbf{v} \times \mathbf{u}$, so each product has two presentations $(\mathbf{u}, \mathbf{v})$ and $(\mathbf{v}, \mathbf{u})$. Once the classifier predicts one ordering correctly, it also predicts the swapped one, effectively reducing the number of distinct outputs it must separate. This symmetry makes $ac\bar{u}$ easier than $a\bar{c}\bar{u}$ and yields earlier transitions.

### 5.4. The Properties of the Structure Tensor on Generalization

We focused in this section on the direct properties of $\mathcal{C}^*$ that influence generalization. Indeed, while the algebraic constraints on the entries of $\mathcal{C}^*$ required to ensure properties

---

[4]The probing from Section 5.2 on representation learning does not target any specific example of representation. It tests whether the model learns a representation that satisfies the constraints similar to those of Proposition 2.3, up to a linear transformation.

such as associativity are well understood, there are other structural characteristics of $\mathcal{C}^*$ whose connection to the underlying algebra remains unclear, especially over finite fields. Here, we examined two such properties: the sparsity $s$ of $\mathcal{C}^*$ and the rank $r^{(3)}$ of the mode-3 unfolding $\mathcal{C}^*_{(3)}$, motivated by our earlier discussion in the real-valued setting.

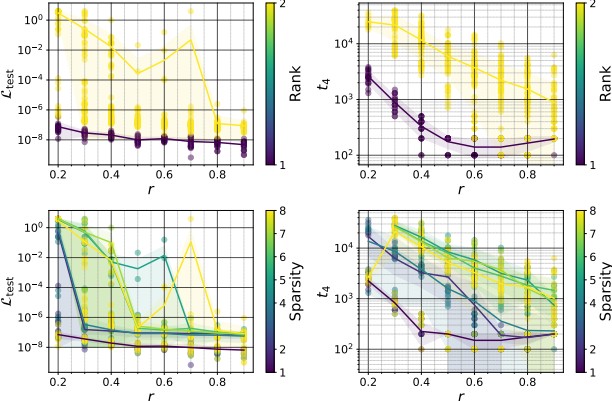

*Figure 6.* Test loss and grokking step as a function of training data fraction $r$ for different $r^{(3)} = \mathrm{rank}(\mathcal{C}^*_{(3)})$ and sparsity level $s$.

Our experiments show that both the test loss and the time to generalize increase with $r^{(3)}$ and $s$ (Figure 6), indicating that higher complexity in these dimensions hinders efficient generalization. This is also true for other mode-unfolding (Figure 13). Note that the rank is basis-invariant (Corollary B.4) and reflects the intrinsic multilinear complexity of the operation, which we claim is why increasing rank leads to longer grokking delays. Sparsity, on the other hand, is not basis-invariant. It reflects a genuine "privileged basis" problem familiar in mechanistic interpretability (Olsson et al., 2022), as the neural network sees raw coordinates, and these coordinates implicitly privilege certain representations. Our experiments, therefore, treat sparsity as a task-dependent difficulty parameter, naturally associated with each algebraic example, in the canonical basis. Our empirical claim is therefore conditional on the canonical basis used.

## 6. Discussion and Limitations

In the case of the Transformer Encoder (Figure 2), $\mathcal{A}^{(\ell)}_{\mathrm{rep}}$ does not change with $\mathcal{A}_{\mathrm{test}}$ for layer $\ell = 0$, the embedding layer. At first glance, this result contradicts previous work on group operations (Power et al., 2022; Liu et al., 2022), for which an emergence of structure is observed in the model's embedding layer as it groks. We believe this difference is due to the embedding and unembedding layers being shared. More specifically, for layer 0, we seek $\mathcal{W}$ such that $\mathbf{f}(\mathbf{u} \times \mathbf{v}) = \mathcal{W} \times_1 \mathbf{f}(\mathbf{u}) \times_2 \mathbf{f}(\mathbf{v}) = \mathcal{W}_{(3)}\big(\mathbf{f}(\mathbf{v}) \otimes \mathbf{f}(\mathbf{u})\big)$ for all $\mathbf{u}, \mathbf{v} \in \mathfrak{A}$, where $f : \mathfrak{A} \to \mathbb{R}^{m^2}$ is the embedding layer. The fact that $\mathcal{A}^{(0)}_{\mathrm{rep}}$ remains almost constant simply shows that such a representation does not emerge in our case,

without ruling out the possibility that another structure might emerge, such as the emergence of low-rank embeddings. We leave further investigation for future works.

Also, restricting most experiments to $n = 2$ over $\mathbb{F}_7$ limits the scope of the empirical evaluation. The space of possible structure tensors grows rapidly, with $|\mathbb{F}_p^{n \times n \times n}| = p^{n^3}$. As a result, exploring this space becomes computationally challenging even for moderate values of $n$ and $p$. We therefore chose a reasonable $(n, p)$ to enable

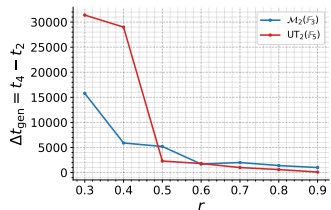

*Figure 7.* Generalization delay $\Delta t_{\mathrm{gen}}$ as a function of training data fraction $r \in (0, 1)$ for $\mathcal{M}_2(\mathbb{F}_3)$ and $\mathrm{UT}_2(\mathbb{F}_5)$, the algebra of $2 \times 2$ matrices ($n = 4$) over $\mathbb{F}_3$ and th $2 \times 2$ upper triangular matrices ($n = 3$) over $\mathbb{F}_5$, respectively.

a controlled and systematic study of how algebraic properties affect generalization and grokking behavior. Figure 7 shows additional experiments with a non-trivial generalization delay on two associative, noncommutative, and unital ($a\bar{c}u$) algebras: the matrix algebras (which arise when $n$ is a perfect square) and the upper triangular matrix algebra, which are not captured in the $n = 2$ setup. In fact, the minimum dimension $n$ to get an $a\bar{c}u$ algebra over any field $\mathbb{F}$ is 3 (Proposition B.7).

Several directions remain open. Theoretically, a tighter link between tensor properties (e.g., rank, coherence) and sample complexity is needed over finite fields. Empirically, scaling to larger algebras and probing real-world tasks with latent algebraic structure could further clarify when and why grokking occurs.

## 7. Conclusion

We introduced a tensorial framework for studying grokking in finite-dimensional algebras, showing that learning multiplication reduces to analyzing properties of structure tensors. This formulation recovers groups as a special case and extends grokking analysis to non-associative, non-commutative, and non-unital algebras. Our experiments demonstrate that algebraic constraints, such as unitality, do not always simplify learning and may even lengthen grokking delays, while structural properties, like sparsity or low rank, often accelerate generalization. We observe that test performance improves only once model layers align with the algebraic multiplication, reinforcing the view of grokking as a representational phase transition. These findings bridge classical algebra with modern learning theory: over real fields, FDA learning connects to low-rank matrix recovery, while over finite fields, grokking arises from the need to form internal representations.

# Acknowledgements

We are grateful to David Kanaa for helpful conversations in the early stages of this work, and to Jonas Ngnawé for his feedback on the first draft of this paper. Pascal Tikeng sincerely acknowledges the support from the Canada Excellence Research Chairs (CERC) program, without which this work would not have been possible. Guillaume Rabusseau acknowledges the support of the CIFAR AI Chair program. Guillaume Dumas was supported by the Institute for Data Valorization, Montreal and the Canada First Research Excellence Fund (IVADO; CF00137433), the Fonds de Recherche du Quebec (FRQ; 285289), the Natural Sciences and Engineering Research Council of Canada (NSERC; DGECR-2023-00089), and the Canadian Institute for Health Research (CIHR 192031; SCALE).

# Impact Statement

This paper aims to advance the field of Machine Learning by improving the understanding of grokking in neural networks. While the focus is on finite-dimensional algebra, our work has practical implications for optimizing model training. We acknowledge the potential ethical considerations of AI technologies, but do not feel any specific issues need to be highlighted here at this time.

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

## A. Notations and Useful Identities

We used the following standard notations.

- For $n \in \mathbb{N}^*$, $[n] := \{1, \ldots, n\}$. For $m, n \in \mathbb{N}$ with $m \geq n$, $[\![n, m]\!] := \{n, n+1, \cdots, m\}$.

- Scalars are denoted by lowercase letters, e.g., $a$; vectors are denoted by boldface lowercase letters, e.g., $\mathbf{a} = [\mathbf{a}_i]_i$; matrices are denoted by boldface capital letters, e.g., $\mathbf{A} = [\mathbf{A}_{i,j}]_{i,j}$; higher-order tensors (order three or higher) are denoted by boldface Euler script letters, e.g., $\boldsymbol{\mathcal{A}} = [\boldsymbol{\mathcal{A}}_{i_1, i_2, i_3, \ldots}]_{i_1, i_2, i_3, \ldots}$.

- The $i^{th}$ row (resp. column) of a matrix $\mathbf{A}$ will be denoted by $\mathbf{A}_{i,:}$ or simply $\mathbf{A}_i$ (resp. $\mathbf{A}_{:,i}$). This notation is extended to slices of a tensor straightforwardly. Commas in subscripts may occasionally be omitted whenever no ambiguity arises, e.g., $\mathbf{A} = [\mathbf{A}_{ij}]_{ij}$ and $\mathbf{A}_{i:}$. Unless otherwise stated, indices of vectors, matrices, and tensors will start at $1$. When 0-based indexing is used, it will be explicitly mentioned.

- For a given dimension $n \in \mathbb{N}^*$, $\mathbf{e}^{(k)} \in \mathbb{R}^n$ is the $k^{th}$ vector of the canonical basis of $\mathbb{R}^n$, $\mathbf{e}_l^{(k)} = \delta_{kl} \forall l \in [n]$. Here, $\delta$ is the Kronecker delta function.

- For a vector $\mathbf{x} \in \mathbb{R}^n$, $\|\mathbf{x}\|_0 = |\{i \in [n], \mathbf{x}_i \neq 0\}|$, $\|\mathbf{x}\|_p = \left(\sum_{i=1}^n |\mathbf{x}_i|^p\right)^{\frac{1}{p}} \forall p \in (0, \infty)$ and $\|\mathbf{x}\|_\infty = \max_{i \in [n]} |\mathbf{x}_i|$.

- For a matrix $\mathbf{A} \in \mathbb{R}^{m \times n}$, the operator $\text{vec}(\mathbf{A}) \in \mathbb{R}^{mn}$ stacks the column of $\mathbf{A}$ in a vector, i.e. $(\text{vec}(\mathbf{A}))_{(j-1)m+i} = \mathbf{A}_{ij}$ for all $(i, j) \in [m] \times [n]$. We have $\|\mathbf{A}\|_F := \|\text{vec}(\mathbf{A})\|_2$.

- For the operations on vectors and matrices, we denote by $\odot$ the Hadamard product, $\otimes$ the Kronecker product, $\star$ the Khatri-Rao product, $\bullet$ the face-splitting product, and $\circ$ the outer product. For $n$ vectors $\mathbf{a}^{(i)} \in \mathbb{R}^{m_i} \ \forall i \in [n]$, $\left(\mathbf{a}^{(1)} \circ \cdots \circ \mathbf{a}^{(n)}\right)_{i_1, \cdots, i_n} = \mathbf{a}_{i_1}^{(1)} \cdots \mathbf{a}_{i_n}^{(n)} \ \forall (i_1, \cdots, i_n) \in [m_1] \times \cdots \times [m_n]$.

Let $\boldsymbol{\mathcal{T}} \in \mathbb{R}^{d_1 \times d_2 \times d_3}$.

- For a vector $\mathbf{a} \in \mathbb{R}^{d_1}$, the matrix $\mathbf{A} = \boldsymbol{\mathcal{T}} \times_1 \mathbf{a} \in \mathbb{R}^{d_2 \times d_3}$ is such that $\mathbf{A}_{jk} = \sum_{i=1}^{d_1} \boldsymbol{\mathcal{T}}_{ijk} \mathbf{a}_i = \boldsymbol{\mathcal{T}}_{:,j,k}^\top \mathbf{a} \ \forall (j, k) \in [d_2] \times [d_3]$, that is $\mathbf{A} = \sum_{i=1}^{d_1} \mathbf{a}_i \boldsymbol{\mathcal{T}}_{i,:,:}$. The matrices $\boldsymbol{\mathcal{T}} \times_2 \mathbf{a} \in \mathbb{R}^{d_1 \times d_3}$ and $\boldsymbol{\mathcal{T}} \times_3 \mathbf{a} \in \mathbb{R}^{d_1 \times d_2}$ are define similarly, for $\mathbf{a} \in \mathbb{R}^{d_2}$ and $\mathbf{a} \in \mathbb{R}^{d_3}$ respectively: $\boldsymbol{\mathcal{T}} \times_2 \mathbf{a} = \sum_{j=1}^{d_2} \boldsymbol{\mathcal{T}}_{:,j,:} \mathbf{a}_j$ and $\boldsymbol{\mathcal{T}} \times_3 \mathbf{a} = \sum_{k=1}^{d_3} \boldsymbol{\mathcal{T}}_{:,:,k} \mathbf{a}_k$.

  So, for $\mathbf{a} \in \mathbb{R}^{d_1}$ and $\mathbf{b} \in \mathbb{R}^{d_2}$, the vector $\mathbf{c} = \boldsymbol{\mathcal{T}} \times_1 \mathbf{a} \times_2 \mathbf{b} = \mathbf{A} \times_2 \mathbf{b} = \mathbf{A}^\top \mathbf{b} \in \mathbb{R}^{d_3}$, with $\mathbf{A} = \boldsymbol{\mathcal{T}} \times_1 \mathbf{a} \in \mathbb{R}^{d_2 \times d_3}$, is such that $\mathbf{c}_k = \mathbf{a}^\top \boldsymbol{\mathcal{T}}_{:,:,k} \mathbf{b} = \mathbf{A}_{:,k}^\top \mathbf{b} \ \forall k \in [d_3]$, i.e. $\mathbf{c} = \sum_{i=1}^{d_1} \sum_{j=1}^{d_2} \mathbf{a}_i \mathbf{b}_j \boldsymbol{\mathcal{T}}_{i,j,:} = \sum_{j=1}^{d_2} \mathbf{b}_j \mathbf{A}_{j,:}$.

  As a special case, identifying matrices with tensors in $\mathbb{R}^{d_1 \times d_2 \times 1}$ gives

  $$\begin{cases} \mathbf{A} \times_1 \mathbf{a} = \mathbf{A}^\top \mathbf{a} \in \mathbb{R}^{d_2} \\ \mathbf{A} \times_2 \mathbf{b} = \mathbf{A}\mathbf{b} \in \mathbb{R}^{d_1} \end{cases} \quad \forall \mathbf{A} \in \mathbb{R}^{d_1 \times d_2}, \mathbf{a} \in \mathbb{R}^{d_1}, \mathbf{b} \in \mathbb{R}^{d_2} \tag{5}$$

- For a matrix $\mathbf{A} \in \mathbb{R}^{m \times d_1}$, the tensor $\boldsymbol{\mathcal{A}} = \boldsymbol{\mathcal{T}} \times_1 \mathbf{A} \in \mathbb{R}^{m \times d_2 \times d_3}$ is such that $\boldsymbol{\mathcal{A}}_{ijk} = \sum_{l=1}^{d_1} \mathbf{A}_{il} \boldsymbol{\mathcal{T}}_{ljk} \ \forall (i, j, k) \in [m] \times [d_2] \times [d_3]$; or equivalently, $\boldsymbol{\mathcal{A}}_{i,:,:} = \sum_{l=1}^{d_1} \mathbf{A}_{il} \boldsymbol{\mathcal{T}}_{l,:,:} \in \mathbb{R}^{d_2 \times d_3} \ \forall i \in [m]$.

  Similarly, for $\mathbf{A} \in \mathbb{R}^{m \times d_2}$, $\boldsymbol{\mathcal{A}} = \boldsymbol{\mathcal{T}} \times_2 \mathbf{A} \in \mathbb{R}^{d_1 \times m \times d_3}$ is such that $\boldsymbol{\mathcal{A}}_{ijk} = \sum_{l=1}^{d_2} \mathbf{A}_{jl} \boldsymbol{\mathcal{T}}_{ilk} \ \forall (i, j, k) \in [d_1] \times [m] \times [d_3]$; or equivalently, $\boldsymbol{\mathcal{A}}_{:,j,:} = \sum_{l=1}^{d_2} \mathbf{A}_{jl} \boldsymbol{\mathcal{T}}_{:,l,:} \in \mathbb{R}^{d_1 \times d_3} \ \forall j \in [m]$;

  In the same way, for $\mathbf{A} \in \mathbb{R}^{m \times d_3}$, the tensor $\boldsymbol{\mathcal{A}} = \boldsymbol{\mathcal{T}} \times_3 \mathbf{A} \in \mathbb{R}^{d_1 \times d_2 \times m}$ is such that $\boldsymbol{\mathcal{A}}_{ijk} = \sum_{l=1}^{d_3} \mathbf{A}_{kl} \boldsymbol{\mathcal{T}}_{ijl} \ \forall (i, j, k) \in [d_1] \times [d_2] \times [m]$; or equivalently, $\boldsymbol{\mathcal{A}}_{:,:,k} = \sum_{l=1}^{d_3} \mathbf{A}_{kl} \boldsymbol{\mathcal{T}}_{:,:,l} \in \mathbb{R}^{d_1 \times d_2} \ \forall k \in [m]$.

  So, for $\mathbf{A}^{(\ell)} \in \mathbb{R}^{m_\ell \times d_\ell} \ \forall \ell \in [3]$, the tensor $\boldsymbol{\mathcal{A}} = \boldsymbol{\mathcal{T}} \times_1 \mathbf{A}^{(1)} \times_2 \mathbf{A}^{(2)} \times_3 \mathbf{A}^{(3)} \in \mathbb{R}^{m_1 \times m_2 \times m_3}$ is such that

  $$\boldsymbol{\mathcal{A}}_{ijk} = \sum_{l=1}^{d_1} \sum_{r=1}^{d_2} \sum_{s=1}^{d_3} \boldsymbol{\mathcal{T}}_{lrs} \mathbf{A}_{il}^{(1)} \mathbf{A}_{jr}^{(2)} \mathbf{A}_{ks}^{(3)} \forall i, j, k \iff \boldsymbol{\mathcal{A}} = \sum_{l=1}^{d_1} \sum_{r=1}^{d_2} \sum_{s=1}^{d_3} \boldsymbol{\mathcal{T}}_{lrs} \mathbf{A}_{:,l}^{(1)} \circ \mathbf{A}_{:,r}^{(2)} \circ \mathbf{A}_{:s}^{(3)} \tag{6}$$

- We denoted by $\boldsymbol{\mathcal{T}}_{(1)} \in \mathbb{R}^{d_1 \times d_2 d_3}$, $\boldsymbol{\mathcal{T}}_{(2)} \in \mathbb{R}^{d_2 \times d_1 d_3}$ and $\boldsymbol{\mathcal{T}}_{(3)} \in \mathbb{R}^{d_3 \times d_1 d_2}$ the mode-1,2,3 unfolding of $\boldsymbol{\mathcal{T}}$, respectively. They are defined by $\left(\boldsymbol{\mathcal{T}}_{(1)}\right)_{i,(j-1)d_3+k} = \left(\boldsymbol{\mathcal{T}}_{(2)}\right)_{j,(k-1)d_1+i} = \left(\boldsymbol{\mathcal{T}}_{(3)}\right)_{k,(j-1)d_1+i} = \boldsymbol{\mathcal{T}}_{ijk} \ \forall (i,j,k) \in [d_1] \times [d_2] \times [d_3]$, or equivalently:

$$
\boldsymbol{\mathcal{T}}_{(1)} = \begin{bmatrix} - \ \mathrm{vec}(\boldsymbol{\mathcal{T}}_{1,:,:}^{\top}) \ - \\ - \ \mathrm{vec}(\boldsymbol{\mathcal{T}}_{2,:,:}^{\top}) \ - \\ - \ \cdots \ - \\ - \mathrm{vec}(\boldsymbol{\mathcal{T}}_{d_1,:,:}^{\top}) - \end{bmatrix} \qquad \boldsymbol{\mathcal{T}}_{(2)} = \begin{bmatrix} - \ \mathrm{vec}(\boldsymbol{\mathcal{T}}_{:,1,:}) \ - \\ - \ \mathrm{vec}(\boldsymbol{\mathcal{T}}_{:,2,:}) \ - \\ - \ \cdots \ - \\ - \mathrm{vec}(\boldsymbol{\mathcal{T}}_{:,d_2,:}) - \end{bmatrix} \qquad \boldsymbol{\mathcal{T}}_{(3)} = \begin{bmatrix} - \ \mathrm{vec}(\boldsymbol{\mathcal{T}}_{:,:,1}) \ - \\ - \ \mathrm{vec}(\boldsymbol{\mathcal{T}}_{:,:,2}) \ - \\ - \ \cdots \ - \\ - \mathrm{vec}(\boldsymbol{\mathcal{T}}_{:,:,d_3}) - \end{bmatrix} \tag{7}
$$

The following identities follow from these definitions using standard tensor and Kronecker-product manipulations.

- For all $\ell \in [3]$, we have

$$
\mathbf{A}\boldsymbol{\mathcal{T}}_{(\ell)} = (\boldsymbol{\mathcal{T}} \times_\ell \mathbf{A})_{(\ell)} \ \forall \mathbf{A} \in \mathbb{R}^{m \times d_\ell} \tag{8}
$$

- We have

$$
(\boldsymbol{\mathcal{T}} \times_2 \mathbf{A} \times_3 \mathbf{B})_{(1)} = \boldsymbol{\mathcal{T}}_{(1)} (\mathbf{A} \otimes \mathbf{B})^{\top} \in \mathbb{R}^{d_1 \times m_2 m_3} \ \forall \mathbf{A} \in \mathbb{R}^{m_2 \times d_2}, \mathbf{B} \in \mathbb{R}^{m_3 \times d_3}
$$
$$
(\boldsymbol{\mathcal{T}} \times_1 \mathbf{A} \times_3 \mathbf{B})_{(2)} = \boldsymbol{\mathcal{T}}_{(2)} (\mathbf{B} \otimes \mathbf{A})^{\top} \in \mathbb{R}^{d_2 \times m_1 m_3} \ \forall \mathbf{A} \in \mathbb{R}^{m_1 \times d_1}, \mathbf{B} \in \mathbb{R}^{m_3 \times d_3} \tag{9}
$$
$$
(\boldsymbol{\mathcal{T}} \times_1 \mathbf{A} \times_2 \mathbf{B})_{(3)} = \boldsymbol{\mathcal{T}}_{(3)} (\mathbf{B} \otimes \mathbf{A})^{\top} \in \mathbb{R}^{d_3 \times m_1 m_2} \ \forall \mathbf{A} \in \mathbb{R}^{m_1 \times d_1}, \mathbf{B} \in \mathbb{R}^{m_2 \times d_2}
$$

- For all $\ell \in [3]$, we have $\boldsymbol{\mathcal{T}} \times_\ell \mathbf{A} \times_\ell \mathbf{B} = \boldsymbol{\mathcal{T}}$ for all $\mathbf{A}, \mathbf{B} \in \mathbb{R}^{d_\ell \times d_\ell}$ such that $\mathbf{B}\mathbf{A} = \mathbb{I}_{d_\ell}$.
- We have

$$
\mathrm{vec}\left((\boldsymbol{\mathcal{T}} \times_1 \mathbf{a})^{\top}\right) = \boldsymbol{\mathcal{T}}_{(1)}^{\top}\mathbf{a} \ \forall \mathbf{a} \in \mathbb{R}^{d_1},
$$
$$
\mathrm{vec}(\boldsymbol{\mathcal{T}} \times_2 \mathbf{a}) = \boldsymbol{\mathcal{T}}_{(2)}^{\top}\mathbf{a} \ \forall \mathbf{a} \in \mathbb{R}^{d_2} \tag{10}
$$
$$
\mathrm{vec}(\boldsymbol{\mathcal{T}} \times_3 \mathbf{a}) = \boldsymbol{\mathcal{T}}_{(3)}^{\top}\mathbf{a} \ \forall \mathbf{a} \in \mathbb{R}^{d_3}
$$

- For $\mathbf{a} \in \mathbb{R}^{d_1}$ and $\mathbf{b} \in \mathbb{R}^{d_2}$, we have

$$
\boldsymbol{\mathcal{T}} \times_1 \mathbf{a} \times_2 \mathbf{b} = \left[\mathbf{a}^{\top}\boldsymbol{\mathcal{T}}_{::k}\mathbf{b}\right]_{k \in [n]}^{\top} = \left[\mathrm{vec}(\mathbf{a}^{\top}\boldsymbol{\mathcal{T}}_{::k}\mathbf{b})\right]_{k \in [n]}^{\top} = \left[(\mathbf{b} \otimes \mathbf{a})^{\top}\mathrm{vec}(\boldsymbol{\mathcal{T}}_{::k})\right]_{k \in [n]}^{\top} = \boldsymbol{\mathcal{T}}_{(3)}(\mathbf{b} \otimes \mathbf{a}) \tag{11}
$$

- If we CP-decompose $\boldsymbol{\mathcal{T}}$ as $\boldsymbol{\mathcal{T}} = [\![\mathbf{A}, \mathbf{B}, \mathbf{C}]\!] := \sum_{\ell=1}^{R} \mathbf{A}_{:,\ell} \circ \mathbf{B}_{:,\ell} \circ \mathbf{C}_{:,\ell}$ with $\mathbf{A} \in \mathbb{R}^{d_1 \times R}$, $\mathbf{B} \in \mathbb{R}^{d_2 \times R}$ and $\mathbf{C} \in \mathbb{R}^{d_3 \times R}$ the three mode loading matrices, then $\boldsymbol{\mathcal{T}}_{(1)} = \mathbf{A}(\mathbf{B} \star \mathbf{C})^{\top}$, $\boldsymbol{\mathcal{T}}_{(2)} = \mathbf{B}(\mathbf{C} \star \mathbf{A})^{\top}$, and $\boldsymbol{\mathcal{T}}_{(3)} = \mathbf{C}(\mathbf{B} \star \mathbf{A})^{\top}$. We recall that $\circ$ is the outer product, so that $\boldsymbol{\mathcal{T}}_{ijk} = \sum_{\ell=1}^{R}(\mathbf{A}_{:,\ell} \circ \mathbf{B}_{:,\ell} \circ \mathbf{C}_{:,\ell})_{ijk} = \sum_{\ell=1}^{R} \mathbf{A}_{i\ell}\mathbf{B}_{j\ell}\mathbf{C}_{k\ell} \ \forall (i,j,k) \in [d_1] \times [d_2] \times [d_3]$.
- The CP-rank of $\boldsymbol{\mathcal{T}}$, denoted $\mathrm{rank}_{\mathrm{CP}}(\boldsymbol{\mathcal{T}})$, is the smallest $R$ for which there exist $\mathbf{A} \in \mathbb{R}^{d_1 \times R}$, $\mathbf{B} \in \mathbb{R}^{d_2 \times R}$ and $\mathbf{C} \in \mathbb{R}^{d_3 \times R}$ such that $\boldsymbol{\mathcal{T}} = [\![\mathbf{A}, \mathbf{B}, \mathbf{C}]\!]$. We always have $\mathrm{rank}_{\mathrm{CP}}(\boldsymbol{\mathcal{T}}) \leq \min\{d_1 d_2, d_1 d_3, d_2 d_3\}$, and when one of the tensor dimensions equals 1, the CP-rank reduces to the usual matrix rank under the natural identification between matrices and third-order tensors with a singleton mode. For instance, if $d_3 = 1$, $\boldsymbol{\mathcal{T}} \in \mathbb{R}^{d_1 \times d_2 \times 1}$ can be identified with a matrix $\boldsymbol{\mathcal{T}}_{:,:,1} = \sum_{\ell=1}^{R} \mathbf{C}_{1,\ell}\mathbf{A}_{:,\ell}\mathbf{B}_{:,\ell}^{\top} \in \mathbb{R}^{d_1 \times d_2}$, which is precisely a rank-$R$ matrix factorization of $\boldsymbol{\mathcal{T}}_{:,:,1}$. Consequently, $\mathrm{rank}_{\mathrm{CP}}(\boldsymbol{\mathcal{T}}) = \mathrm{rank}(\boldsymbol{\mathcal{T}}_{:,:,1})$. The same argument applies whenever $d_1 = 1$ or $d_2 = 1$.

## B. Finite-Dimensional Algebra (FDA)

We intentionally adopt a relatively broad and self-contained exposition in this section in order to make the paper accessible to a wider audience. Readers already familiar with the underlying algebraic background may safely skip most of the introductory material and proceed directly to the proofs of the propositions (many of which rely on standard or straightforward arguments), or move directly to the next section.

### B.1. Definitions

A **monoid** is a set equipped with a binary operation that combines any two elements to form a third element of that set. Specifically, a monoid $(M, \cdot)$ must satisfy $a \cdot b \in M \ \forall a, b \in M$ (*closure*), $(a \cdot b) \cdot c = a \cdot (b \cdot c) \ \forall a, b, c \in M$ (*associativity*), and $\exists e \in M, \ e \cdot a = a \cdot e = a \ \forall a \in M$ (*identity element*).

A **group** is a monoid in which every element admits an inverse. More precisely, a group $(G, \cdot)$ is a monoid such that $\forall a \in G, \exists b \in G, \ a \cdot b = b \cdot a = e$, where $e$ denotes the identity element of $G$. The element $b$ is called the inverse of $a$ and is usually denoted by $a^{-1}$. A group $G$ is said to be abelian or commutative if $a \cdot b = b \cdot a \ \forall a, b \in G$. An additive group is a group where the group operation is typically thought of as addition. This terminology is most commonly used when the group elements are intuitively additive in nature, such as numbers or functions. The group operation is denoted by $+$, the identity element by $0$, and the inverse of an element $a$ is denoted by $-a$. A multiplicative group is a group in which the group operation is typically understood as multiplication. This terminology is used especially when dealing with elements that naturally support a multiplicative structure, such as non-zero numbers, invertible matrices, and permutations. The group operation is denoted by $*$ or simply by juxtaposition $(ab)$, the identity element by $1$, and the inverse of an element $a$ by $a^{-1}$.

A **ring** $(R, +, *)$ is a structure consisting of a set equipped with two binary operations: addition $(+)$ and multiplication $(*)$, such that $(R, +)$ is an abelian (additive) group, $(R, *)$ is a (multiplicative) monoid, and multiplication is distributive over addition ($a * (b + c) = a * b + a * c$ and $(a + b) * c = a * c + b * c$, corresponding to the left and right distributivity respectively). A ring is commutative if $(R, *)$ is a commutative monoid: $a * b = b * a$ for all $a, b \in R$.

*Example* B.1. For $m \in \mathbb{N}^*$, let $\mathcal{M}_m(\mathbb{F})$ denote the set of $m \times m$ matrices with entries in $\mathbb{F}$. Equipped with matrix multiplication, $\mathcal{M}_m(\mathbb{F})$ forms a monoid. The subset $\mathrm{GL}_m(\mathbb{F}) \subset \mathcal{M}_m(\mathbb{F})$ of invertible matrices forms a group under matrix multiplication, the general linear group of degree $m$ over $\mathbb{F}$. Moreover, endowed with the usual matrix addition and multiplication, $\mathcal{M}_m(\mathbb{F})$ forms a non-commutative ring.

A **field** $(\mathbb{F}, +, *)$ is a commutative ring in which every non-zero element has a multiplicative inverse, making $(\mathbb{F} \setminus \{0\}, *)$ an abelian group as well.

*Example* B.2. For a prime integer $p$, $\mathbb{F}_p = \mathbb{Z}/p\mathbb{Z}$ endowed with modular arithmetic forms a finite field, the prime field of order $p$ (also known as the Galois field of order $p$). In general, for any integer $q$, a finite field of order $q$ exists if and only if $q$ is a prime power (i.e., $q = p^k$ where $p$ is a prime number and $k$ is a positive integer). In this case, all fields of order $q$ are isomorphic to each other. In $\mathbb{F}_p$, the multiplicative inverse of an element $\alpha$ is the element $\beta$ (denoted $\alpha^{-1}$) such that $(\alpha\beta)\%p = 1$, and can be computed using the extended Euclidean Algorithm. The result of $\alpha/\beta$ is $(\alpha\beta^{-1})\%p$.

A **vector space** over a field $(\mathbb{F}, +, *)$ is a set $V$ equipped with two operations, vector addition $(V \times V \to V)$ and scalar multiplication $(\mathbb{F} \times V \to V)$, such that $V$ with vector addition forms an abelian group (with identity $0$ and inverses $-\mathbf{v}$) and scalar multiplication is associative and distributive over vector addition, i.e. for all $a, b \in \mathbb{F}$ and $\mathbf{u}, \mathbf{v} \in V$: $a(\mathbf{u} + \mathbf{v}) = a\mathbf{u} + a\mathbf{v}$, $(a + b)\mathbf{u} = a\mathbf{u} + b\mathbf{u}$, $(ab)\mathbf{u} = a(b\mathbf{u})$, $1_{\mathbb{F}}\mathbf{u} = \mathbf{u}$.

A **module** over a ring $R$ generalizes vector spaces by allowing the scalars to come from a ring, not necessarily a field. The key differences are the absence of scalar inverses generally and the fact that the ring does not need to be commutative.

An **algebra** $\mathfrak{A}$ over a field (resp. ring) $(\mathbb{F}, +, *)$ is a vector space (resp. module) equipped with a $\mathbb{F}$-bilinear product[5] $\cdot : \mathfrak{A} \times \mathfrak{A} \to \mathfrak{A}$ such that $(\mathbf{u} + \mathbf{v}) \cdot \mathbf{w} = \mathbf{u} \cdot \mathbf{w} + \mathbf{v} \cdot \mathbf{w}$ (right distributivity), $\mathbf{w} \cdot (\mathbf{u} + \mathbf{v}) = \mathbf{w} \cdot \mathbf{u} + \mathbf{w} \cdot \mathbf{v}$ (left distributivity) and $(\alpha\mathbf{u}) \cdot (\beta\mathbf{v}) = (\alpha * \beta)(\mathbf{u} \cdot \mathbf{v})$ (compatibility with scalars) for all $\mathbf{u}, \mathbf{v}, \mathbf{w} \in \mathfrak{A}$ and $\alpha, \beta \in \mathbb{F}$. When $\cdot$ is associative (resp. commutative), the algebra is said to be associative (resp. commutative). When the multiplicative identity exists, i.e., there exists an element $1_{\mathfrak{A}} \in \mathfrak{A}$ such that $1_{\mathfrak{A}} \cdot \mathbf{u} = \mathbf{u} \cdot 1_{\mathfrak{A}} = \mathbf{u} \ \forall \mathbf{u} \in \mathfrak{A}$, the algebra is said to be unital. The dimension of $\mathfrak{A}$ is its dimension as a $\mathbb{F}$-vector space, denoted by $\dim_{\mathbb{F}} \mathfrak{A}$. We say that $\mathfrak{A}$ is finite-dimensional if $n = \dim_{\mathbb{F}} \mathfrak{A}$ is finite. This means that a finite number of elements in the set can be combined linearly to express any element in the algebra, i.e., there exists a finite basis $\{\mathbf{a}^{(i)}\}_{i \in [n]}$ of $\mathfrak{A}$ (as a vector space over $\mathbb{F}$) such that for every $\mathbf{u} \in \mathfrak{A}$, $\mathbf{u} = \sum_{i=1}^{n} \alpha_i \mathbf{a}^{(i)}$ with $\alpha_i \in \mathbb{F} \ \forall i \in [n]$.

Let $(\mathfrak{A}, \cdot)$ and $(\mathfrak{B}, \times)$ be two $\mathbb{F}$-algebras. A homomorphism of $\mathbb{F}$-algebras $\phi : \mathfrak{A} \to \mathfrak{B}$ is a $\mathbb{F}$-linear map satisfying $\phi(\mathbf{u} \cdot \mathbf{v}) = \phi(\mathbf{u}) \times \phi(\mathbf{v})$ for all $\mathbf{u}, \mathbf{v} \in \mathfrak{A}$; and $\phi(1_{\mathfrak{A}}) = 1_{\mathfrak{B}}$ when $\mathfrak{A}$ and $\mathfrak{B}$ are unitals. The algebra homomorphism $\phi$ is called an isomorphism if it is bijective, or equivalently, if there exists an algebra homomorphism $\varphi : \mathfrak{B} \to \mathfrak{A}$, often denoted $\phi^{-1}$, such that $\varphi(\phi(\mathbf{u})) = \mathbf{u} \ \forall \mathbf{u} \in \mathfrak{A}$ and $\phi(\varphi(\mathbf{v})) = \mathbf{v} \ \forall \mathbf{v} \in \mathfrak{B}$. If further $\mathfrak{B} = \mathfrak{A}$, the isomorphism $\phi$ is called an automorphism of $\mathfrak{A}$.

Two $\mathbb{F}$-algebras $\mathfrak{A}$ and $\mathfrak{B}$ are isomorphic (or, informally, similar) if there exists an algebra isomorphism $\phi : \mathfrak{A} \to \mathfrak{B}$ between them. In other words, $\mathfrak{A}$ and $\mathfrak{B}$ may look different at the level of their elements or chosen bases, but they share exactly the same algebraic structure: the operations in one correspond perfectly to the operations in the other under $\phi$.

---

[5]i.e. a map that is linear in each argument separately: $(\alpha\mathbf{u} + \beta\mathbf{v}) \cdot \mathbf{w} = \alpha(\mathbf{u} \cdot \mathbf{v}) + \beta(\mathbf{u} \cdot \mathbf{w})$ and $\mathbf{w} \cdot (\alpha\mathbf{u} + \beta\mathbf{v}) = \alpha(\mathbf{w} \cdot \mathbf{u}) + \beta(\mathbf{w} \cdot \mathbf{v})$ for all $\mathbf{u}, \mathbf{v}, \mathbf{w} \in \mathfrak{A}$, $\alpha, \beta \in \mathbb{F}$.

## B.2. Structure Constants of FDA

Let $(\mathfrak{A}, \cdot)$ be an $n\mathbb{F}$-FDA and $B = \{\mathbf{a}^{(i)}\}_{i \in [n]}$ a basis of $\mathfrak{A}$ (as a vector space over $\mathbb{F}$). Then the structure of $\mathfrak{A}$ in $B$ is entirely captured by the tensor $\boldsymbol{\mathcal{C}}^{(B)} \in \mathbb{F}^{n \times n \times n}$ defined by

$$\mathbf{a}^{(i)} \cdot \mathbf{a}^{(j)} = \sum_{k=1}^{n} \boldsymbol{\mathcal{C}}_{ijk}^{(B)} \mathbf{a}^{(k)} \tag{12}$$

The elements of $\boldsymbol{\mathcal{C}}^{(B)}$ are called structure coefficients/constants of $\mathfrak{A}$ in $B$. We will call the tensor $\boldsymbol{\mathcal{C}}^{(B)}$ the **structure tensor** of $\mathfrak{A}$ in the basis $B$, and when the context is clear, we will omit $B$ from the notation. That said, we have for all $\mathbf{u} = \sum_{i=1}^{n} \mathbf{u}_i \mathbf{a}^{(i)}$ and $\mathbf{v} = \sum_{i=1}^{n} \mathbf{v}_i \mathbf{a}^{(i)}$ in $\mathfrak{A}$,

$$\mathbf{u} \cdot \mathbf{v} = \sum_{i=1}^{n} \sum_{j=1}^{n} \mathbf{u}_i \mathbf{v}_j (\mathbf{a}^{(i)} \cdot \mathbf{a}^{(j)}) = \sum_{k=1}^{n} \left( \sum_{i=1}^{n} \sum_{j=1}^{n} \boldsymbol{\mathcal{C}}_{ijk} \mathbf{u}_i \mathbf{v}_j \right) \mathbf{a}^{(k)} = \sum_{k=1}^{n} (\boldsymbol{\mathcal{C}} \times_1 \mathbf{u} \times_2 \mathbf{v})_k \, \mathbf{a}^{(k)} \tag{13}$$

Using $(\mathbf{u}, \mathbf{v}) = (\mathbf{a}^{(i)}, \mathbf{a}^{(j)})$, we obtain $\boldsymbol{\mathcal{C}}_{ijk} = \mathbf{a}^{(i)\top} \boldsymbol{\mathcal{C}}_{::k} \mathbf{a}^{(j)}$ for all $i, j, k \in [n]$. Note that (see Equation (11))

$$\boldsymbol{\mathcal{C}} \times_1 \mathbf{u} \times_2 \mathbf{v} = \boldsymbol{\mathcal{C}}_{(3)} (\mathbf{v} \otimes \mathbf{u}) \tag{14}$$

with (Equation (7))

$$\boldsymbol{\mathcal{C}}_{(3)}^\top := \begin{bmatrix} | & | & & | \\ \mathrm{vec}(\boldsymbol{\mathcal{C}}_{::1}) & \mathrm{vec}(\boldsymbol{\mathcal{C}}_{::2}) & \cdots & \mathrm{vec}(\boldsymbol{\mathcal{C}}_{::n}) \\ | & | & & | \end{bmatrix} \in \mathbb{F}^{n^2 \times n} \tag{15}$$

The Proposition 2.1 states that for all $\boldsymbol{\mathcal{C}} \in \mathbb{F}^{n \times n \times n}$, there exists an $n\mathbb{F}$-FDA whose structure tensor is $\boldsymbol{\mathcal{C}}$ is some basis $B$; and that all $n\mathbb{F}$-FDA that have $\boldsymbol{\mathcal{C}}$ as a structure tensor in one of their bases are isomorphic to each other. The proof of the first part is given by Lemma B.1 and the proof of the second part is given by Lemma B.2.

**Lemma B.1.** *For a ring $\mathbb{F}$ (which can be a field), let $\boldsymbol{\mathcal{C}} \in \mathbb{F}^{n \times n \times n}$, and $\times : \mathbb{F}^n \times \mathbb{F}^n \to \mathbb{F}^n$ defined by $\mathbf{u} \times \mathbf{v} = \boldsymbol{\mathcal{C}} \times_1 \mathbf{u} \times_2 \mathbf{v}$. Then $(\mathbb{F}^n, \times)$ is an $n\mathbb{F}$-FDA, and $\boldsymbol{\mathcal{C}}$ its structure tensor in its canonical basis $\{\mathbf{e}^{(i)}\}_{i \in [n]}$.*

*Proof.* It is easy to check that this structure forms an algebra over $\mathbb{F}$. For the structure tensor, observe that $(\mathbf{u} \times \mathbf{v})_k = (\boldsymbol{\mathcal{C}} \times_1 \mathbf{u} \times_2 \mathbf{v})_k = \sum_{=1}^{n} \sum_{j=1}^{n} \boldsymbol{\mathcal{C}}_{ijk} \mathbf{u}_i \mathbf{v}_j$ for all $\mathbf{u}, \mathbf{v} \in \mathbb{F}^n$. So $(\mathbf{e}^{(i)} \times \mathbf{e}^{(j)})_k = \boldsymbol{\mathcal{C}}_{ijk} \, \forall i, j, k \in [n]$. $\square$

**Lemma B.2.** *Let $(\mathfrak{A}, \cdot)$ be an $n\mathbb{F}$-FDA with structure tensor $\boldsymbol{\mathcal{C}} \in \mathbb{F}^{n \times n \times n}$ in a basis $B = \{\mathbf{a}^{(i)}\}_{i \in [n]}$, and $\{\mathbf{e}^{(i)}\}_{i \in [n]}$ be the canonical basis of $\mathbb{F}^n$. The linear map $\phi : \mathbb{F}^n \to \mathfrak{A}$ defined by $\phi(\mathbf{e}^{(i)}) = \mathbf{a}^{(i)} \, \forall i \in [n]$ is an isomorphism (of vector spaces), and we have the relation $\phi(\mathbf{u} \times \mathbf{v}) = \phi(\mathbf{u}) \cdot \phi(\mathbf{v}) \, \forall \mathbf{u}, \mathbf{v} \in \mathbb{F}^n$. As a consequence, by defining the product $\mathbf{u} \times \mathbf{v} = \boldsymbol{\mathcal{C}} \times_1 \mathbf{u} \times_2 \mathbf{v}$ on $\mathbb{F}^n$, $\phi$ becomes an algebra isomorphism between $(\mathbb{F}^n, \times)$ and $(\mathfrak{A}, \cdot)$.*

*Proof.* It is easy to see that $\phi$ is an isomorphism of vector spaces. We also have for all $\mathbf{u}, \mathbf{v} \in \mathbb{F}^n$:

$$\begin{aligned} \phi(\mathbf{u} \times \mathbf{v}) &= \sum_{k=1}^{n} (\boldsymbol{\mathcal{C}} \times_1 \mathbf{u} \times_2 \mathbf{v})_k \, \phi(\mathbf{e}^{(k)}) = \sum_{k=1}^{n} \left( \sum_{i=1}^{n} \sum_{j=1}^{n} \boldsymbol{\mathcal{C}}_{ijk} \mathbf{u}_i \mathbf{v}_j \right) \mathbf{a}^{(k)} \\ &= \sum_{i=1}^{n} \sum_{j=1}^{n} \mathbf{u}_i \mathbf{v}_j \sum_{k=1}^{n} \boldsymbol{\mathcal{C}}_{ijk} \mathbf{a}^{(k)} \\ &= \sum_{i=1}^{n} \sum_{j=1}^{n} \mathbf{u}_i \mathbf{v}_j (\mathbf{a}^{(i)} \cdot \mathbf{a}^{(j)}) = \left( \sum_{i=1}^{n} \mathbf{u}_i \mathbf{a}^{(i)} \right) \cdot \left( \sum_{j=1}^{n} \mathbf{v}_j \mathbf{a}^{(j)} \right) \\ &= \left( \sum_{i=1}^{n} \mathbf{u}_i \phi(\mathbf{e}^{(i)}) \right) \cdot \left( \sum_{j=1}^{n} \mathbf{v}_j \phi(\mathbf{e}^{(j)}) \right) \\ &= \phi \left( \sum_{i=1}^{n} \mathbf{u}_i \mathbf{e}^{(i)} \right) \cdot \phi \left( \sum_{j=1}^{n} \mathbf{v}_j \mathbf{e}^{(j)} \right) = \phi(\mathbf{u}) \cdot \phi(\mathbf{v}) \end{aligned} \tag{16}$$

$\square$

This shows that learning multiplication $\cdot$ in $(\mathfrak{A}, \cdot)$ is equivalent to learning $\times$ in $\mathbb{F}[\mathcal{C}] := (\mathbb{F}^n, \times)$. If we are working in an arbitrary basis $\{\mathbf{a}^{(i)}\}_{i \in [n]}$ of $\mathbb{F}^n$ such that $\mathbf{a}^{(i)} = \sum_{k=1}^{n} \mathbf{P}_{ki} \mathbf{e}^{(k)} \forall i \in [n]$, we need to rescale $\mathcal{C}$ accordingly.

**Proposition B.3.** *Let $\mathcal{C}$ and $\tilde{\mathcal{C}}$ be the structure tensors of an $n\mathbb{F}$-FDA $\mathfrak{A}$ with respect to two bases $B = \{\mathbf{a}^{(i)})\}_{i \in [n]}$ and $\tilde{B} = \{\tilde{\mathbf{a}}^{(i)}\}_{i \in [n]}$ respectively. If $\mathbf{P} \in \mathbb{F}^{n \times n}$ is the basis change matrix from $B$ to $\tilde{B}$, i.e. $\tilde{\mathbf{a}}^{(i)} = \sum_{k=1}^{n} \mathbf{P}_{ki} \mathbf{a}^{(k)} \, \forall i \in [n]$, then we have $\tilde{\mathcal{C}} = \mathcal{C} \times_1 \mathbf{P}^\top \times_2 \mathbf{P}^\top \times_3 \mathbf{P}^{-1}$, or equivalently, $\tilde{\mathcal{C}} \times_3 \mathbf{P} = \mathcal{C} \times_1 \mathbf{P}^\top \times_2 \mathbf{P}^\top$. If further $B$ and $\tilde{B}$ are orthogonal basis, then $\tilde{\mathcal{C}} = \mathcal{C} \times_1 \mathbf{P}^\top \times_2 \mathbf{P}^\top \times_3 \mathbf{P}^\top$.*

*Proof.* For all $i, j \in [n]$, we have

$$
\begin{aligned}
\tilde{\mathbf{a}}^{(i)} \cdot \tilde{\mathbf{a}}^{(j)} &= \left( \sum_{\ell=1}^{n} \mathbf{P}_{\ell i} \mathbf{a}^{(\ell)} \right) \cdot \left( \sum_{r=1}^{n} \mathbf{P}_{rj} \mathbf{a}^{(r)} \right) \\
&= \sum_{\ell=1}^{n} \sum_{r=1}^{n} \mathbf{P}_{\ell i} \mathbf{P}_{rj} \left( \mathbf{a}^{(\ell)} \cdot \mathbf{a}^{(r)} \right) \\
&= \sum_{\ell=1}^{n} \sum_{r=1}^{n} \mathbf{P}_{\ell i} \mathbf{P}_{rj} \sum_{s=1}^{n} \mathcal{C}_{\ell rs} \mathbf{a}^{(s)} \\
&= \sum_{\ell=1}^{n} \sum_{r=1}^{n} \mathbf{P}_{\ell i} \mathbf{P}_{rj} \sum_{s=1}^{n} \mathcal{C}_{\ell rs} \sum_{k=1}^{n} (\mathbf{P}^{-1})_{ks} \tilde{\mathbf{a}}^{(k)} \\
&= \sum_{k=1}^{n} \left( \sum_{\ell, r, s=1}^{n} \mathcal{C}_{\ell rs} (\mathbf{P}^\top)_{i\ell} (\mathbf{P}^\top)_{jr} (\mathbf{P}^{-1})_{ks} \right) \tilde{\mathbf{a}}^{(k)} \\
&= \sum_{k=1}^{n} \left( \mathcal{C} \times_1 \mathbf{P}^\top \times_2 \mathbf{P}^\top \times_3 \mathbf{P}^{-1} \right)_{ijk} \tilde{\mathbf{a}}^{(k)} \text{ (Equation (6))}
\end{aligned}
\tag{17}
$$

So $\tilde{\mathcal{C}} = \mathcal{C} \times_1 \mathbf{P}^\top \times_2 \mathbf{P}^\top \times_3 \mathbf{P}^{-1}$. We have (see Equation (8)) $(\tilde{\mathcal{C}} \times_3 \mathbf{P})_{(3)} = \mathbf{P} \tilde{\mathcal{C}}_{(3)} = \mathbf{P} (\mathcal{C} \times_1 \mathbf{P}^\top \times_2 \mathbf{P}^\top \times_3 \mathbf{P}^{-1})_{(3)} = \mathbf{P} \mathbf{P}^{-1} (\mathcal{C} \times_1 \mathbf{P}^\top \times_2 \mathbf{P}^\top)_{(3)} = (\mathcal{C} \times_1 \mathbf{P}^\top \times_2 \mathbf{P}^\top)_{(3)}$. So $\tilde{\mathcal{C}} \times_3 \mathbf{P} = \mathcal{C} \times_1 \mathbf{P}^\top \times_2 \mathbf{P}^\top$. If further $B$ and $\tilde{B}$ are orthogonal basis, then $\mathbf{P}^{-1} = \mathbf{P}^\top$. In fact, let $\mathbf{B} = [-\mathbf{a}^{(i)} -]_{i \in [n]}^\top \in \mathbb{F}^{n \times n}$ ($\mathbf{a}_i$ is the $i^{th}$ column of $\mathbf{B}$) and $\tilde{\mathbf{B}} = [-\tilde{\mathbf{a}}^{(i)} -]_{i \in [n]}^\top \in \mathbb{F}^{n \times n}$. We have $\tilde{\mathbf{B}} = \mathbf{B} \mathbf{P}$. Assuming $\mathbf{B}^\top \mathbf{B} = \mathbb{I}_n$, this give $\mathbf{P} = \mathbf{B}^\top \tilde{\mathbf{B}}$, so that $\mathbf{P}^\top \mathbf{P} = \mathbb{I}_n$ if $\mathbf{B} \mathbf{B}^\top = \tilde{\mathbf{B}}^\top \tilde{\mathbf{B}} = \mathbb{I}_n$ and $\mathbf{P} \mathbf{P}^\top = \mathbb{I}_n$ if $\tilde{\mathbf{B}} \tilde{\mathbf{B}}^\top = \mathbf{B}^\top \mathbf{B} = \mathbb{I}_n$. $\square$

As a consequence of this proposition, we get that the rank of the structure tensor of a FDA is basis-invariant: it is an intrinsic invariant of the algebra.

**Corollary B.4.** *For all $i \in [3]$, the rank of $\mathcal{C}_{(i)}^{(B)} \in \mathbb{F}^{n \times n^2}$ is independant from the basis $B$.*

*Proof.* If $\mathbf{P}$ is the basis change matrix from an arbitrary basis $B$ to another basis $\tilde{B}$, then $\mathcal{C}^{(\tilde{B})} = \mathcal{C}^{(B)} \times_1 \mathbf{P}^\top \times_2 \mathbf{P}^\top \times_3 \mathbf{P}^{-1}$ by Proposition B.3. This implies (see Equations (8) and (9)):

- $\mathcal{C}_{(1)}^{(\tilde{B})} = \left( \mathcal{C}^{(B)} \times_1 \mathbf{P}^\top \times_2 \mathbf{P}^\top \times_3 \mathbf{P}^{-1} \right)_{(1)} = \mathbf{P}^\top \left( \mathcal{C}^{(B)} \times_2 \mathbf{P}^\top \times_3 \mathbf{P}^{-1} \right)_{(1)} = \mathbf{P}^\top \mathcal{C}_{(1)}^{(B)} (\mathbf{P}^\top \otimes \mathbf{P}^{-1})^\top$

- $\mathcal{C}_{(2)}^{(\tilde{B})} = \left( \mathcal{C}^{(B)} \times_1 \mathbf{P}^\top \times_2 \mathbf{P}^\top \times_3 \mathbf{P}^{-1} \right)_{(2)} = \mathbf{P}^\top \left( \mathcal{C}^{(B)} \times_1 \mathbf{P}^\top \times_3 \mathbf{P}^{-1} \right)_{(2)} = \mathbf{P}^\top \mathcal{C}_{(2)}^{(B)} (\mathbf{P}^{-1} \otimes \mathbf{P}^\top)^\top$

- $\mathcal{C}_{(3)}^{(\tilde{B})} = \left( \mathcal{C}^{(B)} \times_1 \mathbf{P}^\top \times_2 \mathbf{P}^\top \times_3 \mathbf{P}^{-1} \right)_{(3)} = \mathbf{P}^{-1} \left( \mathcal{C}^{(B)} \times_1 \mathbf{P}^\top \times_2 \mathbf{P}^\top \right)_{(3)} = \mathbf{P}^{-1} \mathcal{C}_{(3)}^{(B)} (\mathbf{P} \otimes \mathbf{P})$

In each case, $\mathcal{C}_{(i)}^{(\tilde{B})}$ is obtained from $\mathcal{C}_{(i)}^{(B)}$ by left- and right-multiplication with invertible matrices. Such operations preserve matrix rank, hence $\text{rank} \left( \mathcal{C}_{(i)}^{(\tilde{B})} \right) = \text{rank} \left( \mathcal{C}_{(i)}^{(B)} \right) \, \forall i \in [3]$. $\square$

Sparsity, on the other hand, is not basis-invariant. But it may be possible to find a basis of $\mathfrak{A}$ in which $\mathcal{C}$ is as sparse as possible, making the problem of learning $\mathfrak{A}$ simpler to study; $\tilde{\mathcal{C}} \in \arg\min_{\mathbf{P} \neq \mathbb{I}_n, \text{rank}(\mathbf{P})=n} \left\| \mathcal{C} \times_1 \mathbf{P}^\top \times_2 \mathbf{P}^\top \times_3 \mathbf{P}^{-1} \right\|_0$.

For example, if we go from a basis $B = \{\mathbf{a}^{(i)}\}_{i \in [n]}$ to its orthonormal equivalent $Q = \{\mathbf{q}^{(i)}\}_{i \in [n]}$, then we have $\mathbf{B} = \mathbf{QP}$ under the QR decomposition, with $\mathbf{B} = [-\mathbf{a}^{(i)}-]_{i \in [n]}^{\top} \in \mathbb{F}^{n \times n}$ ($\mathbf{a}_i$ is the $i^{th}$ column of $\mathbf{B}$), $\mathbf{Q} = [-\mathbf{q}^{(i)}-]_{i \in [n]}^{\top} \in \mathbb{F}^{n \times n}$ orthogonal, and $\mathbf{P} \in \mathbb{F}^{n \times n}$ upper triangular. Note that $\mathcal{C}_{ijk}^{(Q)} = \sum_{\ell,r,s} \mathcal{C}_{\ell rs}^{(B)} \mathbf{P}_{\ell i} \mathbf{P}_{rj} (\mathbf{P}^{-1})_{ks}$, with $\mathbf{P}^{-1}$ also upper triangular.

Although any tensor $\mathcal{C} \in \mathbb{F}^{n \times n \times n}$ defines an $n\mathbb{F}$-FDA, a key question is how the algebraic properties of $(\mathfrak{A}, \cdot)$ are reflected in its structure of $\mathcal{C}$. The following proposition makes this connection explicit by characterizing associativity, commutativity, and the existence of a unit directly in terms of tensor equations.

**Proposition B.5.** *Let $(\mathfrak{A}, \cdot)$ be an $n\mathbb{F}$-FDA with structure tensor $\mathcal{C} \in \mathbb{F}^{n \times n \times n}$ in $B = \{\mathbf{a}^{(i)}\}_{i \in [n]}$. $\mathfrak{A}$ is*

(i) *associative if and only if $\sum_{k=1}^{n} \mathcal{C}_{ijk} \mathcal{C}_{klm} = \sum_{k=1}^{n} \mathcal{C}_{ikm} \mathcal{C}_{jlk} \; \forall i,j,l,m \in [n]$;*

(ii) *commutative if and only if $\mathcal{C}$ is symmetric is its first two modes, i.e. $\mathcal{C}_{ij:} = \mathcal{C}_{ji:} \; \forall i,j \in [n]$, or equivalently, $\mathcal{C}_{::k}^{\top} = \mathcal{C}_{::k} \forall k \in [n]$;*

(iii) *unital if and only if there exist $\lambda \in \mathbb{F}^n$ such that $\mathcal{C} \times_1 \lambda = \mathbb{I}_n = \mathcal{C} \times_2 \lambda$, that is $\sum_{i=1}^{n} \lambda_i \mathcal{C}_{i,:,:} = \mathbb{I}_n = \sum_{i=1}^{n} \lambda_i \mathcal{C}_{:,i,:}$, or equivalently, $\mathcal{C}_{(1)}^{\top} \lambda = \mathrm{vec}(\mathbb{I}_n) = \mathcal{C}_{(2)}^{\top} \lambda$. In this case, $1_{\mathfrak{A}} = \sum_{i=1}^{n} \lambda_i \mathbf{a}^{(i)}$, with $\lambda \in \mathbb{F}^n$ the unique solution of the system of equations $\mathcal{C}_{(1)}^{\top} \lambda = \mathrm{vec}(\mathbb{I}_n) = \mathcal{C}_{(2)}^{\top} \lambda$.*

*Proof.* (i) For associative FDA, the multiplication must satisfy $(\mathbf{a}^{(i)} \cdot \mathbf{a}^{(j)}) \cdot \mathbf{a}^{(l)} = \mathbf{a}^{(i)} \cdot (\mathbf{a}^{(j)} \cdot \mathbf{a}^{(l)}) \; \forall i,j,l$. Expanding this using the structure constants (Equation (12)), we require $\sum_{m=1}^{n} (\sum_{k=1}^{n} \mathcal{C}_{ijk} \mathcal{C}_{klm}) \mathbf{a}^{(m)} = \sum_{m=1}^{n} (\sum_{k=1}^{n} \mathcal{C}_{ikm} \mathcal{C}_{jlk}) \mathbf{a}^{(m)} \; \forall i,j,l \in [n]$.

(ii) For commutative FDA, the multiplication must satisfy $\mathbf{a}^{(i)} \cdot \mathbf{a}^{(j)} = \mathbf{a}^{(j)} \cdot \mathbf{a}^{(i)} \; \forall i,j \in [n] \iff \sum_{k=1}^{n} \mathcal{C}_{ijk} \mathbf{a}^{(k)} = \sum_{k=1}^{n} \mathcal{C}_{jik} \mathbf{a}^{(k)} \; \forall i,j \in [n] \iff \mathcal{C}_{ij:} = \mathcal{C}_{ji:} \; \forall i,j \in [n]$.

(iii) First, consider the algebra $(\mathbb{F}[\mathcal{C}], \times)$ generated by $\mathcal{C}$. Suppose $(\mathbb{F}[\mathcal{C}], \times)$ is unital, i.e. there exists $\lambda \in \mathbb{F}^n$ such that $\mathbf{u} \times \lambda = \mathbf{u} = \lambda \times \mathbf{u}$ for all $\mathbf{u} \in \mathbb{F}^n$. Then, expanding using Equation (13), wet get $\mathcal{C} \times_1 \mathbf{u} \times_2 \lambda = \mathbf{u} = \mathcal{C} \times_1 \lambda \times_2 \mathbf{u} \; \forall \mathbf{u} \in \mathbb{F}^n$, which is equivalent to $\mathcal{C} \times_2 \lambda = \mathbb{I}_n = \mathcal{C} \times_1 \lambda$. The direction $\Longleftarrow$ of this equivalence is obvious, considering the fact that $\mathbb{I}_n \times_1 \mathbf{u} = \mathbb{I}_n^{\top} \mathbf{u} = \mathbf{u}$ and $\mathbb{I}_n \times_2 \mathbf{u} = \mathbb{I}_n \mathbf{u} = \mathbf{u}$ for all $\mathbf{u} \in \mathbb{F}^n$ (Equation (5)). The direction $\Longrightarrow$ follows by taking $\mathbf{u}$ as a $k^{\text{th}}$ canonical vector of $\mathbb{F}^n$, for all $k \in [n]$. We also have

$$
\begin{aligned}
\mathcal{C} \times_1 \lambda = \mathbb{I}_n = \mathcal{C} \times_2 \lambda &\iff (\mathcal{C} \times_1 \lambda)^{\top} = \mathbb{I}_n = \mathcal{C} \times_2 \lambda \\
&\iff \mathrm{vec}\left((\mathcal{C} \times_1 \lambda)^{\top}\right) = \mathrm{vec}(\mathbb{I}_n) = \mathrm{vec}(\mathcal{C} \times_2 \lambda) \\
&\iff \mathcal{C}_{(1)}^{\top} \lambda = \mathrm{vec}(\mathbb{I}_n) = \mathcal{C}_{(2)}^{\top} \lambda \;\text{(Equation (10))}
\end{aligned}
\tag{18}
$$

Now, suppose there exist $\lambda, \tilde{\lambda} \in \mathbb{F}^n$ such that $\mathcal{C} \times_1 \lambda = \mathbb{I}_n = \mathcal{C} \times_2 \lambda$ and $\mathcal{C} \times_1 \tilde{\lambda} = \mathbb{I}_n = \mathcal{C} \times_2 \tilde{\lambda}$. From $\mathcal{C} \times_1 \lambda = \mathbb{I}_n$, we get $\mathcal{C} \times_1 \lambda \times_2 \tilde{\lambda} = \mathbb{I}_n \times_2 \tilde{\lambda} = \tilde{\lambda}$; and from $\mathbb{I}_n = \mathcal{C} \times_2 \tilde{\lambda}$, we get $\lambda = \mathbb{I}_n \times_1 \lambda = \mathcal{C} \times_2 \tilde{\lambda} \times_1 \lambda$. So $\lambda = \mathcal{C} \times_1 \lambda \times_2 \tilde{\lambda} = \tilde{\lambda}$. Therefore, when a solution $\lambda$ to $\mathcal{C}_{(1)}^{\top} \lambda = \mathrm{vec}(\mathbb{I}_n) = \mathcal{C}_{(2)}^{\top} \lambda$ exists, it is unique, and thus equals $1_{\mathbb{F}[\mathcal{C}]}$. Finally, since $(\mathbb{F}[\mathcal{C}], \times)$ and $(\mathfrak{A}, \cdot)$ are isomorphic by the linear map $\phi : \mathbb{F}^n \to \mathfrak{A}$ defined by $\phi(\mathbf{e}^{(i)}) = \mathbf{a}^{(i)} \; \forall i \in [n]$ (Lemma B.2), if $(\mathbb{F}[\mathcal{C}], \times)$ is unital with $1_{\mathbb{F}[\mathcal{C}]} = \sum_{i=1}^{n} \lambda_i \mathbf{e}^{(i)}$, then $(\mathfrak{A}, \cdot)$ is also unital with $1_{\mathfrak{A}} = \phi(1_{\mathbb{F}[\mathcal{C}]}) = \sum_{i=1}^{n} \lambda_i \phi(\mathbf{e}^{(i)}) = \sum_{i=1}^{n} \lambda_i \mathbf{a}^{(i)}$. $\qquad \square$

The corollary of (iii) in this proposition tells us how we can get the unit element of $(\mathbb{F}[\mathcal{C}], \times)$ in $B = \{\mathbf{e}^{(i)}\}_{i \in [n]}$ when we only have access to $\mathcal{C}$ and the condition of its existence.

**Corollary B.6.** *Let $(\mathfrak{A}, \cdot)$ be an $n\mathbb{F}$-FDA with structure tensor $\mathcal{C} \in \mathbb{F}^{n \times n \times n}$ in $B = \{\mathbf{a}^{(i)}\}_{i \in [n]}$. Let $\mathbf{A} := \left[\mathcal{C}_{(1)}, \mathcal{C}_{(2)}\right]^{\top} \in \mathbb{F}^{2n^2 \times n}$ and $\mathbf{b} := \left[\mathrm{vec}(\mathbb{I}_n), \mathrm{vec}(\mathbb{I}_n)\right]^{\top} \in \mathbb{F}^{2n^2}$. $\mathfrak{A}$ is unital if and only if $\mathrm{rank}_{\mathbb{F}}(\mathbf{A}) = \mathrm{rank}_{\mathbb{F}}([\mathbf{A}|\mathbf{b}])$, where $[\mathbf{A}|\mathbf{b}] \in \mathbb{F}^{2n^2 \times (n+1)}$ is the augmented matrix of $\mathbf{A}$ with $\mathbf{b}$ added as a column. In this case, $1_{\mathfrak{A}} = \sum_{i=1}^{n} \lambda_i \mathbf{a}^{(i)}$, with $\lambda$ the unique solution to $\mathbf{A}\lambda = \mathbf{b}$.*

*Proof.* We need to check that there exists a joint solution to the two $\lambda$ linear equations $\mathcal{C}_{(k)}^{\top} \lambda = \mathrm{vec}(\mathbb{I}_n)$, $k \in \{1,2\}$, that is $\mathbf{A}\lambda = \mathbf{b}$. The system has $2n^2$ equations for $n$ variables, and solutions are cosets of $\ker(\mathbf{A})$. We just need to check the system's consistency (in $\mathbb{F}$, this precision is important). $\qquad \square$

We can also solve the system $\mathbf{A}\lambda = \mathbf{b}$ to find the coordinates of the unit element of $\mathfrak{A}$ in $B = \{\mathbf{a}^{(i)}\}_{i \in [n]}$. It all depends on the field we are in. This can be done quite easily in $\mathbb{R}$ or $\mathbb{C}$. The system $\mathbf{A}\lambda = \mathbf{b}$ has solutions if and only if $\mathbf{A}\mathbf{A}^{\dagger}\mathbf{b} = \mathbf{b}$, and the solutions are the vectors $\lambda = \mathbf{A}^{\dagger}\mathbf{b} + (\mathbb{I} - \mathbf{A}^{\dagger}\mathbf{A})\mathbf{u}$ for an arbitrary conformable vector $\mathbf{u}$. Here, $\mathbf{A}^{\dagger}$ is the pseudo-inverse of $\mathbf{A}$ : $\mathbf{A}\mathbf{A}^{\dagger}\mathbf{A} = \mathbf{A}$ and $\mathbf{A}^{\dagger}\mathbf{A}\mathbf{A}^{\dagger} = \mathbf{A}^{\dagger}$.

In a field like $\mathbb{F}_p$, a given solution of the system is not guaranteed to be in $\mathbb{F}$, and solving the system is not easy. In fact, in this field, everything is modulo $p$, for example, Equation (13) should be read $\left(\sum_i \mathbf{u}_i \mathbf{a}^{(i)}\right) \cdot \left(\sum_i \mathbf{v}_i \mathbf{a}^{(i)}\right) = \sum_k \left(\left(\sum_{i,j} \mathcal{C}_{ijk}\mathbf{u}_i\mathbf{v}_j\right)\%p\right)\mathbf{a}^{(k)}$, and the system, $(\mathbf{A}\lambda)\%p = \mathbf{b}$. The multiplicative inverse of an element $\alpha$ is the element $\beta$ (denoted $\alpha^{-1}$) such that $(\alpha\beta)\%p = 1$, and can be computed using the extended Euclidean Algorithm. The result of $\alpha/\beta$ is $(\alpha\beta^{-1})\%p$. So, these details must be considered if we opt for Gaussian Elimination to solve the system.

It should be noted that associative, unital, noncommutative algebras only exist for $n \geq 3$ over any field $\mathbb{F}$.

**Proposition B.7.** *For any field $\mathbb{F}$, the smallest integer $n \in \mathbb{N}^*$ for which there exists an associative, unital, noncommutative $n\mathbb{F}$-FDA is $n = 3$. Equivalently, (i) every associative and unital $n\mathbb{F}$-FDA with $n \leq 2$ is commutative; (ii) and there exists an associative, unital, noncommutative $3\mathbb{F}$-FDA.*

*Proof.* We split the proof into two parts.

**(i) Nonexistence for $n \leq 2$.**

For $n = 1$, let $(\mathfrak{A}, \cdot)$ be a unital $1\mathbb{F}$-FDA. Since $\dim_{\mathbb{F}}(\mathfrak{A}) = 1$, the unit element $1_{\mathfrak{A}}$ forms a basis of $\mathfrak{A}$. Therefore $\mathfrak{A} = \{\lambda 1_{\mathfrak{A}}, \lambda \in \mathbb{F}\}$. Hence, for all $\mathbf{u} = \lambda 1_{\mathfrak{A}}$ and $\mathbf{v} = \mu 1_{\mathfrak{A}}$ in $\mathfrak{A}$, we have

$$\mathbf{u} \cdot \mathbf{v} = (\lambda 1_{\mathfrak{A}}) \cdot (\mu 1_{\mathfrak{A}}) = \lambda\mu(1_{\mathfrak{A}} \cdot 1_{\mathfrak{A}}) = \lambda\mu 1_{\mathfrak{A}} = \mu\lambda 1_{\mathfrak{A}} = \mathbf{v} \cdot \mathbf{u} \tag{19}$$

So $\mathfrak{A}$ is commutative. Now let $n = 2$, and let $(\mathfrak{A}, \cdot)$ be an associative and unital $2\mathbb{F}$-FDA. Since $\dim_{\mathbb{F}}(\mathfrak{A}) = 2$ and $1_{\mathfrak{A}} \neq 0$, there exists $\mathbf{e} \in \mathfrak{A}$ such that $B = \{1_{\mathfrak{A}}, \mathbf{e}\}$ is a basis of $\mathfrak{A}$. Since $\mathfrak{A}$ is closed under multiplication, there exist $\alpha, \beta \in \mathbb{F}$ such that $\mathbf{e}^2 = \alpha 1_{\mathfrak{A}} + \beta\mathbf{e}$. Let $\mathbf{u} = a1_{\mathfrak{A}} + b\mathbf{e}$ and $\mathbf{v} = c1_{\mathfrak{A}} + d\mathbf{e}$ be two arbitrary elements of $\mathfrak{A}$, with $a, b, c, d \in \mathbb{F}$. Then

$$\begin{aligned}
\mathbf{u} \cdot \mathbf{v} &= (a1_{\mathfrak{A}} + b\mathbf{e}) \cdot (c1_{\mathfrak{A}} + d\mathbf{e}) \\
&= ac1_{\mathfrak{A}} \cdot 1_{\mathfrak{A}} + ad1_{\mathfrak{A}} \cdot \mathbf{e} + bc\mathbf{e} \cdot 1_{\mathfrak{A}} + bd\mathbf{e}^2 \\
&= ac\,1_{\mathfrak{A}} + ad\mathbf{e} + bc\mathbf{e} + bd(\alpha 1_{\mathfrak{A}} + \beta\mathbf{e}) \\
&= (ac + \alpha bd)1_{\mathfrak{A}} + (ad + bc + \beta bd)\mathbf{e}
\end{aligned} \tag{20}$$

The right-hand side is symmetric in $(a, b)$ and $(c, d)$, since $ac + \alpha bd = ca + \alpha db$ and $ad + bc + \beta bd = cb + da + \beta db$. Therefore $\mathbf{u} \cdot \mathbf{v} = \mathbf{v} \cdot \mathbf{u}$ $\forall \mathbf{u}, \mathbf{v} \in \mathfrak{A}$. So every associative and unital $2\mathbb{F}$-FDA is commutative.

**(ii) Existence for $n = 3$.**

Consider

$$\mathfrak{A} = \mathrm{UT}_2(\mathbb{F}) := \left\{ \begin{pmatrix} a & b \\ 0 & c \end{pmatrix} \ : \ a, b, c \in \mathbb{F} \right\},$$

equipped with the usual matrix multiplication. This is the algebra of upper triangular $2 \times 2$ matrices over $\mathbb{F}$. We have $\mathfrak{A} = \mathrm{span}_{\mathbb{F}}\left\{\mathbf{a}^{(1)}, \mathbf{a}^{(2)}, \mathbf{a}^{(3)}\right\}$ where

$$\mathbf{a}^{(1)} = \begin{pmatrix} 1 & 0 \\ 0 & 0 \end{pmatrix}, \quad \mathbf{a}^{(2)} = \begin{pmatrix} 0 & 1 \\ 0 & 0 \end{pmatrix}, \quad \mathbf{a}^{(3)} = \begin{pmatrix} 0 & 0 \\ 0 & 1 \end{pmatrix} \tag{21}$$

These three matrices are linearly independent, so $\dim_{\mathbb{F}}(\mathfrak{A}) = 3$. Since matrix multiplication is bilinear and associative, $(\mathfrak{A}, \cdot)$ is an associative $3\mathbb{F}$-FDA. It is also unital, with unit $1_{\mathfrak{A}} = \mathbf{a}^{(1)} + \mathbf{a}^{(3)}$. Finally, $\mathfrak{A}$ is not commutative, because $\mathbf{a}^{(1)}\mathbf{a}^{(2)} = \mathbf{a}^{(2)} \neq 0 = \mathbf{a}^{(2)}\mathbf{a}^{(1)}$. $\qquad\square$

### B.3. Examples of FDA

Examples B.3, B.4 and B.5 are associative, commutative and unital. Example B.6 is associative and unital but not commutative. The algebra of quaternions is associative and unitary but non-commutative. The algebra of octonions is unitary but neither associative nor commutative. Lie algebras (Example B.8) are not associative, not commutative, nor unital.

*Example* B.3 (Complex Numbers). $\mathbb{F} = \mathbb{R}$, $n = 2$ and $\mathfrak{A} = \mathbb{C} = (\mathbb{F}^2, \cdot)$. The product $\cdot$ is defined as $(a + b\mathbf{i}) \cdot (c + d\mathbf{i}) := (ac - bd) + (ad + bc)\mathbf{i}$ with $\mathbf{i}^2 = -1$. The structure tensor of $\mathfrak{A}$ in its canonical basis $(\mathbf{a}^{(1)}, \mathbf{a}^{(2)}) \equiv (1, \mathbf{i})$ is

$$\begin{bmatrix} \mathcal{C}_{111} & \mathcal{C}_{112} \\ \mathcal{C}_{121} & \mathcal{C}_{122} \end{bmatrix}, \begin{bmatrix} \mathcal{C}_{211} & \mathcal{C}_{212} \\ \mathcal{C}_{221} & \mathcal{C}_{222} \end{bmatrix} = \begin{bmatrix} 1 & 0 \\ 0 & 1 \end{bmatrix}, \begin{bmatrix} 0 & 1 \\ -1 & 0 \end{bmatrix} \tag{22}$$

since $\mathbf{a}^{(1)} \cdot \mathbf{a}^{(1)} = \mathbf{a}^{(1)} \implies \mathcal{C}_{11:} = [1, 0]$, $\mathbf{a}^{(1)} \cdot \mathbf{a}^{(2)} = \mathbf{a}^{(2)} \cdot \mathbf{a}^{(1)} = \mathbf{a}^{(2)} \implies \mathcal{C}_{12:} = \mathcal{C}_{21:} = [0, 1]$, and $\mathbf{a}^{(2)} \cdot \mathbf{a}^{(2)} = -\mathbf{a}^{(1)} \implies \mathcal{C}_{22:} = [-1, 0]$. In the field $\mathbb{F} = \mathbb{Z}/p\mathbb{Z}$ for $p$ prime, we have $\mathbf{i} = (-1)\%p = p - 1$, so $\mathcal{C}_{221} = p - 1$.

In the same way, we can also calculate the structure constants of quaternions ($n = 4$) and octonions ($n = 8$).

*Example* B.4 (Dual Numbers). $\mathfrak{A} = (\mathbb{F}^2, \cdot)$ with product $\cdot$ is defined as $(a + b\varepsilon) \cdot (c + d\varepsilon) := ac + (ad + bc)\varepsilon$, where $\varepsilon^2 = 0$. The structure tensor of $\mathfrak{A}$ in its canonical basis $(\mathbf{a}^{(1)}, \mathbf{a}^{(2)}) \equiv (1, \varepsilon)$ is

$$\begin{bmatrix} \mathcal{C}_{111} & \mathcal{C}_{112} \\ \mathcal{C}_{121} & \mathcal{C}_{122} \end{bmatrix}, \begin{bmatrix} \mathcal{C}_{211} & \mathcal{C}_{212} \\ \mathcal{C}_{221} & \mathcal{C}_{222} \end{bmatrix} = \begin{bmatrix} 1 & 0 \\ 0 & 1 \end{bmatrix}, \begin{bmatrix} 0 & 1 \\ 0 & 0 \end{bmatrix} \tag{23}$$

*Example* B.5 (Polynomials modulo $x^n - 1$). $\mathfrak{A} = (\mathbb{F}_{n-1}[x], \cdot)$, with $\mathbb{F}_{n-1}[x]$ the set of polynomial of degree at most $n - 1$ on $\mathbb{F}$. The product $\cdot$ is defined as $\left( \sum_{i=0}^{n-1} a_i x^i \right) \cdot \left( \sum_{i=0}^{n-1} b_i x^i \right) := \sum_{i,j=0}^{n-1} a_i b_i x^{(i+j)\%n} = \sum_{k=0}^{n-1} \left( \sum_{i,j=0}^{n-1} a_i b_i \delta_{k,(i+j)\%n} \right) x^k$. Equivalently, $\mathfrak{A} \simeq \mathbb{F}[x]/(x^n - 1)$, with $\mathbb{F}[x]$ the polynomial ring in one indeterminate $x$ over the field $\mathbb{F}$. The structure tensor of $\mathfrak{A}$ in its canonical basis $(\mathbf{a}^{(0)}, \ldots, \mathbf{a}^{(n-1)}) \equiv (x^0, \cdots, x^{n-1})$ is $\mathcal{C}_{ijk} = \delta_{k,(i+j)\%n} \ \forall i, j, k \in [\![0, n-1]\!]$ since $x^i \cdot x^j = x^{(i+j) \bmod n}$. Here we adopt 0-based indexing, i.e., indices range from 0 to $n - 1$, for the convenience of notation.

*Example* B.6 (Square matrices of size $n$). $\mathfrak{A} = (\mathcal{M}_n(\mathbb{F}), \cdot)$, with $\mathcal{M}_n(\mathbb{F})$ the set of square matrices of size $n$ with entries in $\mathbb{F}$. The product $\cdot$ is defined as $(\mathbf{A} \cdot \mathbf{B})_{ij} := \sum_{k=1}^n \mathbf{A}_{ik} \mathbf{B}_{kj}$. Let's consider the canonical basis $B = \{\mathbf{a}^{(i)}\}_{i \in [n^2]} = \{\mathbf{a}^{((q-1)n+r)}\}_{(q,r) \in [n]^2}$ of $\mathcal{M}_n(\mathbb{F})$, where each $\mathbf{a}^{((q-1)n+r)}$ has $1_\mathbb{F}$ at position $(q, r) \in [n]^2$ and $0_\mathbb{F}$ everywhere. Since $\mathbf{a}^{((q_1-1)n+r_1)} \cdot \mathbf{a}^{((q_2-1)n+r_2)} = \delta_{q_2 r_1} \mathbf{a}^{((q_1-1)n+r_2)}$, the structure tensor of $\mathfrak{A}$ in $B$ is $\mathcal{C}_{(q_1-1)n+r_1, (q_2-1)n+r_2, (q_3-1)n+r_3} = \delta_{q_2 r_1} \delta_{q_1 q_3} \delta_{r_2 r_3}$.

*Example* B.7 (Upper triangular matrices of size $t$). $\mathfrak{A} = (\mathrm{UT}_t(\mathbb{F}), \cdot)$, with $\mathrm{UT}_t(\mathbb{F}) := \{\mathbf{A} \in \mathcal{M}_t(\mathbb{F}) : \mathbf{A}_{ij} = 0_\mathbb{F} \forall i > j\}$ the set of upper triangular square matrices of size $m$ with entries in $\mathbb{F}$. The product $\cdot$ is defined as the usual matrix multiplication. The dimension of $\mathrm{UT}_t(\mathbb{F})$ is $n = \frac{t(t+1)}{2}$, since the free entries are exactly those on and above the diagonal.

*Example* B.8. A Lie algebra $(\mathfrak{A}, \cdot)$ is an algebra whose binary operation $\cdot$ (generally denoted by $[\cdot, \cdot]$ and called the Lie bracket) satisfies the alternating property (or antisymmetry) $\mathbf{u} \cdot \mathbf{u} = 0 \ \forall \mathbf{u} \in \mathfrak{A}$, and the Jacobi identity $(\mathbf{u} \cdot \mathbf{v}) \cdot \mathbf{w} + (\mathbf{v} \cdot \mathbf{w}) \cdot \mathbf{u} + (\mathbf{w} \cdot \mathbf{u}) \cdot \mathbf{v} = 0 \ \forall \mathbf{u}, \mathbf{v}, \mathbf{w} \in \mathfrak{A}$.

For any associative algebra $(\mathfrak{A}, \cdot)$, one can define a Lie algebra using $[\mathbf{u}, \mathbf{v}] = \mathbf{u} \cdot \mathbf{v} - \mathbf{v} \cdot \mathbf{u} \ \forall \mathbf{u}, \mathbf{v} \in \mathfrak{A}$. For example, if we replace matrix multiplication $\mathbf{AB}$ with the matrix commutator $\mathbf{A} \cdot \mathbf{B} := \mathbf{AB} - \mathbf{BA}$, then $\mathfrak{A} = (\mathcal{M}_n(\mathbb{F}), \cdot)$ becomes a Lie algebra, with $\mathcal{C}_{(q_1-1)n+r_1, (q_2-1)n+r_2, (q_3-1)n+r_3} = \delta_{q_2 r_1} \delta_{q_1 q_3} \delta_{r_2 r_3} - \delta_{q_1 r_2} \delta_{q_2 q_3} \delta_{r_1 r_3}$. In general, we have the following result about the structure tensor of Lie Algebra.

**Proposition B.8.** *An $n\mathbb{F}$-FDA $(\mathfrak{A}, \cdot)$ with structure tensor $\mathcal{C} \in \mathbb{F}^{n \times n \times n}$ in the basis $B = \{\mathbf{a}^{(i)}\}_{i \in [n]}$ is a Lie algebra if and only if*

(i) **skew-symmetric:**

$$\mathcal{C}_{ijk} = -\mathcal{C}_{jik} \qquad \forall i, j, k \in [n]; \tag{24}$$

(ii) **Jacobi identity:**

$$\sum_{k=1}^n \mathcal{C}_{ijk} \mathcal{C}_{k\ell m} + \sum_{k=1}^n \mathcal{C}_{j\ell k} \mathcal{C}_{kim} + \sum_{k=1}^n \mathcal{C}_{\ell ik} \mathcal{C}_{kjm} = 0 \qquad \forall i, j, \ell, m \in [n] \tag{25}$$

*Proof.* Recall that $(\mathfrak{A}, \cdot)$ is a Lie algebra if and only if its product is bilinear, skew-symmetric, and satisfies the Jacobi identity. Since $(\mathfrak{A}, \cdot)$ is already an $n\mathbb{F}$-FDA, bilinearity is part of the definition. It therefore remains to characterize skew-symmetry and the Jacobi identity in terms of $\mathcal{C}$.

(i) We first characterize skew-symmetry. For all $i, j \in [n]$, we have $\mathbf{a}^{(i)} \cdot \mathbf{a}^{(j)} = \sum_{k=1}^{n} \mathcal{C}_{ijk} \mathbf{a}^{(k)}$ and $\mathbf{a}^{(j)} \cdot \mathbf{a}^{(i)} = \sum_{k=1}^{n} \mathcal{C}_{jik} \mathbf{a}^{(k)}$. Thus,

$$\mathbf{a}^{(i)} \cdot \mathbf{a}^{(j)} = -\mathbf{a}^{(j)} \cdot \mathbf{a}^{(i)} \ \forall i, j \in [n] \iff \sum_{k=1}^{n} \mathcal{C}_{ijk} \mathbf{a}^{(k)} = \sum_{k=1}^{n} (-\mathcal{C}_{jik}) \mathbf{a}^{(k)} \quad \forall i, j \in [n] \tag{26}$$

Since $B = \{\mathbf{a}^{(k)}\}_{k \in [n]}$ is a basis, this holds if and only if Equation (24) holds.

(ii) We now characterize the Jacobi identity. For all $i, j, \ell \in [n]$, we have

$$(\mathbf{a}^{(i)} \cdot \mathbf{a}^{(j)}) \cdot \mathbf{a}^{(\ell)} = \sum_{k=1}^{n} \mathcal{C}_{ijk} \left( \mathbf{a}^{(k)} \cdot \mathbf{a}^{(\ell)} \right) = \sum_{k=1}^{n} \mathcal{C}_{ijk} \sum_{m=1}^{n} \mathcal{C}_{k\ell m} \mathbf{a}^{(m)} = \sum_{m=1}^{n} \left( \sum_{k=1}^{n} \mathcal{C}_{ijk} \mathcal{C}_{k\ell m} \right) \mathbf{a}^{(m)} \tag{27}$$

Similarly,

$$(\mathbf{a}^{(j)} \cdot \mathbf{a}^{(\ell)}) \cdot \mathbf{a}^{(i)} = \sum_{m=1}^{n} \left( \sum_{k=1}^{n} \mathcal{C}_{j\ell k} \mathcal{C}_{kim} \right) \mathbf{a}^{(m)} \quad \text{and} \quad (\mathbf{a}^{(\ell)} \cdot \mathbf{a}^{(i)}) \cdot \mathbf{a}^{(j)} = \sum_{m=1}^{n} \left( \sum_{k=1}^{n} \mathcal{C}_{\ell i k} \mathcal{C}_{kjm} \right) \mathbf{a}^{(m)} \tag{28}$$

Thus, the Jacobi identity

$$(\mathbf{a}^{(i)} \cdot \mathbf{a}^{(j)}) \cdot \mathbf{a}^{(\ell)} + (\mathbf{a}^{(j)} \cdot \mathbf{a}^{(\ell)}) \cdot \mathbf{a}^{(i)} + (\mathbf{a}^{(\ell)} \cdot \mathbf{a}^{(i)}) \cdot \mathbf{a}^{(j)} = 0 \qquad \forall i, j, \ell \in [n] \tag{29}$$

is equivalent to

$$\sum_{m=1}^{n} \left[ \sum_{k=1}^{n} \mathcal{C}_{ijk} \mathcal{C}_{k\ell m} + \sum_{k=1}^{n} \mathcal{C}_{j\ell k} \mathcal{C}_{kim} + \sum_{k=1}^{n} \mathcal{C}_{\ell i k} \mathcal{C}_{kjm} \right] \mathbf{a}^{(m)} = 0 \qquad \forall i, j, \ell \in [n] \tag{30}$$

Since $B$ is a basis, this holds if and only if Equation (25) holds. $\qquad\square$

### B.4. From Groups to FDA: A Unified View

**Definition B.9.** For a group $(G, \circ)$ and a field $\mathbb{F}$, the group algebra of $G$ over $\mathbb{F}$, denoted $(\mathbb{F}[G], \cdot)$, is the set of all linear combinations of elements of $G$ with coefficients in $\mathbb{F}$. If $G$ has $n$ elements $\{g_i\}_{i \in [n]}$, then $\mathbb{F}[G] = \{\sum_{i=1}^{n} \alpha_i g_i \mid \alpha_1, \ldots, \alpha_n \in \mathbb{F}\}$ and $(\sum_{i=1}^{n} \alpha_i g_i) \cdot (\sum_{j=1}^{n} \beta_j g_j) = \sum_{i,j=1}^{n} \alpha_i \beta_i (g_i \circ g_j) = \sum_{k=1}^{n} \left( \sum_{i,j=1}^{n} \alpha_i \beta_j \mathbb{1}(g_i \circ g_j = g_k) \right) g_k$, with $\mathbb{1}$ the indicator function. We have $1_{\mathbb{F}[G]} = e$, with $e$ is the identity of $G$. The structure tensor of $\mathbb{F}[G]$ in $\{g_i\}_{i=1}^{n}$ is $\mathcal{C}_{ijk} = \mathbb{1}(g_k = g_i \circ g_j) \in \{0, 1\} \ \forall i, j, k \in [n]$.

Note that the formal expression $\sum_{i=1}^{n} \alpha_i g_i$ should not be interpreted as a genuine linear combination inside $G$, since a group has no intrinsic compatibility between its operation $\circ$ and the scalar or additive structure of $\mathbb{F}$. As explained in Section 3, this expression acquires meaning once we pass to one-hot encoding. Redefine $\mathbb{F}[G] := \{\sum_{i=1}^{n} \alpha_i \mathbf{a}^{(i)} \mid \alpha \in \mathbb{F}^n\}$ where $\mathbf{a}^{(i)} = \Psi(g_i)$ and $\Psi : G \to \{0_{\mathbb{F}}, 1_{\mathbb{F}}\}^n$ is the one-hot encoding of $G$ over $\mathbb{F}$, given by $\Psi(g_i) := [\mathbb{1}(g_k = g_i)]_{k \in [n]}$. Multiplication in $\mathbb{F}[G]$, which is a free vector space over $G$ with multiplication given by linearly extending the group law, is then defined by $\mathbf{u} \cdot \mathbf{v} := \sum_{i,j} \mathbf{u}_i \mathbf{v}_j \Psi(g_i \circ g_j) = \sum_{k=1}^{n} \left( \sum_{i,j=1}^{n} \mathbf{u}_i \mathbf{v}_j \mathbb{1}(g_i \circ g_j = g_k) \right) \mathbf{a}^{(k)}$ for all $\mathbf{u} = \sum_i \mathbf{u}_i \mathbf{a}^{(i)}$ and $\mathbf{v} = \sum_j \mathbf{v}_j \mathbf{a}^{(j)}$. In this way, linear combinations of group elements become well-defined elements of the vector space $\mathbb{F}[G]$, and the group multiplication naturally extends to an algebra product on $\mathbb{F}[G]$. With this construction, we can show that $(\mathbb{F}[G], \cdot)$ is an $n\mathbb{F}$-FDA, and that training a model on the group $(G, \circ)$ is equivalent to training it on the multiplication in $(\mathbb{F}[G], \cdot)$.

**Proposition B.10.** *For a finite group $(G, \circ)$ with $n$ elements $\{g_i\}_{i \in [n]}$ and identity $e$, $(\mathbb{F}[G], \cdot)$ is an $n$-dimensional associative and unital $\mathbb{F}$-FDA with $1_{\mathbb{F}[G]} = \Psi(e)$. Also, $(\mathbb{F}[G], \cdot)$ is commutative if and only if $G$ is commutative. Moreover, the structure tensor of $\mathbb{F}[G]$ in $B = \{\Psi(g_i)\}_{i \in [n]}$ is given by $\mathcal{C}_{ijk} = \mathbb{1}_{\mathbb{F}}(g_i \circ g_j = g_k) \ \forall i, j, k \in [n]$; and we have $\Psi(g_i \circ g_j) = \mathcal{C} \times_1 \Psi(g_i) \times_2 \Psi(g_j) \ \forall i, j \in [n]$.*

*Proof.* It is easy to check that $\mathbb{F}[G]$ is a vector space over $\mathbb{F}$, and that $\cdot$ verifies the axioms of an algebra operation (bilinearity, right and left distributivity, compatibility with scalars). The associativity and unitality conditions follow from

the fact that $G$ is by definition associative and unital. Now, let $i, j \in [n]$. We have $\Psi(g_i) \cdot \Psi(g_j) = \Psi(g_i \circ g_j)$, so $\mathcal{C}_{ijk} = \mathbb{1}(g_k = g_i \circ g_j) = (\Psi(g_i \circ g_j))_k \; \forall k \in [n]$. If $G$ is commutative, then $\mathbb{F}[G]$ is commutative by Proposition B.5 since $\mathcal{C}_{ijk} = \mathbb{1}(g_i \circ g_j = g_k) = \mathbb{1}(g_j \circ g_i = g_k) = \mathcal{C}_{jik} \; \forall i, j, k \in [n]$. The converse is also true by the same proposition. Finally, we have $\Psi(g_i \circ g_j) = \mathcal{C} \times_1 \Psi(g_i) \times_2 \Psi(g_j)$ for all $i, j \in [n]$ since

$$(\mathcal{C} \times_1 \Psi(g_i) \times_2 \Psi(g_j))_k = \sum_{l,r=1}^{n} \mathcal{C}_{lrk} (\Psi(g_i))_l (\Psi(g_j))_r = \sum_{l,r=1}^{n} \mathcal{C}_{lrk} \delta_{li} \delta_{rj} = \mathcal{C}_{ijk} = (\Psi(g_i \circ g_j))_k \; \forall k \in [n] \quad (31)$$

$$\square$$

*Remark* B.11. Given a group $G = \{g_i\}_{i \in [n]}$, the structure tensor of $\mathbb{F}[G]$ in $B = \{\Psi(g_i)\}_{i \in [n]}$ is the same for all $\mathbb{F} = \mathbb{Z}/p\mathbb{Z}$, $p > 1$. For all $i, j, k \in [n]$, we have $\mathcal{C}_{ijk} = 1$ if $g_i \circ g_j = g_k$ and 0 otherwise. So for groups and finite fields, we can always choose $p = 2$ without loss of generality.

*Remark* B.12. The assignment sending a group $G$ to its group algebra $\mathbb{F}[G]$ is a functor from the category of groups to the category of $\mathbb{F}$-algebras. Indeed, to every group homomorphism $\phi : G \to H$ one can associate the algebra homomorphism $\mathbb{F}[\phi] : \mathbb{F}[G] \to \mathbb{F}[H]$ defined by $\mathbb{F}[\phi] \left( \sum_{g \in G} \alpha_g g \right) := \sum_{g \in G} \alpha_g \phi(g) = \sum_{h \in H} \left( \sum_{g \in G, \, \phi(g)=h} \alpha_g \right) h$. This construction preserves identities ($\mathbb{F}[\text{id}_G] = \text{id}_{\mathbb{F}[G]}$ with $\text{id}_G$ the identity map on the group $G$, and $\text{id}_{\mathbb{F}[G]}$ the identity map on the algebra $\mathbb{F}[G]$) and composition ($\mathbb{F}[\psi \circ \phi] = \mathbb{F}[\psi] \circ \mathbb{F}[\phi]$ for all $\phi : G \to H$ and $\psi : H \to K$). Hence, it defines a functor. As a consequence, $\mathbb{F}[\Psi]$ is an isomorphism between $(\mathbb{F}[G], \cdot)$ and $(\mathbb{F}[\mathcal{C}], \times)$ in Proposition B.10.

$$\begin{array}{ccc} G & \xrightarrow{\phi} & H \\ \downarrow & & \downarrow \\ \mathbb{F}[G] & \xrightarrow{\mathbb{F}[\phi]} & \mathbb{F}[H] \end{array} \quad (32)$$

*Example* B.9 (Group algebra of the additive group $\mathbb{Z}/n\mathbb{Z}$ over $\mathbb{F}$). $\mathfrak{A} = (\mathbb{F}[\mathbb{Z}/n\mathbb{Z}], \cdot)$, with $\mathbb{Z}/n\mathbb{Z} = \{g_i\}_{i \in [\![0,n-1]\!]} \equiv [\![0, n-1]\!]$ and $g_i \circ g_j = (i+j) \mod n = g_{(i+j) \mod n} \; \forall i, j \in [\![0, n-1]\!]$. So $\mathcal{C}_{ijk} = \mathbb{1}(g_k = g_i \circ g_j) = \delta_{k,(i+j)\%n} \; \forall i, j, k \in [\![0, n-1]\!]$. This is the structure tensor of $\mathbb{F}_{n-1}[x] = \mathbb{F}[x]/(x^n - 1)$ in its canonical basis (Example B.5), so $\mathbb{F}[\mathbb{Z}/n\mathbb{Z}] \cong \mathbb{F}_{n-1}[x]$. Here we adopt 0-based indexing for the convenience of notation.

*Example* B.10 (Group algebra of the Dihedral group $D_t$ over $\mathbb{F}$). $\mathfrak{A} = (\mathbb{F}[D_t], \cdot)$, with $D_t = (r_0, \ldots, r_{t-1}, s_0, \ldots, s_{t-1})$ the group of symmetries of a regular $t$-gon, where each $r_i$ denotes the rotation of angle $2\pi i/t$ and each $s_i$ the reflection across the axis passing through vertex $i$ (the axis at angle $\pi i/t$). The group composition is given by

$$r_i \circ r_j = r_{(i+j)\%t}, \quad r_i \circ s_j = s_{(i+j)\%t}, \quad s_i \circ r_j = s_{(i-j)\%t}, \quad s_i \circ s_j = r_{(i-j)\%t} \quad \forall i, j \in [\![0, t-1]\!] \quad (33)$$

We have $n = \dim_{\mathbb{F}} \mathfrak{A} = |D_t| = 2t$. Let us index the elements of $D_t = \{g_i\}_{i \in [\![0,n-1]\!]}$ by $g_i = r_i$ for $i \in [\![0, t-1]\!]$ and $g_i = s_{i-t}$ for $i \in [\![t, n-1]\!]$. Then, for all $i, j, k \in [\![0, n-1]\!]$, the structure tensor of $\mathfrak{A}$ in its canonical basis $\{g_i\}_{i \in [\![0,n-1]\!]}$ is given by

$$\mathcal{C}_{ijk} = \begin{cases} \delta_{k,(i+j)\%t} & \text{if } 0 \le i, j < t & \text{since } g_i \circ g_j = r_j \circ r_i = r_{(i+j)\%t} = g_{(i+j)\%t} \\ \delta_{k,t+(i+j-t)\%t} & \text{if } 0 \le i < t, \, t \le j < 2t & \text{since } g_i \circ g_j = r_i \circ s_{j-t} = s_{(i+j-t)\%t} = g_{t+(i+j-t)\%t} \\ \delta_{k,t+(i-t-j)\%t} & \text{if } t \le i < 2t, 0 \le j < t & \text{since } g_i \circ g_j = s_{i-t} \circ r_j = s_{(i-t-j)\%t} = g_{t+(i-t-j)\%t} \\ \delta_{k,((i-t)-(j-t))\%t} & \text{if } t \le i, j < 2t & \text{since } g_i \circ g_j = s_{i-t} \circ s_{j-t} = r_{(i-t-j+t)\%t} = g_{(i-t-j+t)\%t} \end{cases}$$

*Example* B.11 (Group algebra of the symmetric group $S_t$ over $\mathbb{F}$). $\mathfrak{A} = (\mathbb{F}[S_t], \cdot)$. The elements of $\mathbb{F}[S_t]$ can be seen as endomorphisms of $\mathbb{F}^t$ and can be written $\mathbf{u} = \sum_{\sigma \in S_t} \mathbf{u}_\sigma \sigma$, i.e. $\mathbf{u}$ takes an element of $\mathbb{F}^t$, applies all the permutations of $S_t$ to it, then makes a linear combination of these permutations with coefficients $(\mathbf{u}_\sigma)_{\sigma \in S_t}$ in $\mathbb{F}$. The product $\cdot$ is define as $\left( \sum_{\sigma \in S_t} \mathbf{u}_\sigma \sigma \right) \cdot \left( \sum_{\pi \in S_t} \mathbf{v}_\pi \pi \right) = \sum_{\sigma, \pi \in S_t} \mathbf{u}_\sigma \mathbf{v}_\pi (\sigma \circ \pi) = \sum_{\gamma \in S_t} \left( \sum_{\sigma \circ \pi = \gamma} \mathbf{u}_\sigma \mathbf{v}_\pi \right) \gamma$, where $\circ$ is the composition of permutations. We have $n = \dim_{\mathbb{F}} \mathfrak{A} = |S_t| = t!$.

## B.5. Representation of FDA

Structure-like groups have representations with respect to their elements and the operators acting on them.

**Definition B.13.** A representation of a group $(G, \circ)$ over a field $\mathbb{F}$ is a group homomorphism $\rho : G \to \mathrm{GL}_m(\mathbb{F})$ such that $\rho(g \circ h) = \rho(g)\rho(h) \; \forall g, h \in G$ and $\rho(e) = \mathbb{I}_m$, where $e$ is the identity of $G$.

For example, any cyclic group of order $p$ generated by $g$ has the following 2-dimensional irreducible representation:

$$\rho(g^k) = \begin{pmatrix} \cos\left(\frac{2\pi k}{p}\right) & -\sin\left(\frac{2\pi k}{p}\right) \\ \sin\left(\frac{2\pi k}{p}\right) & \cos\left(\frac{2\pi k}{p}\right) \end{pmatrix} \; \forall k \in [\![0, p]\!] \tag{34}$$

Each element $g^k$ is represented by the angle rotation $2\pi k/p$. An example of a finite cyclic group of order $p$ is the additive group $(\mathbb{F}_p, +)$ for $p$ a prime integer.

**Definition B.14.** The representation of an $\mathbb{F}$-algebra $(\mathfrak{A}, \cdot)$ involves a homomorphism $\rho$ from $\mathfrak{A}$ to the algebra of matrices (Example B.6), preserving both the linear and the multiplicative structures, i.e. $\rho(\alpha \mathbf{u} + \beta \mathbf{v}) = \alpha \rho(\mathbf{u}) + \beta \rho(\mathbf{v})$ and $\rho(\mathbf{u} \cdot \mathbf{v}) = \rho(\mathbf{u})\rho(\mathbf{v})$ for all $\mathbf{u}, \mathbf{v} \in \mathfrak{A}$ and $\alpha, \beta \in \mathbb{F}$. When $\mathfrak{A}$ is unital, it is also required that $\rho(1_\mathfrak{A}) = \mathbb{I}$, the identity matrix.

This notion of algebra representation is central to the interpretation of the results of this paper. However, the theory of algebra representations is somewhat more abstract than that of groups, involving concepts such as (right or left) modules of algebra, semi-simplicity, and others. Nevertheless, the picture becomes less abstract (though still difficult) if we work directly with the structure tensor of the algebra. Let us consider the representations of an $n\mathbb{F}$-FDA $(\mathfrak{A}, \cdot)$ over $\mathbb{F}$ itself, of dimension $m \in \mathbb{N}^*$. From the Definition B.14, we see that it is necessary and sufficient (Proposition B.15) to find a representation $\mathcal{R}_i := \rho(\mathbf{a}^{(i)}) \in \mathbb{F}^{m \times m}$ of the elements of a basis $B = \{\mathbf{a}^{(i)}\}_{i \in [n]}$ of $\mathfrak{A}$ in order to represent all other elements using

$$\rho\left(\sum_{k=1}^n \alpha_k \mathbf{a}^{(k)}\right) = \sum_{k=1}^n \alpha_k \mathcal{R}_k \; \forall(\alpha_1, \cdots, \alpha_n) \in \mathbb{F}^n \tag{35}$$

In this way, we always have $\rho(\alpha \mathbf{u} + \beta \mathbf{v}) = \alpha \rho(\mathbf{u}) + \beta \rho(\mathbf{v})$ for all $\mathbf{u}, \mathbf{v} \in \mathfrak{A}$ and $\alpha, \beta \in \mathbb{F}$. To also ensure that $\rho(\mathbf{u} \cdot \mathbf{v}) = \rho(\mathbf{u})\rho(\mathbf{v}) \; \forall \mathbf{u}, \mathbf{v} \in \mathfrak{A}$, it is necessary and sufficient (Propositon B.16) that

$$P_{i,j}(\mathcal{R}) := \mathcal{R}_i \mathcal{R}_j - \sum_{k=1}^n \mathcal{C}_{ijk}^{(B)} \mathcal{R}_k = 0 \; \forall i, j \in [n] \tag{36}$$

That is,

$$P_{i,j}^{l,r}(\mathcal{R}) := \sum_{s=1}^m \mathcal{R}_{ils} \mathcal{R}_{jsr} - \sum_{k=1}^n \mathcal{C}_{ijk}^{(B)} \mathcal{R}_{klr} = 0 \; \forall i, j \in [n]; \; l, r \in [m] \tag{37}$$

**Proposition B.15.** *Let $(\mathfrak{A}, \cdot)$ be an $n\mathbb{F}$-FDA. For all representation $\rho$ and all basis $B = \{\mathbf{a}^{(i)}\}_{i \in [n]}$ of $\mathfrak{A}$, we have $\rho\left(\sum_{k=1}^n \alpha_k \mathbf{a}^{(k)}\right) = \sum_{k=1}^n \alpha_k \rho\left(\mathbf{a}^{(k)}\right) \; \forall(\alpha_1, \cdots, \alpha_n) \in \mathbb{F}^n$ and $\rho\left(\mathbf{a}^{(i)}\right)\rho\left(\mathbf{a}^{(j)}\right) = \sum_{k=1}^n \mathcal{C}_{ijk}^{(B)} \rho\left(\mathbf{a}^{(k)}\right) \; \forall i, j \in [n]$.*

*Proof.* $\rho\left(\sum_{k=1}^n \alpha_k \mathbf{a}^{(k)}\right) = \sum_{k=1}^n \alpha_k \rho\left(\mathbf{a}^{(k)}\right) \; \forall \alpha \in \mathbb{F}^n$ since $\rho$ is an algebra homomorphism. For all $i, j \in [n]$ :

$$\begin{aligned} \rho\left(\mathbf{a}^{(i)}\right)\rho\left(\mathbf{a}^{(j)}\right) &= \rho\left(\mathbf{a}^{(i)} \cdot \mathbf{a}^{(j)}\right) \text{ since } \rho \text{ is an algebra homomorphism} \\ &= \rho\left(\sum_{k=1}^n \mathcal{C}_{ijk}^{(B)} \mathbf{a}^{(k)}\right) \text{ by the definition of } \mathcal{C}^{(B)} \\ &= \sum_{k=1}^n \mathcal{C}_{ijk}^{(B)} \rho\left(\mathbf{a}^{(k)}\right) \end{aligned} \tag{38}$$

$\square$

**Proposition B.16.** *Let $(\mathfrak{A}, \cdot)$ be an $n\mathbb{F}$-FDA with structure tensor $\mathcal{C} \in \mathbb{F}^{n \times n \times n}$ in $B = \{\mathbf{a}^{(i)}\}_{i \in [n]}$. Fix $m \in \mathbb{N}^*$ and $\mathcal{R} \in \mathbb{F}^{n \times m \times m}$. Set $\mathcal{R}_k = \mathcal{R}_{k,:,:} \in \mathbb{F}^{m \times m} \; \forall k \in [n]$, and define the linear map $\rho : \mathfrak{A} \to \mathcal{M}_m(\mathbb{F})$ by*

$$\rho\left(\sum_{k=1}^n \alpha_k \mathbf{a}^{(k)}\right) := \sum_{k=1}^n \alpha_k \mathcal{R}_k \; \forall(\alpha_1, \cdots, \alpha_n) \in \mathbb{F}^n \tag{39}$$

*Then $\rho$ is an algebra homomorphism if and only if*

$$\mathcal{R}_i \mathcal{R}_j = \sum_{k=1}^{n} \mathcal{C}_{ijk} \mathcal{R}_k \quad \forall i, j \in [n] \tag{40}$$

*If $\mathfrak{A}$ is unital with $1_{\mathfrak{A}} = \sum_{i=1}^{n} \lambda_i \mathbf{a}^{(i)}$, then the additional condition $\rho(1_{\mathfrak{A}}) = \mathbb{I}_m$ is equivalent to*

$$\sum_{i=1}^{n} \lambda_i \mathcal{R}_i = \mathbb{I}_m \tag{41}$$

*Proof.* ($\Longrightarrow$) Suppose $\rho$ is an algebra homomorphism. Then, we have for all $i, j \in [n]$ :

$$
\begin{aligned}
\mathcal{R}_i \mathcal{R}_j &= \rho(\mathbf{a}^{(i)}) \rho(\mathbf{a}^{(j)}) \text{ by the definition of } \rho \\
&= \rho\left(\mathbf{a}^{(i)} \cdot \mathbf{a}^{(j)}\right) \text{ since } \rho \text{ is an algebra homomorphism} \\
&= \rho\left(\sum_{k=1}^{n} \mathcal{C}_{ijk} \mathbf{a}^{(k)}\right) \text{ by the definition of } \mathcal{C} \text{ (Equation (12))} \\
&= \sum_{k=1}^{n} \mathcal{C}_{ijk} \mathcal{R}_k \text{ by the definition of } \rho
\end{aligned}
\tag{42}
$$

($\Longleftarrow$) Assume (40) holds. For all $\mathbf{u} = \sum_{i=1}^{n} \mathbf{u}_i \mathbf{a}^{(i)}$ and $\mathbf{v} = \sum_{i=1}^{n} \mathbf{v}_i \mathbf{a}^{(i)}$, we have

$$
\begin{aligned}
\rho(\mathbf{u})\rho(\mathbf{v}) &= \left(\sum_{i=1}^{n} \mathbf{u}_i \mathcal{R}_i\right)\left(\sum_{i=1}^{n} \mathbf{v}_i \mathcal{R}_i\right) \text{ by the definition of } \rho \\
&= \sum_{i,j=1}^{n} \mathbf{u}_i \mathbf{v}_j \mathcal{R}_i \mathcal{R}_j \\
&= \sum_{i,j=1}^{n} \mathbf{u}_i \mathbf{v}_j \sum_{k=1}^{n} \mathcal{C}_{ijk} \mathcal{R}_k \text{ (Equation (40))} \\
&= \sum_{k=1}^{n} \left(\sum_{i,j=1}^{n} \mathbf{u}_i \mathbf{v}_j \mathcal{C}_{ijk}\right) \mathcal{R}_k \\
&= \rho\left(\sum_{k=1}^{n} \left(\sum_{i,j=1}^{n} \mathbf{u}_i \mathbf{v}_j \mathcal{C}_{ijk}\right) \mathbf{a}^{(k)}\right) \text{ by the definition of } \rho \\
&= \rho\left(\mathbf{u} \cdot \mathbf{v}\right) \text{ (Equation (13))}
\end{aligned}
\tag{43}
$$

Therefore, $\rho$ is multiplicative. Since it is a $\mathbb{F}$-linear map, this proves that it is a homomorphism of $\mathbb{F}$-algebras. $\quad\square$

From this proposition, it follows that we can find a solution $\mathcal{R} \in \mathbb{F}^{n \times m \times m}$ to

$$P_{i,j}(\mathcal{R}) := \mathcal{R}_i \mathcal{R}_j - \sum_{k=1}^{n} \mathcal{C}_{ijk}^{(B)} \mathcal{R}_k = 0 \ \forall i, j \in [n]$$

$$\sum_{i=1}^{n} \lambda_i^{(B)} \mathcal{R}_i = \mathbb{I}_m \text{ if } \mathfrak{A} \text{ is unital with } 1_{\mathfrak{A}} = \sum_{i=1}^{n} \lambda_i^{(B)} \mathbf{a}^{(i)} \tag{44}$$

in order to obtain a representation of size $m$ of $\mathfrak{A}$ over $\mathbb{F}$, or more generally over any field into which $\mathbb{F}$ embeds. In particular, even if $\mathbb{F} = \mathbb{Z}/p\mathbb{Z}$, one may regard its elements as integer representatives and solve the same equations over $\mathbb{R}$ or $\mathbb{C}$, so that real (or complex) matrix representations remain valid.

It should be noted that if we switch from the basis $B$ to another basis $\tilde{B}$, then a solution $\mathcal{R}$ in $B$ to the above system of equations becomes $\mathcal{R} \times_1 \mathbf{P}^{\top}$ in $\tilde{B}$, with $\mathbf{P} \in \mathbb{F}^{n \times n}$ the basis change matrix from $B$ to $\tilde{B}$.

**Proposition B.17.** *Let $\mathcal{C}$ and $\tilde{\mathcal{C}}$ be the structure tensors of an $n\mathbb{F}$-FDA $\mathfrak{A}$ with respect to two bases $B = \{\mathbf{a}^{(i)}\}_{i \in [n]}$ and $\tilde{B} = \{\tilde{\mathbf{a}}^{(i)}\}_{i \in [n]}$ respectively. Let $\mathbf{P} \in \mathbb{F}^{n \times n}$ be the basis change matrix from $B$ to $\tilde{B}$, i.e. $\tilde{\mathbf{a}}^{(i)} = \sum_{k=1}^{n} \mathbf{P}_{ki}\mathbf{a}^{(k)} \ \forall i \in [n]$. For all $\mathcal{R} \in \mathbb{F}^{n \times m \times m}$, by defining $\tilde{\mathcal{R}} := \mathcal{R} \times_1 \mathbf{P}^{\top}$, which is equivalent to $\mathcal{R} = \tilde{\mathcal{R}} \times_1 \mathbf{P}^{-1^{\top}}$, we have:*

- $\mathcal{R}_i \mathcal{R}_j = \sum_{k=1}^{n} \mathcal{C}_{ijk}\mathcal{R}_k \ \forall i,j \in [n] \iff \tilde{\mathcal{R}}_i \tilde{\mathcal{R}}_j = \sum_{k=1}^{n} \tilde{\mathcal{C}}_{ijk}\tilde{\mathcal{R}}_k \ \forall i,j \in [n]$;

- $\sum_{i=1}^{n} \lambda_i \mathcal{R}_i = \mathbb{I}_m \iff \sum_{i=1}^{n} \tilde{\lambda}_i \tilde{\mathcal{R}}_i = \mathbb{I}_m$ *if further $\mathfrak{A}$ is unital with* $1_{\mathfrak{A}} = \sum_{i=1}^{n} \lambda_i \mathbf{a}^{(i)} = \sum_{i=1}^{n} \tilde{\lambda}_i \tilde{\mathbf{a}}^{(i)}$.

*Proof.* Let $\mathcal{R} \in \mathbb{F}^{n \times m \times m}$. Set $\tilde{\mathcal{R}} = \mathcal{R} \times_1 \mathbf{P}^{\top}$. First, note that this is equivalent to $\mathcal{R} = \tilde{\mathcal{R}} \times_1 \mathbf{P}^{-1^{\top}}$ since $(\mathcal{R} \times_1 \mathbf{P}^{\top} \times_1 \mathbf{P}^{-1^{\top}})_{(1)} = \mathbf{P}^{-1^{\top}}(\mathcal{R} \times_1 \mathbf{P}^{\top})_{(1)} = \mathbf{P}^{-1^{\top}}\mathbf{P}^{\top}\mathcal{R}_{(1)} = \mathcal{R}_{(1)}$ and $(\tilde{\mathcal{R}} \times_1 \mathbf{P}^{-1^{\top}} \times_1 \mathbf{P}^{\top})_{(1)} = \mathbf{P}^{\top}\mathbf{P}^{-1^{\top}}\tilde{\mathcal{R}}_{(1)} = \tilde{\mathcal{R}}_{(1)}$ (Equation (8)).

Assume $\mathcal{R}_i \mathcal{R}_j = \sum_{k=1}^{n} \mathcal{C}_{ijk}\mathcal{R}_k \ \forall i,j \in [n]$. Then, for all $i,j \in [n]$, we have:

$$
\begin{aligned}
\tilde{\mathcal{R}}_i \tilde{\mathcal{R}}_j &= \left(\sum_{l=1}^{n} \mathbf{P}_{li}\mathcal{R}_l\right)\left(\sum_{r=1}^{n} \mathbf{P}_{rj}\mathcal{R}_r\right) = \sum_{l=1}^{n}\sum_{r=1}^{n} \mathbf{P}_{li}\mathbf{P}_{rj}\mathcal{R}_l\mathcal{R}_r \\
&= \sum_{l=1}^{n}\sum_{r=1}^{n} \mathbf{P}_{li}\mathbf{P}_{rj}\sum_{s=1}^{n}\mathcal{C}_{lrs}\mathcal{R}_s = \sum_{s=1}^{n}\sum_{l=1}^{n}\sum_{r=1}^{n}\mathcal{C}_{lrs}\mathbf{P}_{li}\mathbf{P}_{rj}\mathcal{R}_s \\
&= \sum_{s=1}^{n}\sum_{l=1}^{n}\sum_{r=1}^{n}\mathcal{C}_{lrs}\mathbf{P}_{li}\mathbf{P}_{rj}\sum_{k=1}^{n}(\mathbf{P}^{-1})_{ks}\tilde{\mathcal{R}}_k \\
&= \sum_{k=1}^{n}\left(\sum_{\ell=1}^{n}\sum_{r=1}^{n}\sum_{s=1}^{n}\mathcal{C}_{lrs}(\mathbf{P}^{\top})_{il}(\mathbf{P}^{\top})_{jr}(\mathbf{P}^{-1})_{ks}\right)\tilde{\mathcal{R}}_k \\
&= \sum_{k=1}^{n}\left(\mathcal{C} \times_1 \mathbf{P}^{\top} \times_2 \mathbf{P}^{\top} \times_3 \mathbf{P}^{-1}\right)_{ijk}\tilde{\mathcal{R}}_k \\
&= \sum_{k=1}^{n}\tilde{\mathcal{C}}_{ijk}\tilde{\mathcal{R}}_k \ \text{(Proposition B.3)}
\end{aligned}
\tag{45}
$$

Further, assume that $\mathfrak{A}$ is unital with $1_{\mathfrak{A}} = \sum_{i=1}^{n} \lambda_i \mathbf{a}^{(i)} = \sum_{i=1}^{n} \tilde{\lambda}_i \tilde{\mathbf{a}}^{(i)}$. We have $\tilde{\lambda} = \mathbf{P}^{-1}\lambda$. So

$$
\begin{aligned}
\sum_{i=1}^{n}\lambda_i\mathcal{R}_i = \mathbb{I}_m &\iff \sum_{i=1}^{n}\lambda_i\sum_{k=1}^{n}(\mathbf{P}^{-1})_{ki}\tilde{\mathcal{R}}_i = \mathbb{I}_m \iff \sum_{k=1}^{n}\left(\sum_{i=1}^{n}(\mathbf{P}^{-1})_{ki}\lambda_i\right)\tilde{\mathcal{R}}_k = \mathbb{I}_m \\
&\iff \sum_{k=1}^{n}\left(\mathbf{P}^{-1}\lambda\right)_k\tilde{\mathcal{R}}_k = \mathbb{I}_m \\
&\iff \sum_{k=1}^{n}\tilde{\lambda}_k\tilde{\mathcal{R}}_k = \mathbb{I}_m
\end{aligned}
\tag{46}
$$

Similarly, assume $\tilde{\mathcal{R}}_i\tilde{\mathcal{R}}_j = \sum_{k=1}^{n}\tilde{\mathcal{C}}_{ijk}\tilde{\mathcal{R}}_k \ \forall i,j \in [n]$. Then, for all $i,j \in [n]$, we have (the derivation steps are similar to those above) $\mathcal{R}_i\mathcal{R}_j = \sum_{k=1}^{n}\left(\tilde{\mathcal{C}} \times_1 \mathbf{P}^{-1^{\top}} \times_2 \mathbf{P}^{-1^{\top}} \times_3 \mathbf{P}\right)_{ijk}\mathcal{R}_k = \sum_{k=1}^{n}\mathcal{C}_{ijk}\mathcal{R}_k$. $\square$

We have a system of $n^2m^2$ (for a non-unital $\mathfrak{A}$) or $(n^2+1)m^2$ (for a unital $\mathfrak{A}$) non-linear equations in $nm^2$ variables, so in general, overdetermined. Solutions form an algebraic variety. For a non-unital $\mathfrak{A}$, the trivial solution is $\mathcal{R} = 0_{n \times m \times m}$, the zero tensor. For a unital $\mathfrak{A}$, there is no trivial solution. **It is worth asking what is the smallest $m$ (or the values of $m$) for which this system admits a non-trivial solution.**, i.e., a solution to

$$
P(\mathcal{R}) = \sum_{i,j}\|P_{i,j}(\mathcal{R})\|_{\mathrm{F}}^2 = \sum_{i,j}\sum_{l,r}\left(P_{i,j}^{l,r}(\mathcal{R})\right)^2 = 0 \qquad \text{subject to} \qquad \sum_{i=1}^{n}\|\mathcal{R}_i\|_{\mathrm{F}}^2 \neq 0
$$

$$
\sum_{i=1}^{n}\lambda_i^{(B)}\mathcal{R}_i = \mathbb{I}_m \text{ if } \mathfrak{A} \text{ is unital with } 1_{\mathfrak{A}} = \sum_{i=1}^{n}\lambda_i^{(B)}\mathbf{a}^{(i)}
\tag{47}
$$

For associatives $n\mathbb{F}$-FDA, the smallest $m$ is less than or equal to $n$, since the tensor $\mathcal{R} \in \mathbb{F}^{n \times n \times n}$ defined by $\mathcal{R}_i = \mathcal{C}_i^\top \forall i \in [n]$ is solution to (44) (Proposition B.18). For non-unital FDA, if there is a solution for a certain $m$, then there is one for all $p \geq m$. In fact, if $\mathcal{R} \in \mathbb{F}^{n \times m \times m}$ is solution to Equation (44), then for all $\mathbf{A} \in \mathbb{F}^{p \times m}$ and $\mathbf{B} \in \mathbb{F}^{m \times p}$ such that $\mathbf{BA} = \mathbb{I}_m$, $\tilde{\mathcal{R}} = [\mathbf{A}\mathcal{R}_i\mathbf{B}]_{i \in [n]} \in \mathbb{F}^{n \times p \times p}$ is also solution to Equation (44) (Proposition B.19). But this is false for unital FDA in general. If $\sum_{i=1}^n \lambda_i \mathcal{R}_i = \mathbb{I}_m$, then we have $\sum_{i=1}^n \lambda_i \mathbf{A}\mathcal{R}_i\mathbf{B} = \mathbf{A}\left(\sum_{i=1}^n \lambda_i \mathcal{R}_i\right)\mathbf{B} = \mathbf{AB}$. For this to equals $\mathbb{I}_p$, we need $p = m$. Describing the solution set of (44) in full generality is challenging. A basic invariance is simultaneous conjugation: if $\mathcal{R} = [\mathcal{R}_1, \ldots, \mathcal{R}_n]$ solves (44) for all $i, j$, then so does $[\mathbf{S}\mathcal{R}_1\mathbf{S}^{-1}, \ldots, \mathbf{S}\mathcal{R}_n\mathbf{S}^{-1}]$ for any $\mathbf{S} \in \mathrm{GL}_m(\mathbb{F})$ (Corollary B.20).

**Proposition B.18** (Left-regular representation)**.** *If* $(\mathfrak{A}, \cdot)$ *is an associative* $n\mathbb{F}$*-FDA with structure tensor* $\mathcal{C} \in \mathbb{F}^{n \times n \times n}$ *in* $B = \{\mathbf{a}^{(i)}\}_{i \in [n]}$, *then the tensor* $\mathcal{R} \in \mathbb{F}^{n \times n \times n}$ *defined by* $\mathcal{R}_i := \mathcal{C}_i^\top \forall i \in [n]$ *satisfies*

$$\mathcal{R}_i \mathcal{R}_j = \sum_{k=1}^n \mathcal{C}_{ijk} \mathcal{R}_k = 0 \; \forall i, j \in [n], \quad \text{and } \sum_{i=1}^n \lambda_i \mathcal{R}_i = \mathbb{I}_n \text{ if } \mathfrak{A} \text{ is unital with } 1_\mathfrak{A} = \sum_{i=1}^n \lambda_i \mathbf{a}^{(i)} \tag{48}$$

*Proof.* Assume $(\mathfrak{A}, \cdot)$ is an associative $n\mathbb{F}$-FDA. That is (Proposition B.5)

$$\sum_{k=1}^n \mathcal{C}_{ijk}\mathcal{C}_{klm} = \sum_{k=1}^n \mathcal{C}_{ikm}\mathcal{C}_{jlk} \; \forall i, j, l, m \in [n] \iff \sum_{k=1}^n \mathcal{C}_{ijk}(\mathcal{C}_k^\top)_{ml} = (\mathcal{C}_i^\top \mathcal{C}_j^\top)_{ml} \; \forall i, j, l, m \in [n]$$

$$\iff \sum_{k=1}^n \mathcal{C}_{ijk}\mathcal{C}_k^\top = \mathcal{C}_i^\top \mathcal{C}_j^\top \; \forall i, j \in [n] \tag{49}$$

$$\iff \sum_{k=1}^n \mathcal{C}_{ijk}\mathcal{R}_k = \mathcal{R}_i \mathcal{R}_j \; \forall i, j \in [n]$$

If further $\mathfrak{A}$ is unital, then $\mathcal{C} \times_1 \lambda = \sum_{i=1}^n \lambda_i \mathcal{C}_i = \mathbb{I}_n$ (Proposition B.5), which is equivalent to $\sum_{i=1}^n \lambda_i \mathcal{R}_i = \mathbb{I}_n$. $\square$

**Proposition B.19** (Invariance of the solution set)**.** *Let* $\mathcal{C} \in \mathbb{F}^{n \times n \times n}$*. For all* $\mathcal{R} \in \mathbb{F}^{n \times m \times m}$*, if* $\mathcal{R}_i \mathcal{R}_j = \sum_{k=1}^n \mathcal{C}_{ijk}\mathcal{R}_k \; \forall i, j \in [n]$*, then* $(\mathbf{A}\mathcal{R}_i\mathbf{B})(\mathbf{A}\mathcal{R}_j\mathbf{B}) = \sum_{k=1}^n \mathcal{C}_{ijk}(\mathbf{A}\mathcal{R}_k\mathbf{B}) \; \forall i, j \in [n]$ *for all for all* $\mathbf{A} \in \mathbb{F}^{p \times m}$ *and* $\mathbf{B} \in \mathbb{F}^{m \times p}$ *such that* $\mathbf{BA} = \mathbb{I}_m$*.*

**Corollary B.20** (Conjugacy invariance of the solution set)**.** *Let* $\mathcal{C} \in \mathbb{F}^{n \times n \times n}$*. For all* $\mathcal{R} \in \mathbb{F}^{n \times m \times m}$*, if* $\mathcal{R}_i \mathcal{R}_j = \sum_{k=1}^n \mathcal{C}_{ijk}\mathcal{R}_k \; \forall i, j \in [n]$*, then* $(\mathbf{S}\mathcal{R}_i\mathbf{S}^{-1})(\mathbf{S}\mathcal{R}_j\mathbf{S}^{-1}) = \sum_{k=1}^n \mathcal{C}_{ijk}(\mathbf{S}\mathcal{R}_k\mathbf{S}^{-1}) \; \forall i, j \in [n]$ *for all* $\mathbf{S} \in \mathrm{GL}_m(\mathbb{F})$*. If further* $\sum_{i=1}^n \lambda_i \mathcal{R}_i = \mathbb{I}_m$*, then* $\sum_{i=1}^n \lambda_i \mathbf{S}\mathcal{R}_i\mathbf{S}^{-1} = \mathbb{I}_m$*.*

It is therefore natural to work up to the equivalence relation defined on $\mathbb{F}^{n \times m \times m}$ by the following equation, i.e., to consider orbits under simultaneous conjugation.

$$\mathcal{R} \sim \tilde{\mathcal{R}} \iff \exists \mathbf{S} \in \mathrm{GL}_m(\mathbb{F}) \text{ such that } \tilde{\mathcal{R}}_i = \mathbf{S}\mathcal{R}_i\mathbf{S}^{-1} \; \forall i \in [n] \tag{50}$$

One might hope this allows choosing a diagonal normal form. However, this would require a single change of basis that diagonalizes all $\mathcal{R}_i$ at once. In general the $\mathcal{R}_i$ need not commute or be simultaneously diagonalizable, so this reduction is not available without additional assumptions. Nevertheless, let us imagine we restrict ourselves to diagonal solutions. In that case, writing $\mathcal{R}_i = \mathrm{diag}(\mathbf{r}_i^{(1)}, \ldots, \mathbf{r}_i^{(m)})$ for all $i \in [n]$, the defining system (Equation (44)) decouples coordinate-wise. For each $t \in [m]$, the scalars $(\mathbf{r}_1^{(t)}, \ldots, \mathbf{r}_n^{(t)})$ must satisfy

$$\mathbf{r}_i^{(t)}\mathbf{r}_j^{(t)} = \sum_{k=1}^n \mathcal{C}_{ijk}\mathbf{r}_k^{(t)} \quad \forall i, j \in [n], \quad \sum_{i=1}^n \lambda_i \mathbf{r}_i^{(t)} = 1 \text{ if } \mathfrak{A} \text{ is unital} \tag{51}$$

Thus, the diagonal ansatz reduces the matrix problem to a family of scalar problems, one for each diagonal entry. A diagonal solution of size $m$ is obtained by picking $m$ points in

$$\mathcal{S}(\mathcal{C}) := \left\{ \mathbf{r} \in \mathbb{F}^n : \mathbf{r}_i\mathbf{r}_j = \sum_k \mathcal{C}_{ijk}\mathbf{r}_k \; \forall (i, j) \in [n]^2 \text{ and } \sum_i \lambda_i \mathbf{r}_i = 1 \text{ if unitality holds} \right\} \tag{52}$$

and placing them on the diagonal (conjugation by permutation matrices only reorders the diagonal). Thus, diagonal solutions correspond to $\mathcal{S}(\mathcal{C})^m / S_m$ [6]. In particular, a nontrivial diagonal solution exists if and only if $\mathcal{S}(\mathcal{C}) \setminus \{0\} \neq \emptyset$. Hence, the minimal matrix size for a diagonal solution is $m = 1$ whenever $\mathcal{S}(\mathcal{C})$ contains a nonzero point; otherwise, no diagonal solution exists for any $m$.

While the defining equations for representations are explicit (see (44) and the diagonal reduction (51)), a general description of their solution set (e.g., orbit structure, invariances, dimensions, and stratification) would require tools from algebraic geometry, invariant theory, or representation theory. For this reason, we now provide a few examples of representation for the algebras of Section B.3 and proceed to the next section, which addresses the problem of learning algebras using deep learning models.

*Example* B.12 (Complex Numbers). For the algebra of complex numbers (Example B.3) with $\mathbb{F} = \mathbb{R}$ and $\{\mathbf{a}^{(1)}, \mathbf{a}^{(2)}\} = \{1, \mathbf{i}\}$, the relations are

$$\mathcal{R}_1^2 = \mathcal{R}_1, \ \mathcal{R}_1 \mathcal{R}_2 = \mathcal{R}_2 \mathcal{R}_1 = \mathcal{R}_2, \ \mathcal{R}_2^2 = -\mathcal{R}_1, \quad \mathcal{R}_1 = \mathbb{I}_m \text{ since } 1_{\mathfrak{A}} = \mathbf{a}^{(1)} \tag{53}$$

Therefore $\mathcal{R}_1 = \mathbb{I}_m$ (representation of multiplication by 1 in $\mathbb{C}$ for $m = 2$) and $\mathcal{R}_2^2 = -\mathbb{I}_m$. This requires $m$ to be even, because over $\mathbb{R}$ there is no odd-dimensional $m \times m$ real matrix squaring to $-\mathbb{I}_m$. The minimal case is $m = 2$ with $\mathcal{R}_2 = \mathcal{C}_2^\top = [[0 \ -1], [1 \ 0]] = -\mathcal{C}_2$ (Equation (2)), which is the standard real representation of multiplication by $\mathbf{i}$. Writing $m = 2k$, the general solution is $\mathcal{R}_2 = \mathbf{S}(\mathbb{I}_k \otimes \mathcal{C}_2^\top)\mathbf{S}^{-1}$ for an arbitrary invertible real matrix $\mathbf{S} \in \mathrm{GL}_m(\mathbb{R})$. Equivalently, $\mathcal{R}_2$ is any real matrix with minimal polynomial $x^2 + 1$. If one requires $\mathcal{R}_2$ to be orthogonal, then $\mathcal{R}_2 = \mathbf{Q}(\mathbb{I}_k \otimes \mathcal{C}_2^\top)\mathbf{Q}^\top$ for any $\mathbf{Q} \in O(m)$, with $O(m)$ the general orthogonal group.

With $\mathbb{F} = \mathbb{Z}/p\mathbb{Z}$ for $p$ prime, we have $\mathcal{C}_{221} = (-1)\%p = p - 1$, so the relation $\mathcal{R}_2^2 = -\mathcal{R}_1 = -\mathbb{I}_m$ over $\mathbb{R}$ becomes $\mathcal{R}_2^2 = (p-1)\mathbb{I}_m$. Since $p - 1 > 0$, the polynomial $x^2 - (p-1) = (x - \sqrt{p-1})(x + \sqrt{p-1})$ has distinct real roots, hence any real solution $\mathcal{R}_2$ is diagonalizable over $\mathbb{R}$ with eigenvalues in $\{\pm\sqrt{p-1}\}$. Equivalently, $\mathcal{R}_2 = (p-1)^{1/2}\mathbf{J}$ with $\mathbf{J}^2 = \mathbb{I}_m$ i.e., $\mathcal{R}_2$ is $\sqrt{p-1}$ times an involution. There is no parity restriction on $m$ (unlike the $\mathbb{F} = \mathbb{R}$ case). A convenient normal form is $\mathcal{R}_2 = (p-1)^{1/2}\mathbf{S} \operatorname{diag}(\boldsymbol{\epsilon})\mathbf{S}^{-1}$ for any $\mathbf{S} \in \mathrm{GL}_m(\mathbb{R})$ and signs $\boldsymbol{\epsilon} \in \{\pm 1\}^m$. If one also requires $\mathcal{R}_2$ to be symmetric (hence orthogonally diagonalizable), then $\mathcal{R}_2 = (p-1)^{1/2}\mathbf{Q}\operatorname{diag}(\boldsymbol{\epsilon})\mathbf{Q}^\top$ for any $\mathbf{Q} \in O(m)$.

*Example* B.13 (Dual Numbers). For the algebra of dual numbers (Example B.4) with $\{\mathbf{a}^{(1)}, \mathbf{a}^{(2)}\} = \{1, \varepsilon\}$, $\mathcal{R}_1 = \mathbb{I}_m$ and $\mathcal{R}_2^2 = 0$ (corresponding to the dual number $\varepsilon$). This yields nontrivial solutions whenever $m \geq 2$, since square-zero (nilpotent) matrices exist, e.g. $\mathcal{R}_2 = \mathbf{u}\mathbf{v}^\top \in \mathbb{F}^{m \times m}$ for any $\mathbf{u}, \mathbf{v} \in \mathbb{F}^m$ such that $\mathbf{u}^\top \mathbf{v} = 0$.

*Example* B.14 (Polynomials modulo $x^n$). $\mathfrak{A} = (\mathbb{F}_{n-1}[x], \cdot)$ with basis $\mathbf{a}^{(i)} = x^i \ \forall i \in [\![0, n-1]\!]$ and product $x^i \cdot x^j = x^{(i+j) \bmod n}$ (Example B.5). The structure constants are $\mathcal{C}_{ijk} = \delta_{k,(i+j)\%n} \ \forall i, j, k \in [\![0, n-1]\!]$, so we have the relations

$$\mathcal{R}_i \mathcal{R}_j = \mathcal{R}_{(i+j)\%n} \ \forall i, j \in [\![0, n-1]\!], \quad \mathcal{R}_0 = \mathbb{I}_m \text{ since } 1_{\mathfrak{A}} = \mathbf{a}^{(0)} \tag{54}$$

Setting $\mathbf{A} := \mathcal{R}_1$, (54) yields $\mathcal{R}_{i+1} = \mathbf{A}\mathcal{R}_i$ and hence $\mathcal{R}_i = \mathbf{A}^i$ for all $i$. In particular, $\mathbf{A}^n = \mathcal{R}_n = \mathcal{R}_0 = \mathbb{I}_m$. Conversely, for any $\mathbf{A} \in \mathcal{M}_m(\mathbb{F})$ with $\mathbf{A}^n = \mathbb{I}_m$, the assignment $\mathcal{R}_i := \mathbf{A}^i \ \forall i \in [\![0, n-1]\!]$ solves (54). In fact, since $\mathbf{A}^n = \mathbb{I}_m$, we have for all $k = qn + r$ with $q \geq 0$ and $r = k\%n \in [\![0, n-1]\!]$ (Euclidean division), $\mathbf{A}^k = \mathbf{A}^{qn+r} = (\mathbf{A}^n)^q\mathbf{A}^r = (\mathbb{I}_m)^q\mathbf{A}^r = \mathbf{A}^r = \mathbf{A}^{k\%n}$. Therefore, size-$m$ solutions are in bijection with matrices $\mathbf{A}$ satisfying $\mathbf{A}^n = \mathbb{I}_m$, via $\mathbf{A} \mapsto (\mathcal{R}_i = \mathbf{A}^i)_{i=0}^{n-1}$.

Over $\mathbb{R}$, a faithful (i.e. injective) size-$n$ representation is obtained by choosing $\mathbf{A}$ to be the $n \times n$ cyclic shift matrix

$$\mathbf{A} = \begin{pmatrix} 0 & 0 & \cdots & 0 & 1 \\ 1 & 0 & \cdots & 0 & 0 \\ 0 & 1 & \ddots & \vdots & \vdots \\ \vdots & \ddots & \ddots & 0 & 0 \\ 0 & \cdots & 0 & 1 & 0 \end{pmatrix} \tag{55}$$

Since $\mathbf{A}\mathbf{e}^{(k)} = \mathbf{e}^{(k+1)} \ \forall k \in [n-1]$ and $\mathbf{A}\mathbf{e}^{(n)} = \mathbf{e}^{(1)}$, we have $\mathbf{A}_n^n\mathbf{e}^{(k)} = \mathbf{e}^{(k)} \ \forall k \in [n]$. Therefore $\mathbf{A}^n = \mathbb{I}_n$. Define $\rho(x^i) = \mathbf{A}^i \ \forall i \in [\![0, n-1]\!]$. Then for all $i, j \in [\![0, n-1]\!]$, $\rho(x^i)\rho(x^j) = \mathbf{A}^i\mathbf{A}^j = \mathbf{A}^{i+j} = \mathbf{A}^{(i+j)\%n} = \rho(x^{(i+j)\%n})$. So

---

[6] The quotient $/S_m$ indicates that two $m$-tuples which differ only by a permutation of their coordinates correspond to the same diagonal solution.

$\rho$ is an algebra homomorphism. This is the regular (cyclic) real representation. More generally, any real solution arises from some $\mathbf{A} \in \mathbb{F}^{m \times m}$ with $\mathbf{A}^n = \mathbb{I}_m$ by $\mathcal{R}_j = \mathbf{A}^j$. Over $\mathbb{R}$ such $\mathbf{A}$ is real-similar to a block diagonal matrix with $1 \times 1$ blocks at $\pm 1$ (when $n$ is even) and $2 \times 2$ rotation blocks corresponding to conjugate pairs $e^{\pm 2\pi i k/n}$ (Equation (34)). If fact, if $\mathbf{A} \in \mathbb{R}^{m \times m}$ satisfies $\mathbf{A}^n = \mathbb{I}_m$, then every eigenvalue $\lambda$ of $\mathbf{A}$ solves $\lambda^n = 1$, i.e. $\lambda = e^{2\pi i k/n}$ for some $k \in [\![0, n-1]\!]$. The real roots among these are $\lambda = 1$ (always) and, when $n$ is even, $\lambda = -1$; these give $1 \times 1$ Jordan blocks [1] and (if $n$ is even) $[-1]$. All remaining roots occur in complex-conjugate pairs $e^{\pm i\theta}$ with $\theta = 2\pi k/n$. Over $\mathbb{R}$, each such conjugate pair corresponds to a $2 \times 2$ real block that acts as a rotation by angle $\theta$.

*Example* B.15 (Square matrices of size $n$). $\mathfrak{A} = (\mathcal{M}_n(\mathbb{F}), \cdot)$, with canonical matrix unit basis $B = \{\mathbf{a}^{((q-1)n+r)}\}_{(q,r) \in [n]^2}$, where each $\mathbf{a}^{((q-1)n+r)}$ has $1_{\mathbb{F}}$ at position $(q, r) \in [n]^2$ and $0_{\mathbb{F}}$ everywhere (Example B.6). Since $\mathbf{a}^{((q_1-1)n+r_1)} \cdot \mathbf{a}^{((q_2-1)n+r_2)} = \delta_{q_2 r_1} \mathbf{a}^{((q_1-1)n+r_2)}$, the structure constants of $\mathfrak{A}$ in $B$ are $\mathcal{C}_{(q_1-1)n+r_1,(q_2-1)n+r_2,(q_3-1)n+r_3} = \delta_{q_2 r_1} \delta_{q_1 q_3} \delta_{r_2 r_3}$. The representation equations, therefore, read

$$\mathcal{R}_{(q_1-1)n+r_1} \mathcal{R}_{(q_2-1)n+r_2} = \sum_{q,r=1}^{n} \delta_{q_2 r_1} \delta_{q_1 q} \delta_{r_2 r} \mathcal{R}_{(q-1)n+r} = \delta_{q_2 r_1} \mathcal{R}_{(q_1-1)n+r_2} \qquad \forall q_1, r_1, q_2, r_2 \in [n]$$

$$\sum_{q,r=1}^{n} \delta_{qr} \mathcal{R}_{(q-1)n+r} = \sum_{i=1}^{n} \mathcal{R}_{(i-1)n+i} = \mathbb{I}_m \text{ since } 1_{\mathfrak{A}} = \mathbb{I}_n = \sum_{q,r=1}^{n} \delta_{qr} \mathbf{a}^{((q-1)n+r)}$$

(56)

Define $m = n$ and set, for each $(q, r) \in [n]^2$, $\mathcal{R}_{(q-1)n+r} = \mathbf{a}^{((q-1)n+r)} \in \mathbb{R}^{n \times n}$. Then

$$\mathcal{R}_{(q_1-1)n+r_1} \mathcal{R}_{(q_2-1)n+r_2} = \delta_{q_2 r_1} \mathbf{a}^{((q_1-1)n+r_2)} = \delta_{q_2 r_1} \mathcal{R}_{(q_1-1)n+r_2} \; \forall q_1, r_1, q_2, r_2 \in [n]$$

$$\sum_{i=1}^{n} \mathcal{R}_{(i-1)n+i} = \sum_{i=1}^{n} \mathbf{a}^{((i-1)n+i)} = \mathbb{I}_n$$

(57)

So (56) holds. This gives the standard left action $\rho : \mathcal{M}_n(\mathbb{F}) \to \mathcal{M}_n(\mathbb{F})$, $\rho(\mathbf{A}) = \mathbf{A}$. For a general size $m = \kappa n$ with $\kappa \in \mathbb{N}$, set $\mathcal{R}_{(q-1)n+r} = \mathbf{a}^{((q-1)n+r)} \otimes \mathbb{I}_\kappa \in \mathbb{F}^{\kappa n \times \kappa n}$. We have

$$\mathcal{R}_{(q_1-1)n+r_1} \mathcal{R}_{(q_2-1)n+r_2} = \left( \mathbf{a}^{((q_1-1)n+r_1)} \mathbf{a}^{((q_2-1)n+r_2)} \right) \otimes \mathbb{I}_\kappa = \delta_{q_2 r_1} \mathbf{a}^{((q_1-1)n+r_2)} \otimes \mathbb{I}_\kappa = \delta_{q_2 r_1} \mathcal{R}_{(q_1-1)n+r_2}$$

$$\sum_{i=1}^{n} \mathcal{R}_{(i-1)n+i} = \sum_{i=1}^{n} \mathbf{a}^{((i-1)n+i)} \otimes \mathbb{I}_\kappa = \mathbb{I}_n \otimes \mathbb{I}_\kappa = \mathbb{I}_{\kappa n}$$

(58)

Thus (56) holds with $m = \kappa n$. By conjugacy invariance (Corollary B.20), the family $\mathcal{R}_{(q-1)n+r} := \mathbf{S}(\mathbf{a}^{((q-1)n+r)} \otimes \mathbb{I}_\kappa) \mathbf{S}^{-1}$ is again a solution for any $\mathbf{S} \in \mathrm{GL}_{\kappa n}(\mathbb{F})$. In this way, one obtains all size-$m = \kappa n$ solutions up to simultaneous conjugation.

*Example* B.16 (Square matrices of size $n$ with the commutator). If we replace matrix multiplication $\mathbf{AB}$ in $\mathfrak{A} = (\mathcal{M}_n(\mathbb{F}), \cdot)$ with the matrix commutator $[\mathbf{A}, \mathbf{B}] = \mathbf{AB} - \mathbf{BA}$, then $\mathcal{C}_{(q_1-1)n+r_1,(q_2-1)n+r_2,(q_3-1)n+r_3} = \delta_{q_2 r_1} \delta_{q_1 q_3} \delta_{r_2 r_3} - \delta_{q_1 r_2} \delta_{q_2 q_3} \delta_{r_1 r_3}$ (Example B.8). The representation equations, therefore, read

$$\mathcal{R}_{(q_1-1)n+r_1} \mathcal{R}_{(q_2-1)n+r_2} = \delta_{q_2 r_1} \mathcal{R}_{(q_1-1)n+r_2} - \delta_{q_1 r_2} \mathcal{R}_{(q_2-1)n+r_1} \; \forall q_1, r_1, q_2, r_2 \in [n] \tag{59}$$

Fix $m \geq 2$ and choose any nonzero matrix $\mathbf{N} \in \mathbb{F}^{m \times m}$ with $\mathbf{N}^2 = 0$. Define $\mathcal{R}_{(q-1)n+r} = \delta_{qr} \mathbf{N} \; \forall (q, r) \in [n]^2$. For all $q_1, r_1, q_2, r_2 \in [n]$,

$$\mathcal{R}_{(q_1-1)n+r_1} \mathcal{R}_{(q_2-1)n+r_2} = (\delta_{q_1 r_1} \mathbf{N})(\delta_{q_2 r_2} \mathbf{N}) = \delta_{q_1 r_1} \delta_{q_2 r_2} \mathbf{N}^2 = 0 \tag{60}$$

and

$$\delta_{q_2 r_1} \mathcal{R}_{(q_1-1)n+r_2} - \delta_{q_1 r_2} \mathcal{R}_{(q_2-1)n+r_1} = \delta_{q_2 r_1} \delta_{q_1 r_2} \mathbf{N} - \delta_{q_1 r_2} \delta_{q_2 r_1} \mathbf{N} = 0 \tag{61}$$

So $\mathcal{R}$ satisfies the system (59). This yields nontrivial solutions whenever $m \geq 2$, since square-zero (nilpotent) matrices exist, e.g. $\mathbf{N} = \mathbf{uv}^\top \in \mathbb{F}^{m \times m}$ for any $\mathbf{u}, \mathbf{v} \in \mathbb{F}^m$ such that $\mathbf{u}^\top \mathbf{v} = 0$. Any simultaneous conjugate $\mathbf{S} \mathcal{R}_{(q-1)n+r} \mathbf{S}^{-1}$ for $\mathbf{S} \in \mathrm{GL}_m(\mathbb{F})$ is again a solution, so replacing $\mathbf{N}$ by any conjugate $\mathbf{S N S}^{-1}$ with $(\mathbf{S N S}^{-1})^2 = 0$ also works. Note that for all $\mathbf{S} \in \mathrm{GL}_m(\mathbb{F})$, we have $\mathbf{N}^2 = 0 \iff (\mathbf{S N S}^{-1})^2 = 0$.

If $m = 1$ over a field of characteristic $\neq 2$ (e.g. $\mathbb{R}$), the only solution is trivial: the equations force all scalars to be 0. Hence, the minimal $m$ for a nontrivial solution is $m = 2$.

**Proposition B.21.** *Let $\rho_G : G \to \mathrm{GL}_m(\mathbb{F})$ be a representation of a goup $(G, \circ)$. Define the linear map*

$$\rho : \mathbb{F}[G] \to \mathcal{M}_m(\mathbb{F}), \ \rho\left(\sum_{i=1}^n \alpha_i g_i\right) := \sum_{i=1}^n \alpha_i \, \rho_G(g_i) \tag{62}$$

*Then $\rho$ is an $\mathbb{F}$-algebra homomorphism, and hence a representation of $(\mathbb{F}[G], \cdot)$.*

*Proof.* $\rho$ is linear by definition. For all $\mathbf{u} = \sum_i \mathbf{u}_i g_i$ and $\mathbf{v} = \sum_j \mathbf{v}_j g_j$, $\rho(\mathbf{u} \cdot \mathbf{v}) = \rho\left(\sum_{i,j} \mathbf{u}_i \mathbf{v}_j (g_i \circ g_j)\right) = \sum_{i,j} \mathbf{u}_i \mathbf{v}_j \rho_G(g_i \circ g_j) = \sum_{i,j} \mathbf{u}_i \mathbf{v}_j \rho_G(g_i) \rho_G(g_j) = \rho(\mathbf{u})\rho(\mathbf{v})$. Finaly, $\rho(1_{\mathbb{F}[G]}) = \rho(e) = \mathbb{I}_m$ since $\rho_G(e) = \mathbb{I}_m$. $\square$

*Example* B.17 (Group algebra of the symmetric group $S_t$ over $\mathbb{F}$). Write $n = t!$. The structure tensor of $\mathfrak{A} = (\mathbb{F}[S_t], \cdot)$ in its canonical basis $(\mathbf{a}^{(1)}, \ldots, \mathbf{a}^{(n)}) \equiv (\sigma)_{\sigma \in S_t}$ is $\boldsymbol{\mathcal{C}}_{ijk} = \mathbb{1}(\mathbf{a}^{(k)} = \mathbf{a}^{(i)} \circ \mathbf{a}^{(j)}) \ \forall i, j, k \in [n]$ (Example B.11), so the representation equations read (by abuse of notations)

$$\boldsymbol{\mathcal{R}}_{\sigma_i} \boldsymbol{\mathcal{R}}_{\sigma_j} = \boldsymbol{\mathcal{R}}_{\sigma_i \circ \sigma_j} \ \forall \sigma_i, \sigma_j \in S_t \tag{63}$$

Take $m = t$ and define $\boldsymbol{\mathcal{R}}_\sigma = \mathbf{P}_\sigma \in \mathbb{F}^{m \times m} \ \forall \sigma \in S_t$, where $\mathbf{P}_\sigma$ is the permutation matrix associated to $\sigma$, i.e. $(\mathbf{P}_\sigma)_{ab} = 1$ if $\sigma(b) = a$ and 0 otherwise (under cycle notation for permutations). Then $\boldsymbol{\mathcal{R}}_{\sigma_i} \boldsymbol{\mathcal{R}}_{\sigma_j} = \mathbf{P}_{\sigma_i} \mathbf{P}_{\sigma_j} = \mathbf{P}_{\sigma_i \circ \sigma_j} = \boldsymbol{\mathcal{R}}_{\sigma_i \circ \sigma_j}$, so (63) is satisfied. This is the standard permutation representation of $S_t$ on $\mathbb{F}^t$. More generally, let $m = n$ and for each $\sigma \in S_t$ set $\boldsymbol{\mathcal{R}}_\sigma :=$ matrix of left multiplication by $\sigma$ on $\mathbb{F}[S_t]$. Explicitly, $(\boldsymbol{\mathcal{R}}_\sigma)_{\pi, \gamma} = \mathbb{1}(\gamma = \sigma \circ \pi) \ \forall \pi, \gamma \in S_t$. Then $\boldsymbol{\mathcal{R}}_{\sigma_i} \boldsymbol{\mathcal{R}}_{\sigma_j} = \boldsymbol{\mathcal{R}}_{\sigma_i \circ \sigma_j}$, so (63) holds. This is the left regular representation (Proposition B.18). In general, for any $m \geq 1$, any group representation $\rho : S_t \to \mathrm{GL}_m(\mathbb{F})$ gives a solution of (63) by setting $\boldsymbol{\mathcal{R}}_\sigma := \rho(\sigma)$ (Proposition B.21). Conversely, any solution of (63) defines a group representation of $S_t$. Thus, the solutions of the FDA system for $\mathbb{F}[S_t]$ are in bijection with group representations of $S_t$.

## C. Learning Finite-Dimensional Algebra

### C.1. A Linear Inverse View for $\mathbb{F} = \mathbb{R}$

In $\mathbb{F} = \mathbb{R}$, the problem of learning a FDA with structure tensor $\boldsymbol{\mathcal{C}}^* \in \mathbb{F}^{n \times n \times n}$ is similar to a matrix factorization problem with matrix $\boldsymbol{\mathcal{C}}^{*\top}_{(3)} \in \mathbb{F}^{n^2 \times n}$ (Equation (4)). Giving the measures $\mathbf{U}, \mathbf{V} \in \mathbb{F}^{N \times n}$, let $\mathbf{X} := \mathbf{V} \bullet \mathbf{U} = [-\mathbf{V}_s \otimes \mathbf{U}_s -]_{s \in [N]} \in \mathbb{F}^{N \times n^2}$. The training loss is

$$\mathcal{L}(\boldsymbol{\mathcal{C}}) = \sum_{s \in [N]} \|(\boldsymbol{\mathcal{C}} - \boldsymbol{\mathcal{C}}^*) \times_1 \mathbf{U}_s \times_2 \mathbf{V}_s\|_2^2 = \sum_{s \in [N]} \left\|\left(\boldsymbol{\mathcal{C}}_{(3)} - \boldsymbol{\mathcal{C}}^*_{(3)}\right) \mathbf{X}_s\right\|_2^2 = \left\|\mathbf{X}\left(\boldsymbol{\mathcal{C}}_{(3)} - \boldsymbol{\mathcal{C}}^*_{(3)}\right)^\top\right\|_F^2 \tag{64}$$

By writing $\boldsymbol{\mathcal{C}}^* = [\![\mathbf{A}^*, \mathbf{B}^*, \mathbf{C}^*]\!] := \sum_{\ell=1}^R \mathbf{A}^*_{:,\ell} \circ \mathbf{B}^*_{:,\ell} \circ \mathbf{C}^*_{:,\ell}$ as the CP decomposition of rank $R$ of $\boldsymbol{\mathcal{C}}^*$, we can use a parametrization $\boldsymbol{\mathcal{C}} = [\![\mathbf{A}, \mathbf{B}, \mathbf{C}]\!]$, so that $\boldsymbol{\mathcal{C}}_{(3)} = \mathbf{C}(\mathbf{B} \star \mathbf{A})^\top$. In that case, if we are interested in the global structure of $\boldsymbol{\mathcal{C}}^*$, we can try to evaluate the effect of the properties of the algebra $\mathfrak{A}$ generated by $\boldsymbol{\mathcal{C}}^*$ on the CP rank $R$ of $\boldsymbol{\mathcal{C}}^*$, and thus classify which properties of this algebra increase its CP rank (and therefore make it more difficult to learn). Note that $\boldsymbol{\mathcal{C}} = [\![\mathbf{A}, \mathbf{B}, \mathbf{C}]\!]$ implies $\boldsymbol{\mathcal{C}}_{(3)} = \mathbf{C}(\mathbf{B} \star \mathbf{A})^\top$ and $\boldsymbol{\mathcal{C}} \times_1 \mathbf{u} \times_2 \mathbf{v} = \mathbf{C}\left((\mathbf{A}^\top \mathbf{u}) \odot (\mathbf{B}^\top \mathbf{v})\right) \ \forall \mathbf{u}, \mathbf{v} \in \mathbb{F}^n$, so that $\mathbf{X}\boldsymbol{\mathcal{C}}_{(3)}^\top = (\mathbf{V} \bullet \mathbf{U})(\mathbf{B} \star \mathbf{A})\mathbf{C}^\top = ((\mathbf{VB}) \odot (\mathbf{UA}))\mathbf{C}^\top$. We recall that $\odot$ is the Hadamard product, $\star$ the Khatri-Rao product, $\bullet$ the face-splitting product, and $\circ$ the outer product.

Another way to make things simple, is to just analyze the rank $r$ of $\boldsymbol{\mathcal{C}}^{*\top}_{(3)}$ to characterize the solvability of the problem using a parameterization $\boldsymbol{\mathcal{C}}_{(3)}^\top = \mathbf{W}^{(L)} \cdots \mathbf{W}^{(1)}$, with $\mathbf{W}^{(L)} \in \mathbb{R}^{n^2 \times d}$, $\mathbf{W}^{(i)} \in \mathbb{R}^{d \times d}$ for $1 < i < L$, and $\mathbf{W}^{(1)} \in \mathbb{R}^{d \times n}$. This corresponds to a linear network with $L$ layers. That said, if we are interested in the global structure of $\boldsymbol{\mathcal{C}}^*$, we can try to evaluate the effect of the properties of the algebra $\mathfrak{A}$ generated by $\boldsymbol{\mathcal{C}}^*$ on the rank $r$ of $\boldsymbol{\mathcal{C}}^{*\top}_{(3)}$, and thus classify which properties of this algebra increase its rank (and therefore make it more difficult to learn). For example, if $\boldsymbol{\mathcal{C}}^* = [\![\mathbf{A}^*, \mathbf{B}^*, \mathbf{C}^*]\!]$, then $\boldsymbol{\mathcal{C}}^{*\top}_{(3)} = (\mathbf{B}^* \star \mathbf{A}^*)\mathbf{C}^{*\top}$, so that[7] $\mathrm{rank}(\mathbf{B}^* \star \mathbf{A}^*) + \mathrm{rank}(\mathbf{C}^*) - R \leq r \leq \min(\mathrm{rank}(\mathbf{B}^* \star \mathbf{A}^*), \mathrm{rank}(\mathbf{C}^*))$. By "making things simple," we mean that this is a well-studied problem in the matrix setting. With $L = 1$, there is a need for $\ell_*$ (nuclear norm) regularization (or any other form of appropriate regularization[8]) to recover $\boldsymbol{\mathcal{C}}^{*\top}_{(3)}$ when $N$ is large enough (Candès

---

[7]$\mathrm{rank}(\mathbf{A}) + \mathrm{rank}(\mathbf{B}) - d_2 \leq \mathrm{rank}(\mathbf{AB}) \leq \min(\mathrm{rank}(\mathbf{A}), \mathrm{rank}(\mathbf{B}))$ for all $\mathbf{A} \in \mathbb{R}^{d_1 \times d_2}$ and $\mathbf{B} \in \mathbb{R}^{d_2 \times d_3}$.

[8]For example, if $\boldsymbol{\mathcal{C}}^{*\top}_{(3)}$ is extremely sparse so that the notion of sparsity prevails over the notion of rank, then $\ell_1$ is needed for generalization under gradient descent optimization (Notsawo et al., 2025).

& Tao, 2010; Candes & Recht, 2012; Notsawo et al., 2025). But when $L \geq 2$ (and the initialization scale is small), there is no need for $\ell_*$ (or any other form of regularization) to recover $\mathcal{C}_{(3)}^{*\top}$ (Gunasekar et al., 2017; Arora et al., 2018; 2019; Gidel et al., 2019; Gissin et al., 2019; Razin & Cohen, 2020; Li et al., 2020). Increasing $L$ implicitly biases $\mathcal{C}_{(3)}^{*\top}$ toward a low-rank solution, which oftentimes leads to more accurate recovery for sufficiently large $N$.

The rank is not the only thing to take into account if we treat the problem as a matrix factorization problem. To see this, consider the matrix $\mathbf{e}^{(k)} \mathbf{e}^{(l)\top}$ for $k, l \in [n]$. Even if the rank of this matrix is 1, it has only zeros everywhere except 1 at position $(k, l)$, so we have very little chance of reconstructing it in high dimension by observing a portion of its inputs uniformly at random. The only way to guarantee observation of the input at position $(k, l)$ is to choose measurements coherently with its singular basis $\mathbf{e}^{(k)} \otimes \mathbf{e}^{(l)}$. This idea is formulated more generally below.

**Definition C.1.** Let $U$ be a subspace of $\mathbb{R}^n$ of dimension $r$ and $\mathbf{P}_U$ be the orthogonal projection onto $U$. Then, the coherence of $U$ vis-a-vis a basis $\{\mathbf{u}^{(i)}\}_{i \in [n]}$ is defined by $\mu(U) = \frac{n}{r} \max_i \|\mathbf{P}_U \mathbf{u}^{(i)}\|^2$.

For a matrix $\mathbf{A} = \mathbf{U} \Sigma \mathbf{V}^\top \in \mathbb{R}^{n_1 \times n_2}$ under the compact SVD, the projection on the left singular value is $\mathbf{x} \to \mathbf{U}\mathbf{U}^\top \mathbf{x}$, and $\|\mathbf{U}\mathbf{U}^\top \mathbf{x}\|_2^2 = \|\mathbf{U}^\top \mathbf{x}\|_2^2$ for all $\mathbf{x}$ (similarly for the right singular value). We have the following definition of coherence, which considers each matrix entry.

**Definition C.2** (Local coherence & Leverage score). Let $\mathbf{A} = \mathbf{U}\Sigma\mathbf{V}^\top \in \mathbb{R}^{n_1 \times n_2}$ be the compact SVD of a matrix $\mathbf{A}$ of rank $r$. The local coherences of $\mathbf{A}$ are defined by

$$
\begin{aligned}
\mu_i(\mathbf{A}) &= \frac{n_1}{r} \|\mathbf{U}^\top \mathbf{e}^{(i)}\|^2 = \frac{n_1}{r} \|\mathbf{U}_{i,:}\|^2 \quad \forall i \in [n_1] \\
\nu_j(\mathbf{A}) &= \frac{n_2}{r} \|\mathbf{V}^\top \mathbf{e}^{(j)}\|^2 = \frac{n_2}{r} \|\mathbf{V}_{j,:}\|^2 \quad \forall j \in [n_2]
\end{aligned}
\tag{65}
$$

with $\mu_i$ for row $i$ and $\nu_j$ for row $j$. In $\mathbf{U}^\top \mathbf{e}^{(i)}$, $\mathbf{e}^{(i)} \in \mathbb{R}^{n_1}$, and in $\mathbf{V}^\top \mathbf{e}^{(j)}$, $\mathbf{e}^{(j)} \in \mathbb{R}^{n_2}$. We did not distinguish them explicitly for simplicity's sake.

The analysis of recovery guarantees for matrix factorization hinges on local coherence of a target matrix $\mathbf{A}^*$. The local coherence measures $(\mu_i, \nu_j)_{(i,j) \in [n_1] \times [n_2]}$ of $\mathbf{A}^* \in \mathbb{R}^{n_1 \times n_2}$ quantify how strongly individual rows and columns align with the top singular vectors. These quantities, also known as leverage scores, indicate the "influence" of each row $i$ or column $j$ on the low-rank structure. A row/column with a high leverage score projects strongly onto the span of the singular vectors, meaning that a relatively small number of its entries capture much of the matrix's structure. Uniformly low coherence ($\mu_i$ and $\nu_i$ close to 1) implies that the matrix's information is well-distributed across rows and columns, thereby reducing the number of samples needed for exact recovery. For example, Chen et al. (2014) show in the context of matrix completion that sampling the training inputs at position $(i, j)$ with probability $p_{ij}$ proportional to $\mu_i + \nu_j$ allows for perfect recovery of $\mathbf{A}^*$ with fewer samples than uniform sampling, and called such sampling strategies *local coherence sampling* (see Theorem 3.2 and Corollary 3.3 in (Chen et al., 2014)). The minimal number of observations required for perfect recovery also depends on the coherence measures (Candès & Tao, 2010; Candes & Recht, 2012). Let $\mu^{(0)}$ and $\mu^{(1)}$ be two constants such that $\mu^{(0)} \geq \max(\max_i \mu_i, \max_i \nu_i)$ and $\max_{i,j} [\mathbf{U}^* \mathbf{V}^{*\top}]_{ij} \leq \mu^{(1)} \sqrt{r/(n_1 n_2)}$[9]. Candes & Recht (2012) show that if $\mu_0$ and $\mu_1$ are low, few samples are required to recover $\mathbf{A}^*$. More precisely, put $n = \max(n_1, n_2)$. Suppose we observe $N$ entries of $\mathbf{A}^*$ with locations sampled uniformly at random. There are numerical constants $C$ and $c$ such that if $N \geq C \max\left(\mu_1^2, \mu_0^{\frac{1}{2}} \mu_1, \mu_0 n^{\frac{1}{4}}\right) nr\beta \log(n)$ for some $\beta > 2$, then perfect recovery of $\mathbf{A}^*$ is possible from this $N$ observations with probability at least $1 - c/n^3$. In addition, if $r \leq n^{1/5}/\mu_0$, then the recovery is exact with probability at least $1 - c/n^3$ provided that $N \geq C\mu_0 n^{6/5} r\beta \log(n)$ (see Theorem 1.3 in (Candes & Recht, 2012))

The research question here can be therefore to know which natural properties of $\mathfrak{A}$ affects the rank and the local coherences of $\mathcal{C}_{(3)}^{*\top}$; or more generally, how do the properties of $\mathfrak{A}$ affect the rank and the local coherences of $\mathcal{C}_{(3)}^{*\top}$. We leave these questions for future work.

## C.2. Finite Fields $\mathbb{F} = \mathbb{Z}/p\mathbb{Z}$ and Representation-Centric Modeling

For a finite-field FDA ($\mathbb{F} = \mathbb{F}_p = \mathbb{Z}/p\mathbb{Z}$, $p$ prime), we treat $\mathfrak{A}$ as a vocabulary, index each element $\mathbf{u} \in \mathfrak{A}$ by $\langle \mathbf{u} \rangle \in [q]$ with $q = |\mathfrak{A}| = p^n$, and learn an embedding matrix $\mathbf{E} \in \mathbb{R}^{q \times d}$ so that $\mathbf{E}_{\langle \mathbf{u} \rangle}$ is the trainable vector attached to $\mathbf{u}$. We

---

[9] Since $\left| [\mathbf{U}^* \mathbf{V}^{*\top}]_{ij} \right| = \left| \sum_k \mathbf{U}_{i,k}^* \mathbf{V}_{j,k}^* \right| \leq \sqrt{\sum_k \mathbf{U}_{i,k}^{*2}} \sqrt{\sum_k \mathbf{V}_{j,k}^{*2}} = \|\mathbf{U}_{i,:}^*\|_2 \|\mathbf{V}_{j,:}^*\|_2 = \frac{r}{\sqrt{n_1 n_2}} \sqrt{\mu_i \nu_j} \leq \frac{r}{\sqrt{n_1 n_2}} \mu_0$ for all $i, j$; we can take any $\mu^{(1)} \geq \mu^{(0)} \sqrt{r}$.

hypothesize that grokking corresponds to the point at which $\mathbf{E}$ and the downstream layers collectively represent the algebra, so a simple linear readout recovers the correct output token. To verify this hypothesis: (i) we vary algebraic properties of $\mathbb{F}_p[\mathcal{C}^*]$ and measure their effect on grokking delay and generalization; (ii) we study how structural features of $\mathcal{C}^*$ shape learning dynamics; and (iii) we probe whether models learn algebraic representations during training. This finite-field setting provides the cleanest laboratory for observing how algebraic structure governs grokking.

# D. Experiment Setup

## D.1. Task Description and Model Architecture

From now on we work over the finite field $\mathbb{F} = \mathbb{F}_p$ (prime $p$). Identify $\mathfrak{A} \equiv \mathbb{F}_p^n$ and set $q := |\mathfrak{A}| = p^n$. Each element $\mathbf{u} \in \mathfrak{A}$ is treated as a vocabulary symbol with index $\langle \mathbf{u} \rangle \in [q]$. The models we use will associate to each of these symbols a trainable vector $\mathbf{E}_{\langle \mathbf{u} \rangle} \in \mathbb{R}^d$. $\mathbf{E} \in \mathbb{R}^{V \times d}$, with $V \geq q$ the vocabulary size, which include the set special tokens $\mathcal{S}$ (see section below). Let $\Psi : \mathfrak{A} \to \{0, 1\}^V$ be the one-hot encoding defined the vocabulary $\mathcal{V} = \mathfrak{A} \cup \mathcal{S}$, $\Psi(\mathbf{a}) := [\mathbb{1}(\mathbf{v} = \mathbf{u})]_{\mathbf{v} \in \mathcal{V}}$ $\forall \mathbf{u} \in \mathcal{V}$. We have $\mathbf{E}_{\langle \mathbf{u} \rangle} = \mathbf{E}^\top \Psi(\mathbf{u})$.

### D.1.1. CLASSIFICATION

Here, $V = q + 2$, with 1 for the special tokens $\mathcal{S} = \{\times, =\}$. The logits for $\mathbf{x} = (\mathbf{u}, \times, \mathbf{v}, =)$ with $(\mathbf{u}, \mathbf{v}) \in \mathfrak{A}^2$ are given by $y_\theta(\mathbf{x}) = \varphi\left(\phi\left(\mathbf{E}_{\langle \mathbf{u} \rangle} \oplus \mathbf{E}_{\langle \times \rangle} \oplus \mathbf{E}_{\langle \mathbf{v} \rangle} \oplus \mathbf{E}_{\langle = \rangle}\right)\right) \in \mathbb{R}^q$, where $\oplus$ is the vector concatenation, and $\varphi$ the classifier. We use a linear classifier $\varphi(\mathbf{z}) = \mathbf{b} + \mathbf{W}\mathbf{z}$. For $\phi$, we use MLP, LSTM and Transformer (encoder). The learnable parameters $\theta$ are the union of $\{\mathbf{E}, \mathbf{W}, \mathbf{b}\}$ and the parameters of $\phi$.

For LSTM and Transformer, $\phi$ takes the embeddings $\mathbf{z} \in \mathbb{R}^{4 \times d}$ and returns a hidden representation of size $m^2$, for instance, the representation of the last token = of $\mathbf{x}$. For a MLP $\varphi \circ \phi$ of $L > 1$ layers with activation function $g$, we defined $\phi$ by

$$\phi(\mathbf{z}) = g\left(\mathbf{b}^{(L-1)} + \mathbf{W}^{(L-1)}g\left(\cdots g\left(\mathbf{b}^{(2)} + \mathbf{W}^{(2)}g\left(\mathbf{b}^{(1)} + \mathbf{W}^{(1)}\,\mathrm{vec}(\mathbf{z})\right)\right)\cdots\right)\right) \tag{66}$$

with $\mathbf{z} = \mathrm{vec}\left(\mathbf{E}_{\langle \mathbf{u} \rangle} \oplus \mathbf{E}_{\langle \mathbf{v} \rangle}\right) \in \mathbb{R}^{2d}$, where $\mathbf{W}^{(i)} \in \mathbb{R}^{d_{i+1} \times d_i}$ and $\mathbf{b}^{(i)} \in \mathbb{R}^{d_{i+1}}$ for all $i \in [L-1]$, $(d_1, d_L) = (2d, m^2)$.

The dataset $\mathcal{D} = \{(\mathbf{x}, \mathbf{y}^*(\mathbf{x})) \mid \mathbf{x} = (\mathbf{u}, \times, \mathbf{v}, =), (\mathbf{u}, \mathbf{v}) \in \mathfrak{A}^2\}$ has size $N = q^2$. $\mathcal{D}$ is randomly partitioned into two disjoint and non-empty sets $\mathcal{D}_{\text{train}}$ and $\mathcal{D}_{\text{test}}$, the training and the validation dataset respectively, following a ratio $r := |\mathcal{D}_{\text{train}}|/|\mathcal{D}| \in (0, 1]$. The models are trained to minimize the average cross-entropy loss $\mathcal{L}_{\text{train}}(\theta)$ given in Equation (67). We denoted by $\mathcal{A}_{\text{train}}$ the corresponding accuracy ($\mathcal{L}_{\text{test}}$ and $\mathcal{A}_{\text{test}}$ on $\mathcal{D}_{\text{test}}$).

$$\mathcal{L}_{\text{train}}(\theta) = \sum_{(\mathbf{x}, \mathbf{y}^*) \in \mathcal{D}_{\text{train}}} \ell\left(y_\theta(\mathbf{x}), \langle \mathbf{y}^* \rangle\right) \quad \text{with } \ell(\mathbf{y}, i) = -\log\left(\frac{\exp(\mathbf{y}_i)}{\sum_{j \in [q]} \exp(\mathbf{y}_j)}\right) \forall \mathbf{y} \in \mathbb{R}^q, i \in [q] \tag{67}$$

### D.1.2. LANGUAGE MODELING

Here, $V = q + 4$, with 4 for the special tokens $\mathcal{S} = \{\times, =, \texttt{bos}, \texttt{eos}\}$. We train the model on the algebra using an auto-regressive approach. For $\mathbf{x} = (\mathbf{u}, \mathbf{v}) \in \mathfrak{A}^2$ with $\mathbf{y} = \mathcal{C}^* \times_1 \mathbf{u} \times_2 \mathbf{v}$, let $s = \langle \texttt{bos} \rangle \langle \mathbf{u} \rangle \langle \times \rangle \langle \mathbf{v} \rangle \langle = \rangle \langle \mathbf{y} \rangle \langle \texttt{eos} \rangle \in [V]^7$. The training is performed by maximizing the likelihood under the direct autoregressive factorization, and the loss (as well as the accuracy) is calculated only on the answer part $s_6 s_7 = \langle \mathbf{y} \rangle \langle \texttt{eos} \rangle$ of the equation. More precisely, let $\phi$ be an encoder that takes a sequence of embedding vectors and returns a hidden representation of size $m^2$, and $\varphi : \mathbb{R}^{m^2} \to \mathbb{R}^V$ be the classifier. For $s \in [V]^7$ and $k \geq 2$, write

$$s^{(k)} = s_1 \cdots s_k \in [V]^k \tag{68}$$

$$\mathbf{E}^{(k)} = [\mathbf{E}_{s_1}, \cdots, \mathbf{E}_{s_k}]^\top \in \mathbb{R}^{k \times d} \tag{69}$$

$$\mathbf{z}^{(k)} = \varphi(\phi(\mathbf{E}^{(k)}) \in \mathbb{R}^V \tag{70}$$

$$\mathbb{P}(i \mid s_{<k}) = \frac{\exp(\mathbf{z}_i^{(k)})}{\sum_j \exp(\mathbf{z}_j^{(k)})} \in \{0, 1\}^V \,\forall i \in [V] \tag{71}$$

The likelihood for $s = s_1 \cdots s_6 s_7 = \langle \texttt{bos} \rangle \langle \mathbf{u} \rangle \langle \times \rangle \langle \mathbf{v} \rangle \langle = \rangle \langle \mathbf{y} \rangle \langle \texttt{eos} \rangle \in [V]^7$ is

$$p_\theta(s_6 s_7 \mid s_{<6}) = \mathbb{P}(s_6 \mid s_{<6}) \mathbb{P}(s_7 \mid s_{<7}) \tag{72}$$

The dataset $\mathcal{D}$ of all possible equations (which has size $q^2$) is randomly partitioned into two disjoint and non-empty sets $\mathcal{D}_{\text{train}}$ and $\mathcal{D}_{\text{test}}$, the training and the validation dataset respectively, following a ratio $r := |\mathcal{D}_{\text{train}}|/|\mathcal{D}|$. The models are trained to minimize the average $\mathcal{L}_{\text{train}}(\theta)$ of the negative loglikelihood $-\log p_\theta(s_6 s_7 \mid s_{<6})$ over $\mathcal{D}_{\text{train}}$ (Equation (73)). We denoted by $\mathcal{A}_{\text{train}}$ the corresponding accuracy ($\mathcal{L}_{\text{test}}$ and $\mathcal{A}_{\text{test}}$ on $\mathcal{D}_{\text{test}}$).

$$\mathcal{L}_{\text{train}}(\theta) = - \sum_{s \in \mathcal{D}_{\text{train}}} \log p_\theta(s_6 s_7 \mid s_{<6}) \tag{73}$$

We use a linear classifier $\varphi(\mathbf{z}) = \mathbf{b} + \mathbf{W}\mathbf{z} \in \mathbb{R}^V$. For $\phi$, we use MLP, LSTM, and Transformer (encoder). The learnable parameters $\theta$ are the union of $\{\mathbf{E}, \mathbf{W}, \mathbf{b}\}$ and the parameters of $\phi$. For LSTM and Transformer, $\phi$ takes the embeddings $\mathbf{z} \in \mathbb{R}^{T \times d}$ and returns a hidden representation of size $m^2$, for instance, the representation of the last element (last token). For MLP, we ignored the special tokens for simplicity, i.e $\phi$ directly takes as input the embedding $\mathbf{z} = \text{vec}\left(\mathbf{E}_{\langle \mathbf{u} \rangle} \oplus \mathbf{E}_{\langle \mathbf{v} \rangle}\right) \in \mathbb{R}^{2d}$. For $L > 1$, we defined $\phi$ by

$$\phi(\mathbf{z}) = g\left(\mathbf{b}^{(L-1)} + \mathbf{W}^{(L-1)} g\left(\cdots g\left(\mathbf{b}^{(2)} + \mathbf{W}^{(2)} g\left(\mathbf{b}^{(1)} + \mathbf{W}^{(1)}\mathbf{z}\right)\right)\cdots\right)\right) \tag{74}$$

where $\mathbf{W}^{(i)} \in \mathbb{R}^{d_{i+1} \times d_i}$ and $\mathbf{b}^{(i)} \in \mathbb{R}^{d_{i+1}}$ for all $i \in [L-1]$, $(d_1, d_L) = (2d, m^2)$. So $\varphi \circ \phi$ is a MLP of $L$ layers with $g$ as activation function.

## D.2. Representation Learning

### D.2.1. PROOF OF PROPOSITON 5.1

**Proposition D.1.** *Assume that $(\mathfrak{A}, \times)$ has a group structure (of size $q$), and denote the inverse of an element $\mathbf{u} \in \mathfrak{A}$ by $\mathbf{u}^{-1}$. Let $\rho : \mathfrak{A} \to \mathbb{R}^{m \times m}$ be a faithful matrix representation of $\mathfrak{A}$. Suppose a model $y_\theta(\mathbf{x}) = \mathbf{W}\phi(\mathbf{x}) \in \mathbb{R}^q$ encodes pairs $\mathbf{x} = (\mathbf{u}, \mathbf{v}) \in \mathfrak{A}^2$ as $\phi(\mathbf{u}, \mathbf{v}) = \rho(\mathbf{u} \times \mathbf{v})$ and decodes with a linear classifier parameterized by weights $\mathbf{W} \in \mathbb{R}^{q \times m^2}$ containing $\rho(\mathbf{w}^{-1})$ for each $\mathbf{w} \in \mathfrak{A}$. Then the predicted label satisfies $\arg\max_{i \in [q]} y_\theta(\mathbf{x})[i] = \mathbf{u} \times \mathbf{v}$. As a consequence, the classifier linearly separates all $q$ outputs.*

*Proof.* Consider the linear classifier $y_\theta(\mathbf{x}) = \mathbf{W}\phi(\mathbf{u}, \mathbf{v})$ acting on $\phi(\mathbf{u}, \mathbf{v}) \in \mathbb{R}^{m^2}$, that is $y_\theta(\mathbf{u}, \mathbf{v})[i] = \langle \mathbf{W}_i, \phi(\mathbf{u}, \mathbf{v}) \rangle$ where the inner product is the Frobenius product on $\mathbb{R}^{m \times m}$. Write $\mathbf{W}_i = \rho(\mathbf{w}^{-1})$ for some $\mathbf{w} \in \mathfrak{A}$, so that

$$y_\theta(\mathbf{u}, \mathbf{v})[i] = \langle \rho(\mathbf{u} \times \mathbf{v}), \rho(\mathbf{w}^{-1}) \rangle = \text{tr}\left(\rho(\mathbf{u} \times \mathbf{v})\rho(\mathbf{w}^{-1})^\top\right) \tag{75}$$

Since $\rho(\mathbf{w}^{-1})^\top = \rho(\mathbf{w}^{-1})$ for an orthogonal (unitary) representation, this reduces to

$$y_\theta(\mathbf{u}, \mathbf{v})[i] = \text{tr}\left(\rho(\mathbf{u} \times \mathbf{v})\rho(\mathbf{w}^{-1})\right) = \chi_\rho\left((\mathbf{u} \times \mathbf{v}) \times \mathbf{w}^{-1}\right) \tag{76}$$

where $\chi_\rho$ is the character of $\rho$. Now, recall that $\chi_\rho$ uniquely distinguishes group elements whenever $\rho$ is faithful. In particular, $\max_{\mathbf{w} \in \mathfrak{A}} \chi_\rho\left((\mathbf{u} \times \mathbf{v})\mathbf{w}^{-1}\right)$ is attained uniquely at $\mathbf{w} = \mathbf{u} \times \mathbf{v}$. Therefore,

$$\arg\max_{i \in [q]} y_\theta(\mathbf{u}, \mathbf{v})[i] = \arg\max_{\mathbf{w} \in \mathfrak{A}} \chi_\rho\left((\mathbf{u} \times \mathbf{v}) \times \mathbf{w}^{-1}\right) = \mathbf{u} \times \mathbf{v} \tag{77}$$

$\square$

### D.2.2. REPRESENTATION QUALITY

Let $\mathbf{f}^{(\ell)}(\mathbf{u}) \in \mathbb{R}^{m_1^2}$ and $\mathbf{f}^{(r)}(\mathbf{u}) \in \mathbb{R}^{m_1^2}$ denote the feature vectors produced by the model for $\mathbf{u} \in \mathfrak{A}$ when $\mathbf{u}$ appears on the left or on the right of an equation, respectively; and let $\mathbf{f}(\mathbf{u}) \in \mathbb{R}^{m^2}$ denote the feature vector of $\mathbf{u}$ produced by the unembedding layer (the classifier). For the first layer of the model (the embedding), we have $\mathbf{f}^{(\ell)}(\mathbf{u}) = \mathbf{f}^{(r)}(\mathbf{u}) = \mathbf{E}_{\langle \mathbf{u} \rangle}$ and thus $m_1^2 = d$. For Transformer, $m_1^2 = d$ for all layers, but for LSTM (resp. MLP), $m_1^2$ is the hidden dimension (resp. number of hidden units in the considered layer), which can differ from the embedding size $d$. We want to determine whether there exists $\mathcal{W} \in \mathbb{R}^{m_1^2 \times m_1^2 \times m^2}$ such that

$$\mathbf{f}(\mathbf{u} \times \mathbf{v}) = \hat{\mathbf{f}}(\mathbf{u} \times \mathbf{v}) := \mathcal{W} \times_1 \mathbf{f}^{(\ell)}(\mathbf{u}) \times_2 \mathbf{f}^{(r)}(\mathbf{v}) = \mathcal{W}_{(3)}\left(\mathbf{f}^{(r)}(\mathbf{v}) \otimes \mathbf{f}^{(\ell)}(\mathbf{u})\right) \quad \forall (\mathbf{u}, \mathbf{v}) \in \mathfrak{A}^2 \tag{78}$$

with $\boldsymbol{\mathcal{W}}_{(3)} \in \mathbb{R}^{m^2 \times m_1^4}$. Here, we are seeking a linear transformation from $\mathbb{R}^{m_1^4}$ to $\mathbb{R}^{m^2}$, with $m_1^4 \approx m^4$ (since $m$ and $m_1$ are of similar order). Because $N = |\mathfrak{A}^2| = q^2$ is not very large, the problem may become trivial if the dimension is too high[10]. To avoid this, we first project $\mathbf{f}^{(\ell)}(\mathbf{u})$ and $\mathbf{f}^{(r)}(\mathbf{v})$ into a lower-dimensional space of size $m_2^2 \leq m$. Formally, we want $\boldsymbol{\mathcal{W}} \in \mathbb{R}^{m_2^2 \times m_2^2 \times m^2}$, $\mathbf{P}^{(\ell)} \in \mathbb{R}^{m_2^2 \times m_1^2}$ and $\mathbf{P}^{(r)} \in \mathbb{R}^{m_2^2 \times m_1^2}$ such that

$$\mathbf{f}(\mathbf{u} \times \mathbf{v}) = \hat{\mathbf{f}}(\mathbf{u} \times \mathbf{v}) := \boldsymbol{\mathcal{W}}_{(3)}\Big(\big(\mathbf{P}^{(r)}\mathbf{f}^{(r)}(\mathbf{v})\big) \otimes \big(\mathbf{P}^{(\ell)}\mathbf{f}^{(\ell)}(\mathbf{u})\big)\Big) = \boldsymbol{\mathcal{W}}_{(3)}\,\mathbf{P}\,\big(\mathbf{f}^{(r)}(\mathbf{v}) \otimes \mathbf{f}^{(\ell)}(\mathbf{u})\big) \quad \forall (\mathbf{u},\mathbf{v}) \in \mathfrak{A}^2 \quad (79)$$

with $\boldsymbol{\mathcal{W}}_{(3)} \in \mathbb{R}^{m^2 \times m_2^4}$ and $\mathbf{P} = \mathbf{P}^{(r)} \otimes \mathbf{P}^{(\ell)} \in \mathbb{R}^{m_2^4 \times m_1^4}$. We do not learn $\mathbf{P}^{(\ell)}, \mathbf{P}^{(r)}$ jointly with $\boldsymbol{\mathcal{W}}$, since this would give $\boldsymbol{\mathcal{W}}$ too much flexibility and risk overfitting (the decomposition (79) could still hold trivially without revealing whether algebra-like structure emerges in the model's features). Instead, we fix $\mathbf{P}^{(\ell)}, \mathbf{P}^{(r)}$ in advance, either by drawing them at random or by using a PCA-based construction. We use the mean squared error $\mathcal{L}_{\text{rep}}$ and the cosine similarity $\mathcal{A}_{\text{rep}}$ associated with solving this problem (i.e. finding $\boldsymbol{\mathcal{W}}$) as measures of how algebra-like structure emerges in the model's feature space.

$$\mathcal{L}_{\text{rep}} = \frac{1}{|\mathfrak{A}^2|} \frac{1}{\sum_{(\mathbf{u},\mathbf{v}) \in \mathfrak{A}^2} \|\mathbf{f}(\mathbf{u} \times \mathbf{v})\|_2^2} \sum_{(\mathbf{u},\mathbf{v}) \in \mathfrak{A}^2} \left\|\mathbf{f}(\mathbf{u} \times \mathbf{v}) - \hat{\mathbf{f}}(\mathbf{u},\mathbf{v})\right\|_2^2 \tag{80}$$

$$\mathcal{A}_{\text{rep}} = \frac{1}{|\mathfrak{A}^2|} \sum_{(\mathbf{u},\mathbf{v}) \in \mathfrak{A}^2} \frac{\langle \mathbf{f}(\mathbf{u} \times \mathbf{v}), \hat{\mathbf{f}}(\mathbf{u},\mathbf{v})\rangle}{\|\mathbf{f}(\mathbf{u} \times \mathbf{v})\|_2 \, \|\hat{\mathbf{f}}(\mathbf{u},\mathbf{v})\|_2} \tag{81}$$

**Random projections** A natural choice is to draw $\mathbf{P}^{(\ell)}, \mathbf{P}^{(r)}$ at random, for example with i.i.d. Gaussian entries followed by row-orthonormalization. Random projections preserve pairwise geometry up to small distortions (Johnson–Lindenstrauss lemma (Johnson & Lindenstrauss, 1984)) and avoid introducing systematic biases. They are also simple, fast to generate, and hard to overfit since they are fixed across training runs. To pick $m_2$, we balance two constraints: (i) avoid the trivial regime where $m_2^4 \gg N$, and (ii) retain enough information from the features. In practice we set $m_2 \approx \min\left(\lfloor m^{1/2} \rfloor, \lfloor (N/\kappa)^{1/4} \rfloor\right)$ with $\kappa \geq 1$ a safety factor. This ensures $m_2^4 \lesssim N$ while keeping $m_2^2$ on the same scale as $m$. To ensure robustness, we repeat the probe with several independent draws of $\mathbf{P}$ and report mean $\pm$ standard deviation of $(\mathcal{L}_{\text{rep}}, \mathcal{A}_{\text{rep}})$.

**PCA projections** When the feature distribution is highly anisotropic, random projections may discard important directions. In this case, we compute $\mathbf{P}^{(\ell)}$ and $\mathbf{P}^{(r)}$ using PCA, fitted separately on $\{\mathbf{f}^{(\ell)}(\mathbf{u})\}_{\mathbf{u} \in \mathfrak{A}}$ and $\{\mathbf{f}^{(r)}(\mathbf{u})\}_{\mathbf{u} \in \mathfrak{A}}$, respectively. We retain the top $m_2^2$ components and whiten them to equalize variances. This choice improves the signal-to-noise ratio by ensuring that the projection focuses on the most informative directions while still enforcing the same dimensionality constraint $m_2^4 \leq m^2$. Thus, PCA projections can complement random projections: the former adapt to data anisotropy, while the latter serve as a neutral and robust baseline.

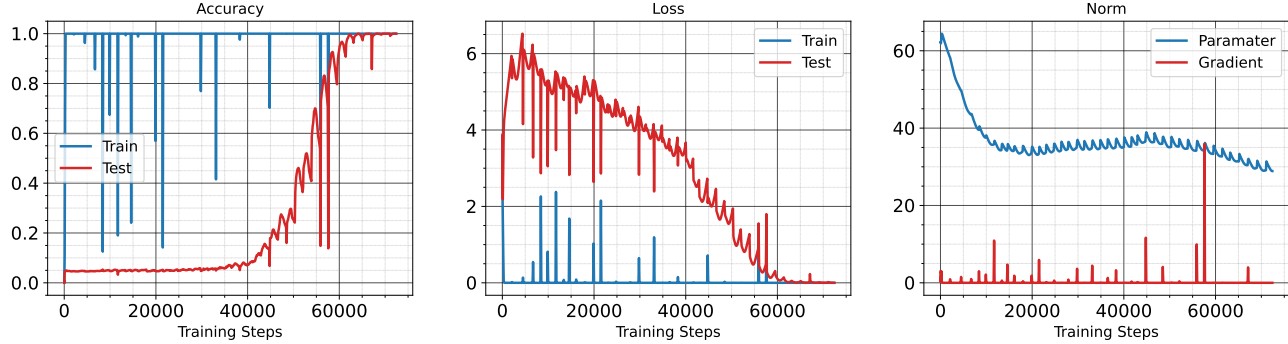

*Figure 8.* Grokking on 2 layers Tansformer trained on the algebra of complex numbers in $\mathbb{Z}/7\mathbb{Z}$ using a training data fraction of $r = 0.5$.

# E. Experimentation Details and Additional Experiments

**MLP** For Figures 1, 2 (a), 3 (a), 4, 5, 6, 7, 11, 12, 14 and 13, $\varphi \circ \phi$ is implemented as a three-layer MLP with ReLU activations and input, hidden, and output dimensions all set to $m^2 = 2^8$. More precisely, we define $\varphi(\mathbf{z}) =$

---

[10]In fact, if $m_1^4 \gg N$, then the system is overparameterized and admits near-perfect solutions regardless of whether the features encode algebraic structure.

$\mathbf{W}\mathbf{z} \in \mathbb{R}^q$, $\mathbf{W} \in \mathbb{R}^{q \times m^2}$ and $\phi(\mathbf{z}) = g\left(\mathbf{b}^{(2)} + \mathbf{W}^{(2)}g\left(\mathbf{b}^{(1)} + \mathbf{W}^{(1)}\operatorname{vec}(\mathbf{z})\right)\right)$ where $\mathbf{W}^{(1)} \in \mathbb{R}^{m^2 \times 2d}$, $\mathbf{b}^{(1)} \in \mathbb{R}^{2d}$, $\mathbf{W}^{(2)} \in \mathbb{R}^{m^2 \times m^2}$, $\mathbf{b}^{(2)} \in \mathbb{R}^{m^2}$, and $g(z) = \max(0, z)$. The embedding dimension is therefore $d = m^2/2$. All models are initialized using PyTorch's default initialization scheme. Training was carried out with the AdamW optimizer, a learning rate of $10^{-2}$, weight decay of $10^{-1}$, and a minibatch size of $2^{10}$. Models were optimized on a NVIDIA Tesla T4 GPU. For Figure 1, the training data fraction is $r = 0.5$.

**Transformer** For Figures 2 (b), 3 (b) and 8, $\phi$ is implemented as a two-layer Transformer Encoder with 2 heads and a hidden/embedding dimension of $d = m_1^2 = 2^6$. The models are initialized using PyTorch's default initialization scheme and trained with the AdamW optimizer, a learning rate of $10^{-3}$, weight decay of $10^{-1}$, and a minibatch size of $2^{10}$.

**LSTM** For Figures 9 and 10, $\phi$ is implemented as a three-layer LSTM with hidden dimension $m_1^2$ equal to the embedding dimension $d = 2^8$. The models are initialized using PyTorch's default initialization scheme and trained with the AdamW optimizer, a learning rate of $10^{-2}$, weight decay of $10^{-1}$, and a minibatch size of $2^{10}$.

**Linear Probing** For the probing experiments (Figures 2, 3 and 10), we apply PCA and retain the top $m_2^2 = m$ components to construct the projection matrices $\mathbf{P}^{(\ell)}, \mathbf{P}^{(r)} \in \mathbb{R}^{m_2^2 \times m_1^2}$ with $m_1 = m$. The problem (79) is then solved using ordinary least squares with $\ell_2$ regularization ($10^{-3}$), implemented via `sklearn`'s Ridge regression.

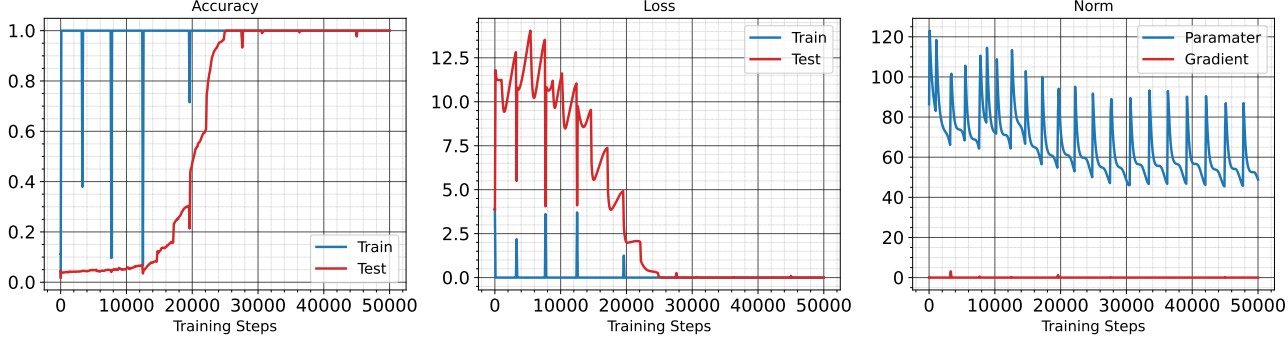

*Figure 9.* Grokking on 3 layers LSTM trained on the algebra of complex numbers in $\mathbb{Z}/7\mathbb{Z}$ using a training data fraction of $r = 0.3$.

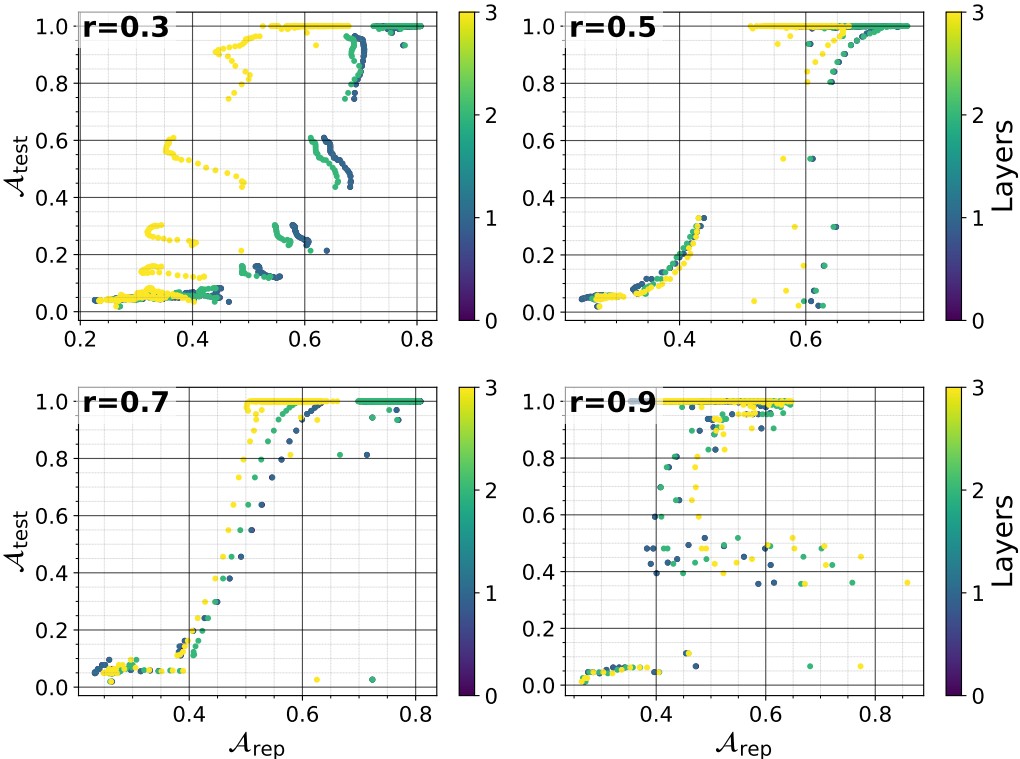

*Figure 10.* Generalization accuracy as a function of representation quality for different training data size ($r$) and model layers (0 for first layer, 1 for the second, etc.) for a LSTM: $\mathcal{A}_{\text{test}}$ increases with $\mathcal{A}_{\text{rep}}$ in a low-data regime.

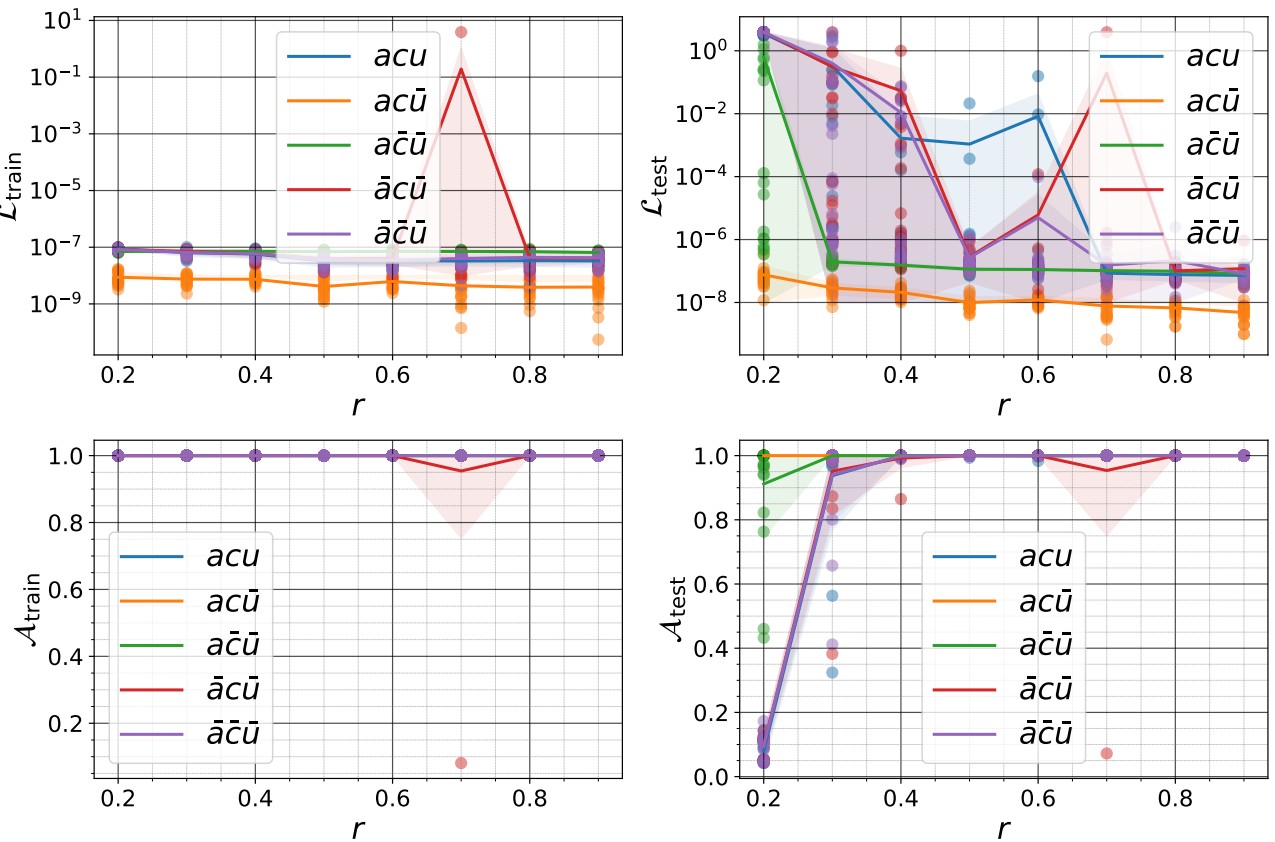

*Figure 11.* Training and test loss and accuracy as a function of the training data fraction.

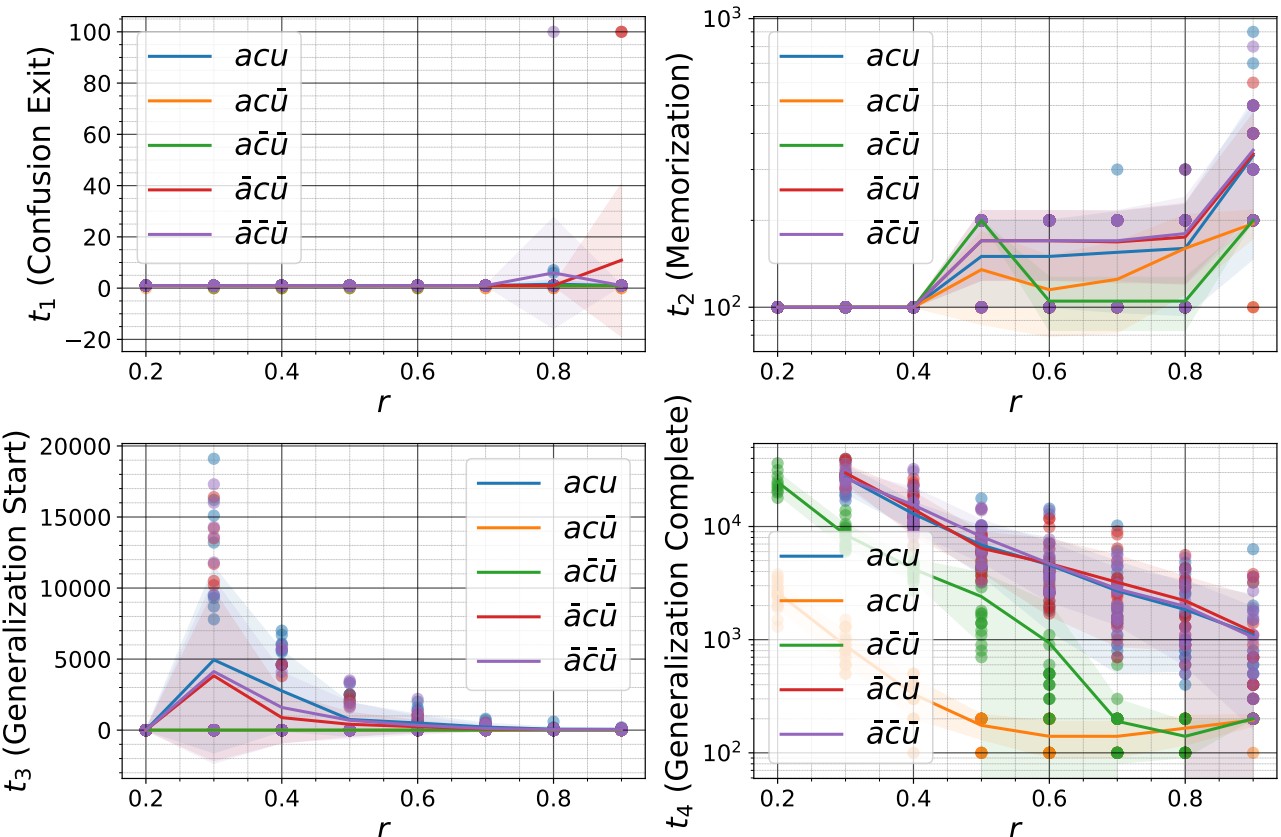

*Figure 12.* Phase transition times as a function of the fraction of data used to train the model: $t_1$ is the step a which the training accuracy $\mathcal{A}_{\text{train}}$ reaches a value strictly greater than $5\%$ for the first time, $t_2$ the step at which $\mathcal{A}_{\text{train}}$ reaches $99\%$ for the first time, $t_3$ the step at which the test accuracy $\mathcal{A}_{\text{test}}$ reaches a value strictly greater than $5\%$ for the first time, and $t_2$ the step at which $\mathcal{A}_{\text{test}}$ reaches $\approx 99\%$ for the first time.

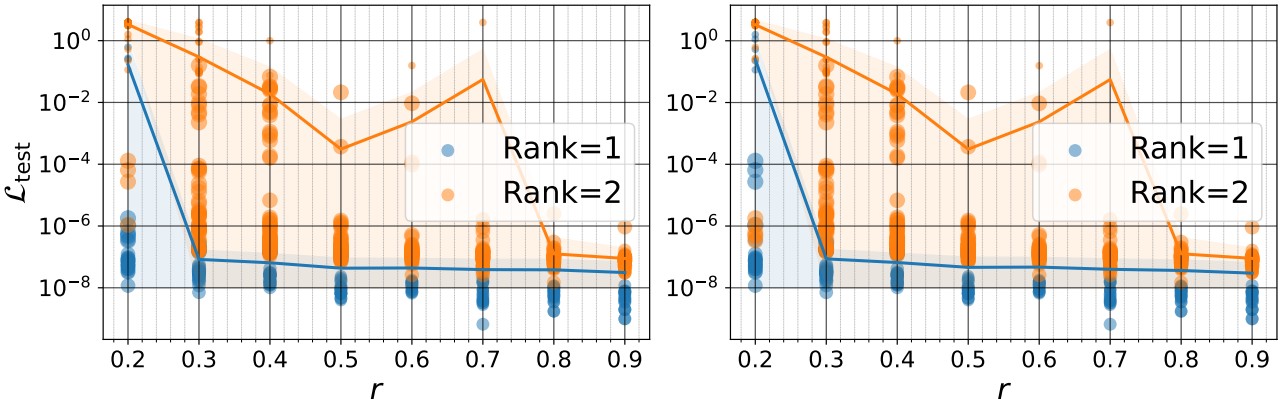

*Figure 13.* Test loss as a function of training data fraction $r$ for $\text{rank}(\mathcal{C}^*_{(1)})$ (left) and $\text{rank}(\mathcal{C}^*_{(2)})$ (right). The sizes of the dots are proportional to $t_4$.

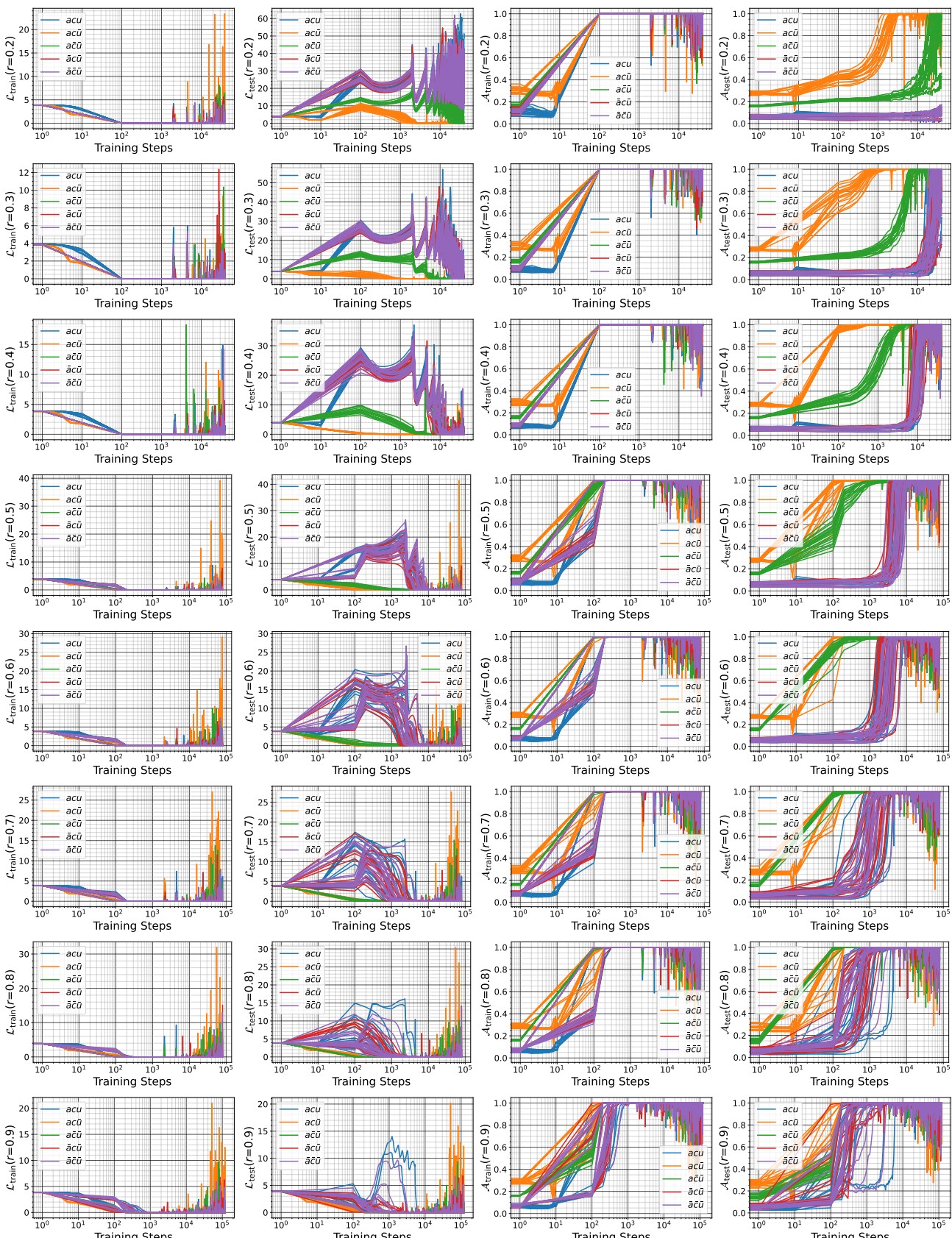

*Figure 14.* Training and test loss and accuracy as a function of training steps for different values of the training data fraction $r$.

