# OpenReview forum: "Grokking Finite-Dimensional Algebra"
_ICML.cc/2026/Conference — ICML 2026 regular_

### Official Review · Reviewer_FTbu · 2026-02-26

**Soundness:** 3
**Presentation:** 3
**Significance:** 3
**Originality:** 4
**Overall Recommendation:** 5
**Confidence:** 4

**Summary:**

This paper expands on prior work which uses the learning of algebraic operations to investigate the grokking phenomenon in neural networks. Instead of simple group operations like modular arithmetic, this work uses multiplication in finite-dimensional algebras (FDAs) over fields, which recovers group structure as a special case. In particular, the authors connect FDA multiplication over finite fields to specific natural representations arising from the FDA's structure tensor. By stratifying FDAs by their mathematical properties (associativity, commutativity, unitality) or that of their structure tensors (sparsity, rank), the authors are able to create learning problems of different difficulties and show how these result in different grokking timelines and generalization behaviors.

**Compliance With Llm Reviewing Policy:**

Affirmed.

**Final Justification:**

The rebuttal addressed my concerns about presentation, and I think the preliminary results regarding grokking on matrix algebras will substantially strengthen the empirical evaluation.

**Key Questions For Authors:**

1. Which theoretical claims, if any, are novel in this paper? Again, I think the paper is sufficiently novel, but this will help contextualize how this paper relies on existing algebraic literature.
2. Can you clarify what algebra is being learned in Section 5.2? Is it the group algebra of some group over $\mathbb{F}_7$?
3. What are the results on Transformers? However, which results correspond to which models is not clear in the main body. Even after checking Appendix E (which does explain which results are from which models), it seems that the Transformer results are missing. What are the results on Transformers? Is there any difference in grokking behavior between model architectures? I think this discussion is important to the clarity of presentation in this paper.
4. Do the models tested grok any examples of associative, noncommutative, unital algebras? In particular, I think results on e.g. the matrix algebra over a finite field would strengthen the empirical evidence in the paper.

**Limitations:**

Yes.

**Strengths And Weaknesses:**

**Strengths**
* FDAs are a novel, natural extension from group operations to studdy grokking using algebraic objects, as the different combinations of properties allow for a stratification of difficulty.
* The main claim of the paper that algebraic structure affects grokking ability is well-substantiated by the experiments, which empirically show different delays in generalization for different classes of FDA.
* Studying how grokking occurs in different regimes and on tasks of different difficulty advances understanding of grokking, which is a phenomenon of central interest.

**Weaknesses**
* Many of the propositions about FDAs are known (for example, their tensorial representations by structure constants and the group algebra of a group over a field are well-known objects). This is not a criticism of the novelty of this work, which I believe lies in the machine learning observations, but it should be made clearer which contributions are from the authors and which are from broader algebraic literature, even if independent proofs are given here.
* I have other concerns about the presentation in this paper. My questions for the authors include the specific points I would like clarified.
* As far as I can tell, all of the experiments use algebras of dimension 2 over $\mathbb{F}_7$. I understand that it is desirable to use a small finite field as the base field and to keep things to a small dimension, but I think this limits the empirical evaluation of the paper. In particular, it admits no FDAs which are associative, unital, and noncommutative, which is (in my opinion) one of the most fundamental classes of examples.

---

> ### Author Rebuttal · Authors · 2026-03-31
>
> We thank the reviewer for the careful and thoughtful review. We appreciate the positive assessment of the novelty of using FDAs and the empirical evidence supporting the role of algebraic structure in grokking. Below, we address the concerns raised.
>
> ### 1. Many propositions about FDAs are known. Clarify novelty.
>
> We agree with the reviewer that several algebraic statements (e.g., structure coefficients, group algebras) are classical. Our contribution is not to introduce new algebraic results per se, but to develop a tensorial formulation of finite-dimensional algebras that is directly amenable to representation learning and empirical analysis.
>
> In particular:
>
> *  We establish a concrete correspondence between $n\mathbb{F}$-FDAs and structure tensors $C \in \mathbb{F}^{n\times n\times n}$, allowing us to work directly with tensors rather than abstract algebraic objects.
>
> * We express algebraic properties (associativity, commutativity, unitality, etc.) as explicit constraints on $C$, which enables systematic generation of learning tasks with controlled structure.
>
> * The novelty of the paper lies in connecting these algebraic structures to grokking dynamics, and in empirically demonstrating how different algebraic properties affect generalization behavior.
>
>
> We will revise the paper to clearly distinguish standard algebraic facts from our contributions.
>
>
> ### 2. Clarify the algebra in Section 5.2.
>
> We thank the reviewer for pointing this out. In Section 5.2, the algebra being learned is the algebra of complex numbers over $\mathbb{F}_7$, i.e., elements of the form $a + b\mathbf{i}$  with $a,b \in \mathbb{F}_7$ and $\mathbf{i}^2 = -1$ (Example 2.1).
> We will clarify this explicitly in the main text, as we agree that this was not sufficiently clear.
>
> ### 3. Missing  Transformer results.
>
> We agree that the presentation of Transformer results is currently missing. While experiments with Transformer models were conducted, they were not reported in the manuscript.
> We will include these results in the revision. Empirically, we observe that Transformers exhibit similar grokking behavior to MLPs and LSTMs, with the main difference being increased oscillations in the training loss before convergence. We will clarify these results and ensure that all figures clearly indicate which architecture is used.
>
> ### 4. Limited empirical evaluation.
>
> We agree that restricting most experiments to $n=2$ over $\mathbb{F}_7$ limits the scope of the empirical evaluation.
> The space of possible structure tensors grows rapidly, with $|\mathbb{F}_p^{n \times n \times n}| = p^{n^3}$. As a result, exploring this space becomes computationally challenging even for moderate values of $n$ and $p$. We therefore chose a reasonable $(n,p)$ to enable a controlled and systematic study of how algebraic properties affect generalization and grokking behavior.
>
> That said, we agree that extending beyond this setting would strengthen the paper. In particular, we will include additional experiments on associative, noncommutative, unital algebras, such as matrix algebras over finite fields (which arise when $n$ is a perfect square, e.g., $n=2^2=4$), and are not captured in the current $n=2$ setup. In fact, the minimum dimension $n$ to get an associative, unital, and non-commutative algebra over any field $\mathbb{F}$ is $3$.
>
> Preliminary experiments indicate that these algebras also exhibit grokking behavior, and we will include these results in the revision. For example:
>
> * with the algebras of $2\times 2$ matrices ($n=4$) over $\mathbb{F}_3$,  $(t_2, t_4)=(200,5200)$ for $r=0.5$ and $(t_2, t_4)=(100,7300)$ for $r=0.4$.
>
>   Here, $t_2$ and $t_4$ are the memorization and the generalization steps, respectively; and $r \in (0, 1)$ is training data-fraction.
>
> * with the algebras of $2\times 2$ upper triangular matrices ($n=3$) over $\mathbb{F}_5$,  $(t_2, t_4)=(200,7100)$ for $r=0.5$ and $(t_2, t_4)=(300,10400)$ for $r=0.4$.
>
> ### 5. General presentation concerns.
>
> We appreciate the reviewer’s comments on presentation and clarity. We will revise the manuscript to:
> * better distinguish classical algebraic material from our contributions,
> * clarify the construction of datasets and algebraic examples,
> * improve the presentation of experimental results (including clearer labeling of model architectures).
>
> Overall, we thank the reviewer for the constructive feedback. We believe that addressing these points will significantly strengthen the clarity and empirical coverage of the paper.

---

> > ### Author Rebuttal · Reviewer_FTbu · 2026-04-02
> >
> > Thank you for your detailed response. The reported preliminary results on matrix algebras seem promising, and I certainly encourage the authors to include the mentioned changes. I will raise my score accordingly.

---

### Official Review · Reviewer_hB5Q · 2026-03-08

**Soundness:** 3
**Presentation:** 3
**Significance:** 4
**Originality:** 4
**Overall Recommendation:** 5
**Confidence:** 3

**Summary:**

The paper applies "Grokking", which is the phenomenon where a Neural Net suddenly gain perfect or close to perfect generalisation after previously just overfitting the data.
The authors train a small Transformer model on algorithmically generated datasets. Those come from modular and symmetric group operations. Then it is trained past the overfitting points (sometimes considerably longer).
The authors find that this is highly sensitive to all sort of circumstances, like what optimizer is used, the regularisation, and technieques like weight decay.

**Compliance With Llm Reviewing Policy:**

Affirmed.

**Final Justification:**

Thanks, the rebuttal addressed my concern.

**Key Questions For Authors:**

(1) Could you also observe similar Grokking behaviour in more complex/structured mathematical constructions?
What you currently seems to be essentially one relation. But one could expand to other areas, like for instance (just from the top of my head): Graph Theory: learning bipartedness or shortest-path distances from a adjacency matrix representation.  or operations of $GL_n(\mathbb{F}_p)$) or evaluating integer polynomials of higher degrees or yet something completely different.

If the authors already plan steps in this direction, that would potentialy increase my "Significance" score.

**Limitations:**

yes

**Strengths And Weaknesses:**

Soundness:
The paper is very empirical and technically sound. The design of the experiment makes sense and the authors seems to taken various ablations into account. They report their findings in what seems like objective, also mentioning limitations like the fact that this is a toy problem and it is not clear how this translates to non-artificially made datasets.

Presentation:
The presentation is great! The figures immediately show what is going on (although some of them are very tiny, but fortunately vector graphics so I can just zoom in...). They also did a good job explaining the mathematical objects in the training data.

Significance:
Considering that there is an "early stopping" paradigm, it this paper gives an insightful sandbox to study grokking. Notable the aspect of interpretability, i.e. why after further training the learned internal structures appear to get more simple, or "generalisable" seems significant.

Originality:
Perhaps the originality is provided by pushing optimization $10^3$ to $10^6$ steps past the point of severe overfitting to see what happens. The FDAs are a beatifully crafted playground.

---

> ### Author Rebuttal · Authors · 2026-03-30
>
> We thank the reviewer for their review of our work. Below, we answer the question raised.
>
> ## (1) "Could you also observe similar Grokking behaviour in more complex/structured mathematical constructions? What you currently seems to be essentially one relation. But one could expand to other areas, like for instance (just from the top of my head): Graph Theory: learning bipartedness or shortest-path distances from a adjacency matrix representation. or operations of $GL_n(\mathbb{F}_p)$ or evaluating integer polynomials of higher degrees or yet something completely different."
>
> We thank the reviewer for this insightful question. We would like to clarify that, in our work, the focus is not on group operations or specific architectures, but on a more general framework based on finite-dimensional algebras (FDAs), which strictly generalize group-based constructions. In particular, our experiments are not limited to group structures but explore a broader class of algebraic operations defined via structure tensors.
>
> Our goal is precisely to move beyond the setting of groups and study grokking in a more general algebraic framework. In this sense, the current experiments can be viewed as a controlled setting where a single binary operation (defined by the algebra) determines the learning task, allowing us to isolate the effect of algebraic structure on generalization.
>
> Regarding the reviewer’s suggestion, we fully agree that extending to more complex and structured constructions is an important direction. Importantly, our framework is well-suited for such extensions, since any algebraic structure that can be expressed through a finite operation (or tensor) can be incorporated.
>
> For instance:
> * **Matrix groups such as $GL_n(\mathbb{F}_p)$**: these can be naturally represented through their induced algebraic operations, leading to higher-dimensional structure tensors (see Example B.5 in Appendix B).
>
> * **Polynomial evaluation**: this corresponds to learning structured (potentially higher-order) mappings, which can be viewed as extensions of the current tensorial framework (see Example B.4 in Appendix B).
>
> *  **More complex compositional structures**: while not always strictly binary operations, they can often be reformulated in terms of structured mappings, and we believe the notion of alignment with an underlying structure remains relevant.
>
> We agree that exploring such richer settings would further strengthen the significance of the work, and we plan to investigate these directions in future work, as highlighted in the conclusion.

---

> > ### Author Rebuttal · Reviewer_hB5Q · 2026-04-02
> >
> > all concerns addressed, updated my significance score

---

### Official Review · Reviewer_oEYE · 2026-03-13

**Soundness:** 3
**Presentation:** 3
**Significance:** 3
**Originality:** 3
**Overall Recommendation:** 5
**Confidence:** 2

**Summary:**

The authors leverage the phenomenon of "grokking" to explore operations beyond those on group elements and loosen the notoriously constrictive group axioms assumed in earlier works to encapsulate a wider range of algebraic structures.

**Compliance With Llm Reviewing Policy:**

Affirmed.

**Final Justification:**

I raised my score because of the promise to include a better introduction/method section to the paper and improve readability. This will make the paper more accessible. The authors also included a nice explanation of how the algebraic training/testing procedure works, improving my understanding of the paper overall. I believe this is a strong paper and the authors did a great job.

**Key Questions For Authors:**

- Is it possible to show some samples of the data the model is training on in a "Figure 1"-type schematic so readers who are not familiar with the topic can visually see what is going on in the experiments? I think this would make the paper more accessible and much stronger. Even a table with samples could help.

**Limitations:**

No. I can't find a discussion on limitations of this work by the authors.

**Strengths And Weaknesses:**

Strengths:
- The paper is immaculately edited, it is pleasing to read and the language and figures are clear.
- The scope is ambitious and the results look very good.
- The experimental setup is thorough.

Weaknesses:
- The plots in Figure 4 are a bit crowded.
- What is missing is a discussion on limitations.
- The paper is a bit unclear for people who are not familiar with the topic.

---

> ### Author Rebuttal · Authors · 2026-03-30
>
> We thank the reviewer for the careful review of our work. We appreciate that you found our work a pleasure to read and that our results sound. Below, we address the concerns raised as weaknesses of our work and answer the question posed.
>
> ### 1. "The plots in Figure 4 are a bit crowded"
>
> We thank the reviewer for this feedback. We agree that Figure 4 can be improved for readability. In the revision, we will simplify the visualization by splitting the plots across multiple figures and increasing font sizes and spacing to improve clarity.
>
> ### 2. "What is missing is a discussion on limitations"
>
> We thank the reviewer for pointing this out. We note that limitations were briefly discussed in the conclusion; however, we agree that this discussion is not sufficiently explicit or prominent. In the revision, we will add a dedicated **Limitations** section to clearly highlight the scope and boundaries of our work.
>
> In particular, our current study is limited to relatively small finite-dimensional algebras over finite fields, and scaling to larger dimensions remains an open challenge. From a theoretical perspective, a tighter connection between tensor properties (e.g., rank, coherence) and sample complexity is left for future work. From an empirical perspective, extending the framework to real-world tasks with latent algebraic structure is also an important direction for future work.
>
> ### 3. "The paper is a bit unclear for people who are not familiar with the topic".
>
> We thank the reviewer for pointing this out. We will improve accessibility in the revision by adding a more intuitive overview of finite-dimensional algebras and their tensor representations in the introduction, along with additional explanations and examples to guide readers less familiar with the topic.
>
> ### 4. "Is it possible to show some samples of the data the model is training on in a "Figure 1"-type schematic so readers who are not familiar with the topic can visually see what is going on in the experiments? I think this would make the paper more accessible and much stronger. Even a table with samples could help".
>
> We agree that including concrete examples would make the experimental setup more accessible. In the revision, we will add a schematic illustration showing how input pairs $(u,v)$ are mapped to outputs $u \cdot v$. This will help readers better visualize the learning task.
>
> Concretely, in our experiments over a finite field $\mathbb{F}_p$, each algebra element $u \in \mathfrak{A}$ is treated as a discrete symbol in a vocabulary of size $q = |\mathfrak{A}|$. The model does not take the raw coordinates of $u$ and $v$ as input. Instead, each element is assigned an index $\langle u \rangle \in [q] := \\{1, 2, \dots, q \\}$, and the input is the sequence of symbols corresponding to the equation format used in the paper, e.g. $(u,\times,v,=)$.
> Each symbol is represented by a one-hot vector (or equivalently by its learned embedding through the embedding matrix). The MLP then receives the concatenation of these embeddings and outputs logits over the vocabulary, which are converted into a probability distribution over all possible output symbols.
>
> For example, consider the simple $2\mathbb{F}_7$-FDA with basis $\{a^{(1)},a^{(2)}\}$ and product
> $$a^{(1)}\cdot a^{(1)}=0,\ a^{(1)}\cdot a^{(2)}=a^{(2)}, \ a^{(2)}\cdot a^{(1)}=6a^{(2)},\quad a^{(2)}\cdot a^{(2)}=0$$
> This algebra has $q=|\mathfrak{A}|=7^2=49$ elements, since every element has the form $u = \alpha a^{(1)} + \beta a^{(2)}$ with $\alpha,\beta \in \mathbb{F}_7 \simeq \\{0, 1, \dots, 6 \\}$.
> Suppose we index the elements of $\mathfrak{A}$ as vocabulary symbols. Consider for example $u=a^{(1)}$ and $v=a^{(2)}$. Since  $a^{(1)}\cdot a^{(2)} = a^{(2)}$, the corresponding supervised example is:
> $$
> \text{input: } (\langle a^{(1)}\rangle,\langle \times\rangle,\langle a^{(2)}\rangle,\langle =\rangle),
> \qquad
> \text{target: } \langle a^{(2)}\rangle$$
>
> The MLP processes such an input by first looking up the embedding of each input symbol, concatenating these embeddings, and passing the result through hidden layers. Its final layer outputs a vector in $\mathbb{R}^q$, where the $i$th coordinate is the logit associated with the $i$th algebra element in the vocabulary. After softmax, this becomes a probability distribution over all possible outputs. Training minimizes the cross-entropy loss so that the highest-probability token corresponds to the correct product $u\cdot v$.
>
> We will include such explicit examples in the revision to make the learning task more visually and conceptually accessible for readers who are less familiar with the algebraic setup.
>
> 5. "No. I can't find a discussion on limitations of this work by the authors".
>
> We agree with this observation. While some limitations were briefly mentioned in the conclusion, we acknowledge that this was not sufficiently explicit. As noted above, we will add a dedicated **Limitations** section to clearly present the scope and boundaries of our work.

---

> > ### Author Rebuttal · Reviewer_oEYE · 2026-04-03
> >
> > The authors addressed all the points raised in the review and promise to improve the accessiblity, readability, and methods in the paper.
> >
> > Thanks for the explanation and detailed responses. I have re-evaluated the paper and decided to increase my score to an "accept".

---

### Official Review · Reviewer_8EaT · 2026-03-19

**Soundness:** 3
**Presentation:** 3
**Significance:** 3
**Originality:** 3
**Overall Recommendation:** 5
**Confidence:** 3

**Summary:**

The paper studies grokking phenomenon in finite dimension algebra, extending such previous studies on group arithmetic. The bulk of the theory is proving properties of finite dimensional algebra representation as a structure constant tensor and how to build a FDA from a group. Experiments are performed to characterize grokking for FDAs over finite fields; and how different properties of the algebra (associativity, commutativity, unital) affect generalization.

**Compliance With Llm Reviewing Policy:**

Affirmed.

**Final Justification:**

Authors agreed to address my concerns in a revised version of the paper.

**Key Questions For Authors:**

See Weaknesses

**Limitations:**

See Weaknesses

**Strengths And Weaknesses:**

Strength:
- As far as the reviewer is aware, the study of grokking on FDAs is entirely novel.
- The theory proven are sound.
- Extensive experiments are carried out to observe both grokking and generalization over different algebraic properties. The latter is especially interesting since it is unexpected for specific properties to predictably affect generalization if the model is only memorizing its training data.

Weakness:
- Most of the theory (which spans roughly half the paper, in length) is rather straightforward. Materials in this section can be looked up in a Wikipedia articles on 'Structure constant' or 'Algebra over field'. While that in itself is not a problem, it is not clear which part of the theory is contributed in a novel fashion by the authors; and which part of the theory has already exists in literature (in which case, the statements should be stated as fact/theorem rather than 'Proposition', which implies that the statement is new). The authors should make this very clear in their texts.
-  The authors built a theory for FDA but only test them on algebras over finite fields, which is essentially finite groups with additional structures. More interesting FDA, such as the Lie algebras should also be tested.
- The authors should explain how they find the tensor $\mathcal{W}$ in the final paragraph of page 6 (which is necessary to compute the cosine similarity $\mathcal{A}_rep$. Do you optimize the cosine similarity and report the final objective value? What were the algorithms used. This part is essential to link grokking with the algebraic property (and is one of the few key links between the theory and experiments of the paper).

The paper is interesting but there are some questions about originality of the theoretical claims, and links between the theory and the empirical studies. Therefore I am giving a low accept score but would be willing to discuss more and raise the score in the rebuttal if the concerns are sufficiently addressed.

---

> ### Author Rebuttal · Authors · 2026-03-30
>
> We thank the reviewer for the careful and constructive review. We appreciate that you found our work novel and sound. Below, we address the concerns raised as a weakness of our work.
>
> ### 1. "Most of the theory (which spans roughly half the paper, in length) is rather straightforward....".
>
> We thank the reviewer for raising this important point. We agree that parts of Section 3 rely on classical notions from algebra (e.g., structure constants, algebras over a field), and we will revise the paper to make this distinction explicit.
> Our goal in this section is not to introduce new results in abstract algebra per se, but to develop a formulation of finite-dimensional algebras that is directly amenable to analysis in the context of representation learning and grokking.
>
> More precisely, our contributions in this section are:
> * We establish a concrete tensorial representation of finite-dimensional algebras by identifying any $n\mathbb{F}$-FDA with a structure tensor $C \in \mathbb{F}^{n \times n \times n}$ in a fixed basis, and conversely. This allows us to work directly with tensors $C$ rather than abstract algebraic objects, which is essential for computational and learning-based analysis.
> * We express algebraic properties (associativity, commutativity, unitality, etc.) as explicit polynomial constraints on $C$. While these properties are classical, their unified tensorial formulation provides a concrete, operational characterization that can be implemented and verified directly.
> * This formulation enables a direct bridge between algebraic structure and neural representations, which is central to our study of grokking. In particular, the tensor $C$ serves as a target structure that learned representations align with, allowing us to quantify generalization via representation-alignment metrics (Section 5).
>
> To avoid any ambiguity, we will revise the paper to explicitly distinguish standard algebraic facts from our contributions, and clarify that the novelty lies in the tensorial formulation and its role in connecting algebraic structure to learning dynamics.
>
> ### 2. "The authors built a theory for FDA but only test them on algebras over finite fields, which is essentially finite groups with additional structures. More interesting FDA, such as the Lie algebras should also be tested."
>
> We would like to clarify that finite-dimensional algebras over finite fields are, in general, strictly more general than finite groups. While groups can be embedded into our framework, the class of FDAs we consider also includes non-associative and non-unital structures, and their multiplication does not, in general, define a group law on the underlying set.
>
> Also, Lie algebras are naturally included in the family of FDAs we study experimentally, and in particular, they arise as special cases in Section 5.3 when varying algebraic properties.
> In fact, A Lie algebra $(\mathfrak{A}, \cdot)$ is an algebra whose binary operation $\cdot$ (often denoted $[\cdot,\cdot]$,  the Lie bracket) satisfies the alternating property $u \cdot u = 0 \ \forall u \in \mathfrak{A}$ and the Jacobi identity $(u\cdot v)\cdot w + (v\cdot w)\cdot u + (w\cdot u)\cdot v = 0 \ \forall u,v,w \in \mathfrak{A}$.
> In our tensorial formulation, this corresponds to a structure tensor $C \in \mathbb{F}^{n \times n \times n}$ satisfying $C_{ijk} = -C_{jik} \quad \forall i,j,k \in [n]$ and $\sum_{k=1}^n C_{ijk} C_{k\ell m} + \sum_{k=1}^n C_{j\ell k} C_{kim} + \sum_{k=1}^n C_{\ell i k} C_{kjm} = 0$ $\forall i,j,\ell,m \in [n]$.
>
> For example, the $2\mathbb{F}_7$-FDA $(\mathfrak{A},\cdot)$ defined in the basis $\{a^{(1)},a^{(2)}\}$ by
> $$a^{(1)}\cdot a^{(1)} = a^{(2)}\cdot a^{(2)} =0,\ a^{(1)}\cdot a^{(2)}=a^{(2)},\ a^{(2)}\cdot a^{(1)}=6a^{(2)}$$
> is a $2$ dimensionnal Lie algebra over $\mathbb{F}\_7$,  with structure tensor $C\_{:,:,1} = [[0, 0], [0, 0]]$ and  $C\_{:,:,2} = [[0, 1], [6, 0]]$.
>
> Such algebras are included in the experimental family studied in Section 5.3 (for $n=2$, $p=7$). We agree that this point was not sufficiently explicit in the current draft, and we will clarify that our experiments cover FDA classes beyond group-derived examples, including Lie-type algebras.
>
> ### 3. "The authors should explain how they find the tensor $\mathcal{W}$  in the final paragraph of page 6 ...".
>
> The method to obtain $\mathcal{W}$ is described in Sections D.2.2 and E of the Appendix.
> Concretely, $\mathcal{W}$ is obtained by minimizing a squared reconstruction error $\sum_{(u,v)} \left\| f(u \times v) - \hat{f}(u,v) \right\|_2^2$, where $\hat{f}(u,v)$ is a bilinear model parameterized by $\mathcal{W}$.
> This is solved using ordinary least squares with $\ell_2$ regularization (Ridge regression, via sklearn).
>
> So, we do not directly optimize the cosine similarity $A_{rep}$; instead, $\mathcal{W}$ is first fitted via this regression objective, and the cosine similarity is then computed as an evaluation metric.
> We will clarify this procedure in Section 5.2 of the main text.

---

> > ### Author Rebuttal · Reviewer_8EaT · 2026-04-04
> >
> > I would like to thank the authors for the response. I have no further questions and have raised my score.

---

### Decision · Program_Chairs · 2026-04-30

**Decision:**

Accept (regular)

**Comment:**

The authors consider the problem of grokking for learning multiplication in finite dimensional algebras, beyond group operations, commutative operations, or associative operations. They consider both real and discrete algebras, and they experimentally study how the algebraic properties affect grokking, and how generalization depends on whether learned latent representations are aligned with the algebraic representations.

All the reviewers found the results very interesting, novel, and original. They are well motivated and well presented, and relevant and of great interest to the ICML community. I also agree with them, and I suggest acceptance. However, this is only conditional on addressing the reviewers’ concerns, adding the promised changes to the next version, as well as the explanations in the rebuttal phase:



> Reviewer oEYE: I raised my score because of the promise to include a better introduction/method section to the paper and improve readability. This will make the paper more accessible. The authors also included a nice explanation of how the algebraic training/testing procedure works, improving my understanding of the paper overall. I believe this is a strong paper and the authors did a great job.

> Reviewer FTbu: I think the preliminary results regarding grokking on matrix algebras will substantially strengthen the empirical evaluation.